# Clarifying the effect of biodiversity on productivity in natural ecosystems with longitudinal data and methods for causal inference

Laura E. Dee [1] ✉, Paul J. Ferraro [2,3] ✉, Christopher N. Severen[4], Kaitlin A. Kimmel[1], Elizabeth T. Borer[5], Jarrett E. K. Byrnes[6], Adam Thomas Clark[7], Yann Hautier[8], Andrew Hector[9], Xavier Raynaud[10], Peter B. Reich [11,12,13], Alexandra J. Wright[14], Carlos A. Arnillas[15], Kendi F. Davies[1], Andrew MacDougall[16], Akira S. Mori[17], Melinda D. Smith [18,19], Peter B. Adler [20], Jonathan D. Bakker [21], Kate A. Brauman [22], Jane Cowles [5], Kimberly Komatsu [23], Johannes M. H. Knops[24], Rebecca L. McCulley [25], Joslin L. Moore [26], John W. Morgan[27], Timothy Ohlert[18], Sally A. Power [13], Lauren L. Sullivan[28,29], Carly Stevens [30] & Michel Loreau[31]

Causal effects of biodiversity on ecosystem functions can be estimated using experimental or observational designs – designs that pose a tradeoff between drawing credible causal inferences from correlations and drawing generalizable inferences. Here, we develop a design that reduces this tradeoff and revisits the question of how plant species diversity affects productivity. Our design leverages longitudinal data from 43 grasslands in 11 countries and approaches borrowed from fields outside of ecology to draw causal inferences from observational data. Contrary to many prior studies, we estimate that increases in plot-level species richness caused productivity to decline: a 10% increase in richness decreased productivity by 2.4%, 95% CI [−4.1, −0.74]. This contradiction stems from two sources. First, prior observational studies incompletely control for confounding factors. Second, most experiments plant fewer rare and non-native species than exist in nature. Although increases in native, dominant species increased productivity, increases in rare and non-native species decreased productivity, making the average effect negative in our study. By reducing the tradeoff between experimental and observational designs, our study demonstrates how observational studies can complement prior ecological experiments and inform future ones.

Motivated by global changes in biodiversity, ecologists have advanced our understanding of the consequences of biodiversity change for ecosystem functioning[1–11]. One particularly active area of this research has focused on how plant species diversity affects ecosystem productivity[1–4,12]. To shed light on this causal relationship, studies have used both experimental and non-experimental designs, each of which presents distinct advantages and disadvantages for elucidating causal relationships in natural ecosystems.

**A. The Challenge of Generalizability in Experimental Designs**

**B. The Challenge of Causal Inference in Observational Designs**

**Fig. 1 | Challenges in estimating the causal effect of species richness on productivity. A** Experimental designs permit credible causal inferences with few modeling assumptions. Yet experiments often manipulate richness in random permutations, plant limited sets of species, and weed out colonizing species. Such designs can yield ecological processes that differ from processes in natural systems. In **A**, when common species are more likely to be planted in experiments than rare species, the proportion of common species is higher than the proportion of rare species regardless of the planted richness level. In contrast, in natural communities, higher species richness is associated with greater numbers of rare species than common species. **B** Observational designs include natural processes but causal inferences are challenged by confounding variables (U) associated with both richness (R) and productivity (P); e.g., precipitation can increase both R and P, thereby inducing a positive correlation between the two, even if the true causal relationship were zero or negative. Some of these confounding variables, like topography, may be time-invariant (or slow-changing) over the study period at the level of the plot ($U_p$) or site ($U_s$). Others may be time-varying at the site ($U_{st}$), such as weather, or the plot ($U_{pt}$), such as micro-climate. To estimate the effect of R on P without bias does not require data on variables I that only affect P, or Z that only affect R, nor on mechanisms (M), such as selection or complementarity. However, data on Z and M can help address unobserved confounders and differentiate the effect of R on P from the effect of P on R.

Experimentalists that manipulate plant species richness often infer that increases in richness cause increases in biomass[1,2,7,13]. Although experimental manipulations facilitate causal inferences, most experiments that manipulate richness are designed to test theory[14,15] rather than to simulate how species richness changes in natural ecosystems[16–18]. If the effect of richness on productivity depends on the specific species gained and lost, and how they are gained and lost, inferences from experiments may not generalize to natural ecosystems (Fig. 1A)[16,17,19]. For example, many biodiversity experiments simulate random gains and losses of species (but see[20–22]), which may not mimic changes in species richness in nature. Moreover, most experiments plant common, native species (but see[16,19–21,23]). However, in diverse natural ecosystems, most species are rare[24] and non-native species are increasingly prevalent[25].

Observational studies can capture the consequences of changes in species richness that occur in nature. However, determining the causal effect of richness on productivity in observational studies requires strong assumptions[26,27]. Confounding variables associated with both richness and productivity[15] can mask or mimic a causal relationship between them (Fig. 1B). For example, unobserved differences in soil nitrogen across locations can mask a positive relationship between richness and productivity if more nitrogen reduces richness and increases productivity[28]. To eliminate confounding effects, common study designs in ecology require identifying, measuring, and statistically controlling for all confounding variables[29]. This task is daunting in natural ecosystems given myriad confounding variables that could influence both richness and productivity (e.g., land-use history, herbivory, disturbance). Yet, failure to control for all confounding variables can lead to inferences of the wrong sign or magnitude (i.e., due to statistical bias)[26,30]. Consequently, the mixed evidence on the effect of species richness on productivity in observational studies[3,4,12,13,31,32] may reflect differences in the degree of control for confounding factors across studies.

To isolate and quantify causal relationships between biodiversity and ecosystem function, the ideal study design would combine the strength of experiments in enabling causal inferences from correlations with the strength of observational designs in facilitating generalizable inferences about natural ecological processes. Experimental designs with more realistic extinction processes are one step in that direction (e.g.,[16,20,21]). Here, we develop a complementary approach by leveraging a global grassland dataset[33] and methods designed for inferring causality from observational data[26,27,34,35]. This suite of methods now comprises the dominant approach to causal inference in fields outside of ecology, such as economics, medicine, and public health. When combined with our global longitudinal dataset, they allow us to account for the ecological complexity of grasslands without making strong assumptions about our ability to measure all confounding variables[36], and they allow us to isolate the effect of biodiversity on productivity separate from the reverse relationship[35,37].

Applying traditional methods to our data, we would conclude that, on average, an increase in biodiversity increases productivity in grasslands, a result found in many prior studies. However, applying the suite of methods that control for a broader set of confounding variables, we come to the opposite conclusion: an increase in biodiversity reduces productivity in grasslands, on average.

## Results and discussion
### Study context and design
We use repeated observations between 2007-2017 from 151 unmanipulated plots in 43 grassland sites in 11 countries[33] from the Nutrient Network (https://nutnet.org), including mesic grasslands and prairies, savanna, desert grasslands, montane meadows, old fields, and alpine tundra (Table S1 in Supplementary Information (SI); SI Section 3). We define "productivity" as aboveground live biomass per year per 1m² (following refs. 3,9,31). Each 1 m² plot has between 1 and 37 species in a year, with an average of 11.3 (SD = 5.7) and median of 10. We use plots with five or more years of data, in contrast to most observational studies of biodiversity effects on productivity, which use a single year[3,9,12,31,38]. Data from multiple years offer three advantages: (1) an opportunity to study natural changes in richness; (2) enhanced generalizability; and (3) ways to control for a broad set of confounding variables, including unobserved ones (see "Methods").

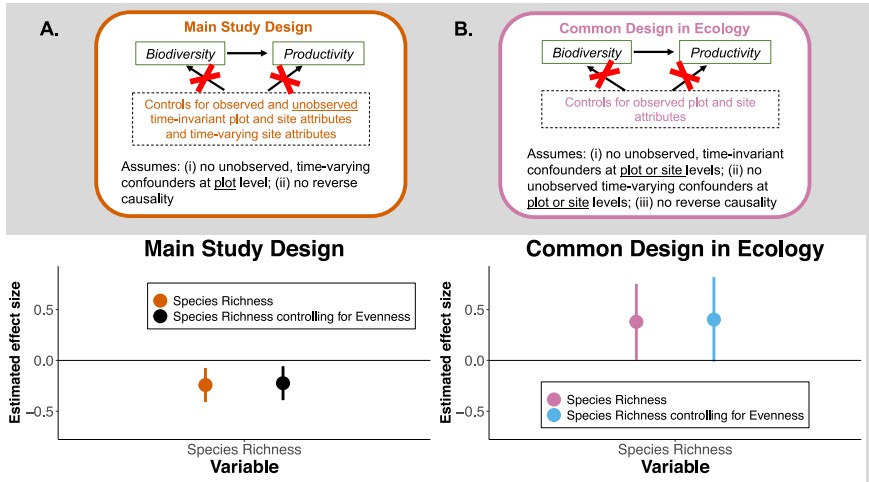

**Fig. 2 | The effect of biodiversity on productivity, estimated as the mean % change in productivity from a 1% increase in richness.** The top panel summarizes the assumptions needed for drawing causal inferences in each design. A red X on an arrow implies that the design blocks the confounding pathways described in the box with dotted lines. The *Common Design*, a multivariate mixed model that is common in ecological analyses of observational data, requires much stronger assumptions to interpret estimates as a causal effect than our *Main Design*, assumptions that are unlikely to be met in these data. The bottom panel shows the estimates with 95% confidence intervals for two designs: **A** Our *Main Design*, a panel data design (see SI Section S4) with n = 1231 observations; and **B** The *Common Design*, a multivariate mixed model (see SI Section S7) with n = 675. The positive estimated effect from the *Common Design* is not driven by having to drop sites that did not measure all the covariates (the sites in France, Portugal, and South Africa did not collect the soil data). If we use only the 675 observations from the multilevel modeling in our *Main Design*, we still obtain a negative estimated effect of richness on productivity, albeit less precisely estimated because of the smaller sample size (see SI Section S7).

Our study differs from prior ecology publications by combining three features: (1) causal diagrams to inform the design and transparently communicate the assumptions required for inferring a causal relationship from a correlation[3,39]; (2) regression models that leverage repeated observations on the same plots and sites to control for confounding variables (see Fig. 1B), both observable and unobservable[26,30,36,40]; and (3) rigorous assessments of the robustness of our inferences to violations of the assumptions required for inferring causal relationships from the data (these assumptions are described in more detail in Figs. 2 and 3, "Results", and "Methods"). The SI offers a primer on these ideas and compares them to approaches widely used in ecology (SI Sections S2, S4, S7 and S9).

In prior observational studies of how richness affects productivity, controlling for the wide range of potential confounding variables in multilevel or structural equation models has posed challenges (e.g.,[3,8,9,31,41]). In those studies, researchers who wanted to interpret an estimated effect as a causal effect had to assume that no confounding variables were left out of the models[30,36]. In complex ecological systems, however, it is unlikely that one can measure all possible confounding variables. Moreover, when these confounding variables are measured with error, statistically controlling for them can introduce other biases[36]. In other words, prior studies require a strong assumption for interpreting the correlation between richness and productivity as causal: any site or plot attributes not included in the statistical estimation model are assumed to be uncorrelated with species richness and therefore not a source of statistical bias. Our design relaxes this strong assumption.

We improve upon prior observational studies by controlling for a broader suite of confounding variables without needing to directly measure them as covariates (see "Methods" and refs. 36,40). To understand the intuition for how this control is possible, recall that, in contrast to most prior observational studies on this topic (e.g.,[3,8,9,31,36]), our multi-site data is longitudinal ('panel data') and thus includes variation in species richness in both time and space. Confounding variables that affect both richness and productivity could arise from conditions at the plot or the site. The values of these variables may be essentially invariant during the study period (e.g., soil texture,

topography, land-use history) or they may vary through time (e.g., surrounding land-use change, drought conditions that differ by both site and year). With our multi-site panel data, we can directly control for time-varying, site-level conditions, whether they are observable or not, via a regression estimator that includes a simple interaction of binary variables for each site and year (see "Methods", Eq. 2). Further, we can eliminate the confounding effects of time-invariant plot and site conditions by taking deviations from mean conditions, after which variables that do not change over time no longer have any explanatory role and thus are eliminated as a source of bias ("Methods"). Using alternative designs, we can also quantify the potential threat of additional sources of bias from unobserved, time-varying plot-level confounders and from reverse causality (by bias from reverse causality, we mean bias that could arise when a causal effect also runs from productivity to richness; see "Methods"). In contrast to our approach, virtually all observational analyses reviewed in[4] omit important confounding variables (e.g., from human activities and land management) and ignore the potential for reverse causality (reviewed in ref. 42).

To demonstrate how our study design builds on and advances prior research, we first apply two study designs that have been used in prior studies and then contrast them to our design. Specifically, we estimate a simple bivariate correlation of richness and productivity (like ref. 31) and then we estimate the relationship between richness and productivity using a multivariate design that mirrors advanced statistical designs that aim to control for confounding variables by directly measuring and including them as covariates in regression models (a "conditioning on observables" analysis, like ref. 6). The multivariate design, which we label *"Common Design in Ecology,"* controls for over 60 variables (far more than prior studies), including attributes of the soil, habitat, historical management, and weather (Table S10). More details are provided in "Methods" section.

## Main results

We first report the bivariate correlation between-plot richness and productivity. Consistent with prior studies[31], we find a statistically weak, positive relationship between richness and productivity when we do not control for any confounding variables: a 10% increase in

richness is associated with a 1.4% increase in productivity, 95% CI [−0.6, 3.4]. To give that estimated correlation a causal interpretation requires an implausible assumption that there are no confounding variables in the system (or that they perfectly cancel each other out).

Consistent with prior multivariate studies[3], the *Common Design in Ecology* yields a statistically significant, positive relationship: a 10% increase in plot richness increased productivity by 3.8% on average, 95% CI [0.01, 7.5] (Fig. 2B). To give that estimated effect a causal interpretation, however, requires a strong assumption: all possible confounding variables are measured accurately and are included in the model.

In contrast to prior analyses, our *Main Design* controls for a much broader set of potential confounders and comes to the opposite conclusion (Fig. 2A; Tables S2–S3; SI sections S4, S7 and S9). We find a 10% increase in plot richness decreased plot productivity by 2.4% on average, 95% CI [−4.1, −0.74]. The estimate is similar if we measure biodiversity using Simpson's Diversity (Table S2), control for concomitant changes in species evenness (Fig. 2A), or measure species richness and productivity as untransformed variables in linear or non-linear specifications (Table S3). In extended analyses (Tables S4–S6), we find no evidence that the effect of species richness on productivity is moderated by the site's productivity or total number of species (as in[38]).

Switching from the *Common Design* to our *Main Design* flips the estimated effect of richness on productivity from positive to negative (Fig. 2). This sign-switching likely occurs for two reasons. First, on average, many of the observed site and plot variables at these 43 sites are negatively associated with richness and positively associated with productivity (or vice-versa). Controlling for them in the *Common Design* moves the estimated effect in the positive direction from the bivariate correlation. Second, unobserved site and plot variables ($U_s$, $U_p$, $U_{st}$ in Fig. 1B) are, on average, positively associated with both richness and productivity. We can infer the sign of these associations by observing how the estimated effect changes with and without the controls for unobserved time-varying, site-level conditions and time-invariant plot and site conditions (Fig. 2A versus Fig. 2B; see "Methods"). Failing to control for the time-varying confounders is a particular problem in the *Common Design*. In other words, the Nutrient Network sites experience site-specific "shocks" that vary each year (e.g., weather shocks, like a particularly dry April, or herbivory shocks, like higher herbivore pressure than the prior year) and failing to control for them creates statistical biases in the positive direction. We cannot observe the exact components of these shocks, but because we observe the same sites over many years, we can control for them. The *Main Design*, with its greater set of controls, is thus less biased[36]. More details are available in SI Sections S7 and S9. Future research could elucidate what shocks are most relevant, thereby providing a way for researchers without longitudinal data to potentially control for the confounding effects of these shocks.

## Results are robust to alternative assumptions for inferring causality

A hallmark of modern approaches to causal inference is to probe the robustness of results to potential violations in the assumptions used to infer causality from correlation[35]. Using four additional approaches, we use assumptions different from those made in our *Main Design* (Fig. 2A) and assess how our conclusions change (see "Methods"). In all four approaches, the estimated effect of richness on productivity is negative (Fig. 3; Tables S7–S8).

Based on the first two approaches, we conclude that we are not mistaking the effect of productivity on richness[3,15] for the effect of richness on productivity ("reverse causality"). The first approach employs an instrumental variable design, which uses an observable source of variation in richness ($Z$ in Fig. 1B) that is assumed to have no connection to productivity after conditioning on the site and plot variables addressed in the *Main Design* (see "Methods"). When this assumption is valid, the design addresses both reverse causality and all forms of confounding in Fig. 1B, at the cost of drawing inferences from only a subset of the data, which can dramatically decrease the precision of the estimate. Our second approach assumes, based on[3], that a negative effect of productivity on richness would be mediated ($M$ in Fig. 1B), at least in part, by shading or factors for which shading is a proxy (e.g., overcrowding). To block the effect of this mechanism, we add a shading variable to the *Main Design* (see "Methods"). If the estimated effect changes, reverse causality may be a source of bias. In both approaches, the estimated effect remains negative, suggesting that, if either of the approaches' assumptions are valid, reverse causality is not driving our results. This conclusion does not mean productivity cannot affect richness, only that such a relationship is not a likely source of bias in our *Main Design*.

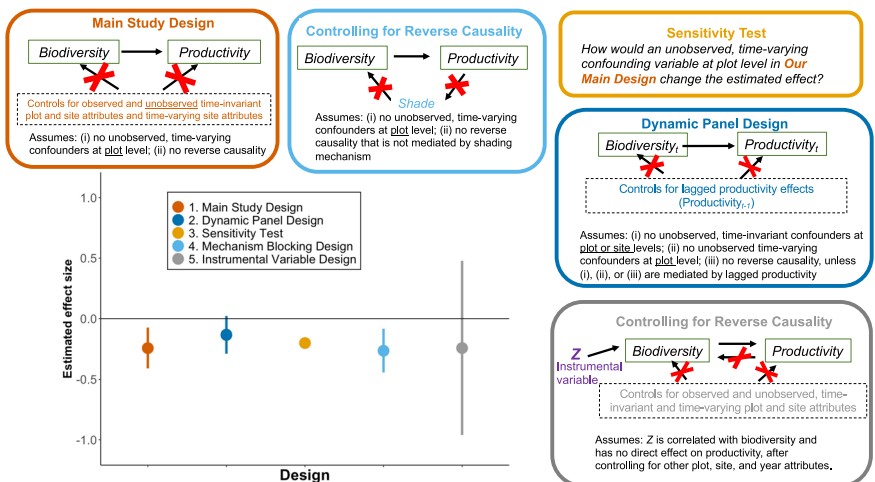

**Fig. 3 | The effect of species richness on productivity (robustness checks), estimated as the mean % change in productivity from a 1% increase in richness.** The bottom left panel shows the estimates of mean effect with 95% confidence intervals, except for the sensitivity test estimate, from (1) our main study design (*n* = 1231), (2) a dynamic panel design (*n* = 1063), (3) a sensitivity test (*n* = 1231), (4) a mechanism-blocking design (*n* = 1063), and (5) an instrumental variable design (*n* = 1212). The diagrams summarize the assumptions needed for drawing causal inferences in each design, where a red X on an arrow implies that the confounding pathway is blocked by the design. Using four approaches that make assumptions that differ from the assumptions in our *Main Design* (Fig. 2A), we find no evidence for a positive effect of species richness on productivity.

**A. Richness is a heterogeneous treatment: R|C ➔ P**

The effect on productivity (P) when richness (R) changes from one value
to another value depends on (|) the compositions (C) at the two values.

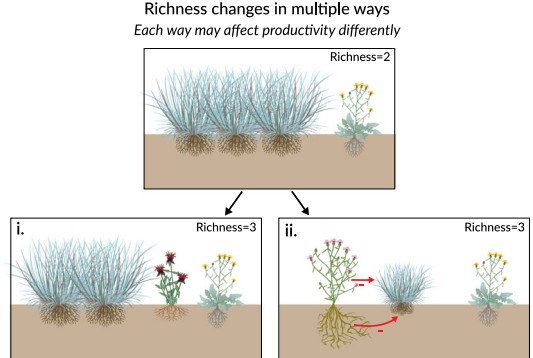

**B. Changes in richness in natural ecosystems**

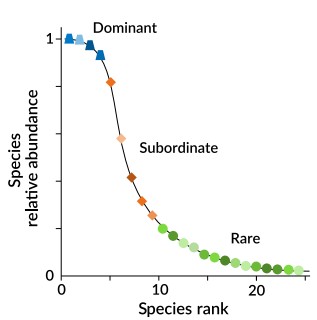

**Fig. 4 | Composition and its role in the effect of species richness on productivity.** This figure illustrates why results from observational and experimental studies may differ. Composition, the identities of species that could potentially grow at a site at a given level of richness, makes species richness a heterogeneous treatment in both experimental and non- experimental systems (**A**). The way richness changes in nature can thus influence how changes in richness affect productivity (**B**). In **B** (left), Richness has changed from 2 to 3 species in case (i), a case where rare species take space formerly occupied by more productive dominant species. In case (ii), where Richness also has changed from 2 to 3 species, rare species have negative effects on dominant species productivity (e.g., via belowground competition or allelopathy). **B** (right) A species rank abundance curve that illustrates that most species in high diversity grasslands are rare. Thus, as diversity changes from low values to higher values, the way in which rare species affect productivity will become more influential in affecting productivity levels. For simplicity, figure focuses on a contrast between rare and dominant species, but the ideas can also apply to differences between native and non- native species. Plant species images in Panel B are from Tracy Saxby, IAN Image Library (https://ian. umces.edu/imagelibrary).

Based on the final two approaches, we conclude that the estimated negative effect in the *Main Design* is robust to potential biases from unobserved confounding variables at the plot level that vary over time ($U_{pt}$ in Fig. 1B). First, in a dynamic panel design, we address bias that would arise if the prior year's productivity affects richness and productivity in the current year (e.g., via soil fertility[43]). The estimated effect is similar to the estimate from the *Main Design*. Second, in a sensitivity analysis[34], we assess how the *Main Design* estimate would change if there were a strong, unobserved confounding variable that is negatively associated with species richness and positively associated with productivity (e.g., measurement error or plot-level drivers of disturbance). If such a confounding variable were to exist, it could create a spurious negative correlation between richness and productivity. The analysis implies that, even in the presence of a such an unobserved confounder, we would still infer that there is a negative relationship between richness and productivity.

**The role of rare species and non-native species**
In contrast to our study, many experimental studies report positive effects of richness on productivity[1,2,7,13]. One difference between experimental and natural systems is that most species in natural ecosystems are rare, whereas most species planted in experiments are not rare (Figs. S11 and S12). Rare and dominant species can affect productivity differently[44]. Thus, the effect on productivity from an increase in richness (e.g., from 4 to 8 species) could differ when the additional species are rare versus not rare. In the jargon of the causal inference literature, richness is a compound treatment with multiple

versions, or a "heterogenous treatment"[45]. (Fig. 4A). Another difference between experimental and natural systems is the number of non-native species, which are absent in many experiments but increasingly prevalent in real ecosystems[25]. If non-native are more competitive but less productive, as in ref. 46, this could also explain the divergence between our results and those of experimental studies.

Rare and non-native species could reduce productivity through multiple channels (Fig. 4B, left). For example, these species may compete with more productive species (e.g., via allelopathy of rare invaders[47]). Further, they may produce less aboveground biomass than common, native species[44] and so when they enter a plot, they may take space formerly occupied by more productive species. These productivity-reducing effects would be strengthened if, as richness increases in a plot, rare and non-native species are more likely to be the incrementally added species (Fig. 4B, right). In experimental systems, species enter plots with equal probability. In contrast, as richness increases in our 43 grassland sites, the probability that the incremental species is a rare or non-native species also increases (Figs. S11 and S12).

Given differences in the species pools studied in our natural systems versus many experimental systems, we explore whether changes in rare species and non-native species richness affect productivity differently from changes in native, non-rare species richness. We classify species into four categories: (1) rare, native species; (2) non-rare, native species; (3) non-rare, non-native species; and (4) rare, non-native species (see "Methods"). We then estimate the effect of each category's richness on productivity using our *Main Design*.

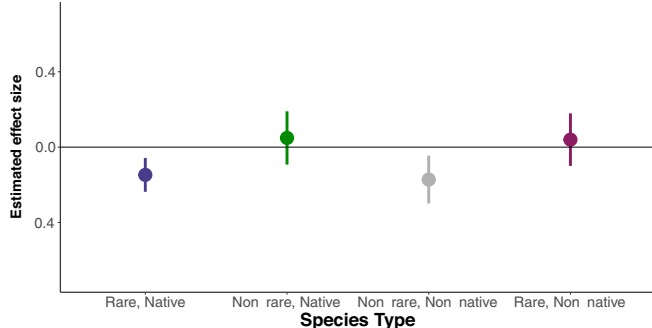

**Fig. 5 | Estimates of the mean effect of species richness on biomass production conditional on species type.** All estimated mean effects are on a log-inverse-hyperbolic-sine scale and shown with 95% confidence intervals and with $n = 1175$ within 42 sites. Given the inverse hyperbolic-sine transformation of the richness variable, the estimated effects cannot be interpreted as elasticities without further manipulation, but their signs and relative magnitudes can be compared. We can reject the null hypothesis that the estimated effects of these four types of species are equal (ChiSq = 9.82, $Pr$(ChiSq = 0.02)). Dropping observations from one site without data to define the species types does not change our estimates in Figs. 2 and 3.

Our results imply that the negative average effect of richness on productivity in Fig. 2 is driven by changes in the numbers of rare, native species and non-native, non-rare species (Fig. 5). Consistent with results from experimental studies, an increase in species richness that came from a non-rare, native species increased productivity. But increases in richness decreased productivity when these increases came from non-rare, non-native species or rare, native species (inferences are similar using different definitions for rarity; Tables S11–S15). We acknowledge that there could also be positive and negative interactions across these species' types, but we do not have sufficient statistical power to explore these potential interactions.

We conjecture that the proposed mechanisms through which richness positively affects productivity in archetypical experiments— i.e., niche complementarity and positive selection[2,14,15]—may operate primarily among non-rare, native species. Testing this conjecture would require experiments that successfully grow representative proportions of rare species (in experiments that planted rare species, these species failed to consistently emerge, see SI Section S9 – Box 2).

**Implications for experimental and observational biodiversity research**

Leveraging methodological advances for causal inference in observational designs, our study uncovers ecological relationships in grasslands that deserve closer attention. In the 43 grassland sites in our study, an increase in species richness decreases productivity on average (Fig. 2). This effect appears to arise because an increase in rare species and non-rare, non-native species decreases productivity on average (Fig. 5) and these species comprise most species in an ecosystem (Fig. 4). These effects will be missed in observational designs that do not adequately control for a wide range of confounding factors and in experiments that do not plant a representative mix of species in diversity patterns that occur in natural systems. Our results also highlight the challenge of determining the representativeness of experimental systems. For example, a recent comparison of natural and experimental systems identified many similarities in attributes but did not assess whether the patterns of rare or non-native species in experimental systems match the patterns in natural systems[48].

Our results point to promising areas for future research, including studies that experimentally manipulate rare species and non-native species and seek to identify traits of these species that drive their effects on productivity. For example, in a recent study[22], researchers

experimentally removed non-dominant species from randomly assembled communities and reported an increase in biomass one year after the removal; a result consistent with our results. Extensions of their study can help elucidate the traits of these species that drive their effects on productivity. Our results also imply that any estimated effect of changes in species richness on productivity may not generalize to different spatial or temporal scales, or to other ecosystems, forms of biodiversity, or ecosystem functions. Multiple ecological mechanisms underlie a relationship between richness and productivity (e.g., belowground competition, niche complementarity), and their strength may vary across places and time depending on which types of species are changing and how (Fig. 4). When the operative mechanisms depend on the version of richness that is changing, ecologists face what has been called "treatment-variation relevance"[45] or "consequential variation of the treatment"[49]. In such cases, interpreting and generalizing causal effects is challenging, whether ecologists use experimental or observational designs. The challenge is best met by using large observational datasets that capture the types of species changes that occur in nature and then, using methods like the ones in our study, determine which changes – in terms of which species are changing and how (Fig. 5) – are consequential for the effect of richness on ecosystem function. The results of these observational studies could then guide experimentalists in selecting experimental designs that can help confirm the results from observational studies and elucidate the underlying mechanistic processes.

Most importantly, our study extends prior research[3] that highlights the importance of study design in credibly isolating causal relationships in natural ecosystems. Other fields have made important advances in observational analyses − advances that have not yet permeated into ecology and other natural sciences. By demonstrating how to apply these advances to an ecological question and data, our study aims to spur broader adoption of these advances in ecology. Given the challenges of randomizing all the important elements of ecosystems at larger spatial and temporal scales, observational designs like ours that leverage these advances offer important complements to experimental designs in research to elucidate how natural ecosystems function.

## Methods

To ensure reproducible results, we implemented all analyses in two software programs (R using the 'fixest' package v 0.8.2 and Stata v.16) and multiple researchers confirmed the results. The code for reproducing all analyses, figures, and tables in this study are available through Zenodo (https://doi.org/10.5281/zenodo.7675340). A RMarkdown tutorial on the main methods can also be found on our Zenodo release (https://doi.org/10.5281/zenodo.7675340) and as Supplementary Data.

### Target causal effect

To formalize the causal relationship we seek to estimate, we use the potential outcomes framework[30,50,51]. The causal effect of a change in richness from $R'$ to $R''$ on productivity $P$ in plot $i$ is defined as $[P_i(R'') − P_i(R')]$, where $P_i(R'')$ is the potential productivity outcome when $R = R''$ and $P_i(R')$ is the potential productivity outcome when $R = R'$ $(R' \neq R'')$. The difference in these two potential productivity outcomes (i.e., productivity under two potential richness conditions) is the causal effect of a change in richness in a plot. For a specific location and time, only one of these potential outcomes will be directly observable; the counterfactual values for the other potential outcomes must be estimated from data. The average causal effect of a change in biodiversity from $R'$ to $R''$ across all plots is $E[P_i(R'') − P_i(R')]$, where $E[\cdot]$ is the expectation operator. We seek to estimate the average causal response of an incremental change in $R$ across all plots (i.e., the average effect across all possible one-unit changes). When used for causal inference, non-experimental studies aim to replicate, conceptually, the idealized experimental design in which the factor or factors that affect variation

in $R$ only affect $P$ via their effects on $R$. In other words, to permit credible causal inferences, a non-experimental design seeks, via design and statistical methods, to eliminate the confounding effects of $U_p$, $U_s$, $U_{st}$, and $U_{pt}$ in Fig. 1, as well as the effects of reverse causality in Fig. 3.

## Data description

Study sites include mesic grasslands and prairies, savanna, desert grasslands, montane meadows, old fields, and alpine tundra from 11 countries. To measure productivity, we use plant above- ground live mass (biomass) (see Fig. S1). Biomass production supports many ecosystem processes and services and this measure of productivity has been widely used in studying the relationship between diversity and productivity with both observational (e.g.,[3,9,31]) and experimental data (reviewed in ref. [4,52,53]). For herbaceous vegetation, aboveground live biomass provides a reasonable estimate of primary productivity[54]. Biodiversity measures are determined from species cover data from the Nutrient Network (SI Section 3a).

## Common design estimator

To show how our *Main Design* differs from more common designs in ecology, we constructed what we call a "*Common Design in Ecology*": a multivariate design that controls for over 60 variables (far more than prior studies), including attributes of the soil, habitat, historical management, and weather (Table S10). This design captures the strong assumption that is inherent in prior observational ecological studies that aim to estimate the causal effect of richness on productivity: there are no variables omitted from the statistical model that are correlated with both richness and productivity. When this assumption is not met, the design suffers from bias. We compare this *Common Design* to our *Main Design*, which relaxes that strong assumption.

For the *Common Design in Ecology*, we estimate the effect of richness on productivity using the following regression equation:

$$\ln LiveMass_{pst} = \beta \ln Richness_{pst} + X_p + X_s + X_{st} + \varepsilon_{pst}. \qquad (1)$$

We use a ln-ln model specification (for rationale, see SI Section S3b). Recall that observations in our data come from a plot $p$ located within a site $s$ in a year $t$. Thus, $X_p$ is a vector of plot-specific attributes that do not vary over the study period (e.g., soil type), $X_s$ is a vector of site attributes that do not vary over the study period (e.g., habitat, historical management, elevation), and $X_{st}$ is a vector of site attributes that vary by year (e.g., temperature seasonality, maximum and mean temperatures of the warmest month). Together, these vectors include over 60 variables, which are directly controlled for in the regression (see Table S10). In this equation, we can see that the effects of any omitted variables on productivity (i.e., variables not controlled for in the $X$ vectors in the model) reside in the error term $\varepsilon_{pst}$. We can rewrite this error term as a combination of a random error, $\sigma_{pst}$, which only affects productivity, and unobserved confounding variables, $U$, which affect both richness and productivity at either the plot or site level and either in all years or only some years. Thus the error term can be rewritten as $\varepsilon_{pst} = U_p + U_s + U_{st} + U_{pt} + \sigma_{pst}$, where $U_p$ and $U_s$ are vectors of plot and site-level variables that do not change over the study period, and $U_{st}$ are vectors of time-varying site-level variables, and $U_{pt}$ are vectors of time-varying plot-level variables. If a study design has any of these omitted $U$ variables, the estimator ($\beta$) would be biased—known as omitted variables bias. In other words, to interpret the estimate of $\beta$ as an estimate of our target causal effect would require one to assume that the observed covariates in the model capture all relevant $U_p$, $U_s$, and $U_{st}$ and $U_{pt}$ does not exist (because no time-varying covariates are measured at the plot level).

## Main design estimator

Each observation in our study comes from a plot $p$ located within a site $s$ in a year $t$. With this longitudinal data structure, i.e. with repeated

observations of the same plots, one can control for all dimensions of confounding variables that do not vary over the study period and all dimensions of time-varying site-level confounding variables without having to observe all of these dimensions. To achieve this control, we estimate an equation of the following form:

$$\ln LiveMass_{pst} = \beta \ln Richness_{pst} + \delta_p + \mu_{st} + \varepsilon_{pst} \qquad (2)$$

As noted above, we use a ln-ln model specification (for rationale, see SI Section S3b). Given that we have a ln-ln specification, $\beta$ can be interpreted as an elasticity: the expected percent change in productivity given a one percent change in richness. We also tested the robustness of results to this modeling decision (see SI Section S5).

The *time-invariant plot* attributes ($\delta_p$) are modeled in a fully flexible way that allows each plot to have its own effect on productivity (details on estimation procedure below). In the Economics literature, $\delta_p$ would be called "plot-level fixed effects." Note that the phrase "fixed effects" has a different meaning in economics than in ecology (see S1 Glossary). In economics, including $\delta_p$ is said to control for "unobserved heterogeneity" across plots that can be a potential source of bias. Note that $\delta_p$ is not part of the error term, as it would be in mixed (multilevel) models[55] or in a Common Design as in Eq. 1 (i.e., $U_p$ and $U_s$). Rather, it is a parameter to be estimated, just like $\beta$. In other words, $\beta$ and $\delta_p$ are assumed to be fixed and estimable, rather than assumed to follow a distribution. Time-invariant *site* attributes are not explicitly included in the equation because they are subsumed into the time-invariant plot attributes (i.e., plots are nested within sites and so fixed site attributes are controlled via fixed plot attributes). In other words, this variable captures all attributes of a plot at a given site that vary little over the study period. Thus, $\delta_p$ captures both $U_p$ and $U_s$ in the decomposition of the error term from the previous section (i.e., $\varepsilon_{pst} = U_p + U_s + U_{st} + U_{pt} + \sigma_{pst}$, where $\sigma_{pst}$ is a random error that only affects productivity and $U$ are unobserved confounding variables that affect both richness and productivity).

To show how the estimator in Equation 2 can efficiently control for time-invariant confounders, we subtract the productivity observation within a plot in one year ($t - 1$) from the productivity observation within the same plot in the next year ($t$), yielding an equation for the change ($\Delta$) in productivity from one year to the next:

$$\Delta \ln LiveMass_{pst} = \beta \Delta \ln Richness_{pst} + \Delta \mu_{st} + \Delta \varepsilon_{pst} \qquad (3)$$

The variable $\delta_p$, which captures the effects of time-invariant plot attributes, has been differenced away, allowing for efficient estimation of $\beta$; in other words, we control for time-invariant plot attributes without having to estimate them and use up many degrees of freedom. In this differenced version of Equation 2, one can see that we are estimating the effect of richness on biomass from changes in richness within plots, where the confounding effects of between-plot differences are absent. Rather than first-differencing the equations to eliminate $\delta_p$, one can instead take deviations from plot-level means, which is the approach we take to estimating Equation 2 because it can be more efficient. Thus, we can also describe our estimation strategy as estimating a correlation between deviations of productivity around its mean and the corresponding deviations in richness around its mean.

The *time-varying* site attributes ($\mu_{st}$) are also modeled in a fully flexible way that allows a year- specific effect for each site (in the estimation, an indicator for each year is interacted with an indicator for each site). Explicitly estimating $\mu_{st}$ flexibly controls for confounding variation due to conditions at a site that change through time, such as weather (e.g., time-varying patterns of temperature and precipitation), herbivory, and surrounding land management conditions. In other words, this variable captures all year-specific conditions experienced by every plot at a given site. Thus $\mu_{st}$ captures $U_{st}$ in the decomposition of the error term from the previous section ($\varepsilon_{pst} = U_p + U_s + U_{st} + U_{pt} + \sigma_{pst}$). The estimator in Eq. (2) is often called a "two-way fixed effects

estimator" because, by taking deviations from the means, one controls for time- invariant confounding and, by including the site-by-year effects, one controls for time-varying confounding.

Thus, in contrast to the *Common Design in Ecology*, we control for a broad suite of plot-level and site-level confounders without having to measure them directly. The *Main Design* also controls for non-linear relationships between the confounding variables and productivity or richness, as well as linear and non-linear interactions among those variables. Not having to measure the confounding variables also yields another benefit: if the observable confounders were measured with error and that error were correlated with the measure of richness, the *Common Design* would have another source of bias. Moreover, because the model specification comprises only the richness variable and a set of binary indicator variables and their interactions, the risk of mis-specification bias from how confounders are modeled is lower. To better understand how our design differs from more common designs in Ecology such as mixed-effect modeling approaches and convergent cross-mapping approaches[56], see SI Sections S4c and S7.

The term $\varepsilon_{pst}$ in Eq. (2) is a time-varying random error term at the plot level, assumed to have mean zero and no correlation with ln *Richness*, i.e., it corresponds to $I_{pst}$ in Fig. 1B. These plot-level errors may be serially correlated (i.e., temporally dependent even after conditioning on richness and site-by-year effects), and thus we cluster the standard errors at the plot level[57]. Our clustered estimation of the variance allows for arbitrary serial correlation within each plot, as well as heteroskedasticity across plots[36,58]. Errors at a given site may also be correlated (even after conditioning on site-by-year effects) and thus, as a robustness check, we also estimate standard errors clustered at the site level (Table S2).

Our *Main Design* has weaker assumptions than the *Common Design in Ecology*, but both have one assumption in common: there are no unobserved time-varying plot-level confounders in the error term (no $U_{pt}$). In other words, we assume that, after controlling for time-invariant plot and site attributes that are correlated with richness and productivity, and time-varying site attributes that are correlated with richness and productivity, the remaining temporal variation in richness in a plot is "as if randomly assigned," independently across time. This assumption is equivalent to assuming that the remaining variation in richness is driven by variables that have no link to productivity other than through their effect on richness (i.e., $Z_{pst}$ in Fig. 1) and thus there is no correlation between Δln *Richness*$_{pst}$ and $\Delta\varepsilon_{pst}$ in Eq. (3). If our assumption is correct, we can give a causal interpretation to the estimate of $\beta$. Unlike prior ecology studies, however, we assess the sensitivity of our results to violations of this assumption (see *Robustness Checks* next).

### Robustness checks: modifying the main design
**Robustness checks: model specifications.** In the SI (Table S3), we present the results from variations in the specification of Eq. (2): (1) we include a control for species evenness; (2) we change the measure of diversity from species richness to Simpson's Diversity index; (3) we include the lagged effect of species richness in the prior year (ln *SpeciesRichness*$_{t-1}$); and (4) we vary the functional form by (i) taking the natural logarithm of productivity but using the untransformed richness values, (ii) using both untransformed richness and untransformed productivity values, and (iii) using untransformed richness and untransformed productivity values in a non-linear, quadratic specification (i.e., we include ln *SpeciesRichness*$_t$ and ln *SpeciesRichness*$^2$$t$).

**Robustness checks: causal assumptions.** As noted above, the key, untestable assumption for drawing a causal inference from the estimator in our Main Design is the following: after controlling for time-invariant plot confounders and time-varying site confounders, the remaining factors that drive changes in richness only affect productivity via their effects on richness. We consider potential violations of this assumption and the implications for our inferences -- i.e.,

whether our conclusions could change -- by conducting a series of analyses that rely on alternative assumptions for causal inference (Fig. 3).

**Instrumental variable design for unobservable confounders and reverse causality.** First, we explore the potential violation of our assumption that the effect we are estimating goes from richness to productivity, and not the other way around. Richness and biomass measures in our data are taken simultaneously each year, as they are in most ecological datasets. Thus, we cannot rely on temporal sequencing of the data to rule out reverse causality.

To assess the potential threat of reverse causality, we adopt a statistical approach that is common in economics and public health, but rare in ecology: an instrumental variable design[59–63]. When its underlying causal assumptions are valid, this design allows researchers to eliminate not only the influence of reverse causality but also the influence of unobservable confounders, both static and dynamic.

To implement this design, we need to measure an attribute of the system that has a relationship with richness, but, after conditioning on the other plot and site attributes in Equation 2, has no relationship with productivity other than through its relationship with richness. Such an attribute is conceptually illustrated by the variable $Z$ in Figs. 1, S4 and S5. In economics and biostatistics, $Z$ is called an instrumental variable (IV) or a surrogate variable. An example of a potential IV is randomization of planted richness by an experimenter. In field experiments, randomization of richness helps isolate the causal effect of richness on productivity, but only when the randomization affects productivity in a plot solely through its effect on richness, an assumption called excludability or the exclusion restriction[64]. In other words, one must assume there is no arrow going from $Z$ directly to $P$ in Fig. 1.

In the absence of randomization, one must use theory and experience to identify a naturally occurring IV (reviewed in ref. 62). Each of the plots in our sample are unmanipulated plots that are embedded in blocks of manipulated plots in the Nutrient Network. In other words, each unmanipulated plot in our sample is surrounded by a set of plots with experimental nutrient additions (see ref. 65). These manipulated experimental plots received randomized amounts of nutrient additions, which subsequently affected the experimental plots' richness[66]. We assume that the experimentally manipulated richness in these plots can also affect the richness in unmanipulated plots in the same block through ecological dispersal channels but does not affect the productivity of these unmanipulated plots except through the effect on the plots' richness (an assumption made more plausible by the randomization of nutrients in the neighboring plots). If that assumption is correct (called an "excludability assumption"), we can use the average richness of an unmanipulated plot's neighboring manipulated plots in the same block as an IV for richness in the unmanipulated plot. The SI Section 6bii provides justification and further discussion of this IV.

The cost of using the IV design is that we can only estimate the average effect on productivity for the subset of the changes in richness that are affected by the IV. This subset is comprised of what are called "compliers" – plot-year observations for which the richness value would have been different had the average richness in surrounding plots been different. Thus, the IV design has much lower statistical power than our Main Design[62,67].

To implement the IV design, we use a two-stage, least squares estimator[26]:

$$\text{First Stage}: \ln Richness_{pst} = \gamma IV + \delta_p + \mu_{st} + P_{pst} \qquad (4)$$

$$\text{Second Stage}: \ln LiveMass_{pst} = \beta \ln \widehat{Richness}_{pst} + \delta_p + \mu_{st} + \varepsilon_{pst} \qquad (5)$$

In the first stage (Eq. 4), we predict richness, and, in the second stage, we use the predicted values of richness to estimate the effect of richness on productivity (see Table S8 for the results from both stages). We can reject the null hypothesis of a weak instrument using the Montiel-Pflueger effective F-statistic, which is a test that is robust to heteroscedasticity, serial correlation, and clustering[68];. For further discussions of the IV design and its assumptions, see SI Section 6b.ii.

**Blocking a mechanism for reverse causality.** As an alternative approach to address the potential threat of reverse causality in our design, we posit a mechanism through which productivity affects richness: shading (based on[3,69]). Although productivity could affect species richness through non-light pathways, such as soil resource use, the effect of productivity on richness is expected to be, at least in part, mediated by reductions in light from increased biomass that, in turn, reduces richness in a plot[69]. As an estimate of shading, we measure the fraction of photosynthetically active radiation (e.g., light used by plants) that reaches the soil. See SI Section S7b for details.

If the estimated negative relationship between richness and productivity in Fig. 2 were an artifact of reverse causality, then putting our shading variable in Equation 2 as a covariate would block the effect of productivity on richness that arises via shading. The sign of the coefficient on richness ($\beta$) would then become positive (or small and statistically insignificant if the true relationship between richness and productivity were zero). Yet the estimated effect remains unchanged (Fig. 3). If shading were not an important mechanism through which productivity would affect richness in our sample, or if our measure of shading is a poor measure of the shading mechanism, our mechanism-blocking design would fail to quantify the potential threat of reverse causality. For this reason, we also implement an instrumental variables design, described above, that makes different assumptions to account for reverse causality.

**Bracketing the "True Causal Effect": accounting for potential bias from dynamics.** The IV design not only addresses reverse causality but it also addresses all forms of dynamic, plot-level confounding variables (e.g., past productivity). However, it relies on untestable assumptions that may not be satisfied (e.g., excludability assumption). To supplement that analysis, we also explored a range of potential sources of bias from dynamic confounders. Here, we report on the methods used for one of these analyses, with results shown in Fig. 3. The other analyses and methods are reported in the SI Section S6.

In this analysis, we consider the possibility that prior productivity affects both current richness and productivity. We re-estimate the effect of richness on productivity using a lagged-dependent variable (LDV) design[70], which relies on different causal assumptions for identifying a causal effect. The Main Design assumes that the relevant confounders are time-invariant over the study period, or they vary over time at the site level rather than the plot level (e.g., site and plot-level differences in evolutionary history, age in community assembly, grazing intensity at a site, and history of disturbances and recovery stage in each plot). Instead, the LDV design considers: "What if current species richness and productivity were determined by last year's productivity, in addition to, or instead of, site-level conditions varying through time (e.g., precipitation)?" The LDV design, in contrast to the Main Design, assumes that the relevant confounders vary over time at the site level and, at the plot level, their static and dynamic effects can be controlled by controlling for past productivity (in other words, the effects of confounders are mediated directly and indirectly through prior productivity at the plot level). To achieve this control, we estimate an equation of the following form:

$$\ln LiveMass_{pst} = \beta \ln Richness_{pst} + \theta \ln LiveMass_{ps(t-1)} + \mu_{st} + \varepsilon_{pst} \quad (6)$$

Under certain conditions, the estimated effects of richness in our Main Design and in the LDV design "bracket" the true causal effect[26,71]. If the assumptions of the Main Design are valid, but the LDV design are invalid, the estimate from the LDV design provides an upper bound estimate. If the assumptions of the LDV are valid, but the Main Design are invalid, the estimate from the Main Design provides a lower bound estimate. As observed in Fig. 3B, this bracketing exercise implies the true effect is negative.

**Sensitivity test: would unobserved confounding variables change our conclusions?.** To further explore the potential effect of violations in our assumption that there are no time- varying plot attributes that are systematically correlated with richness and productivity, we explore how our estimated effect would change if there were an unobserved confounder that was negatively correlated with richness and positively correlated with productivity (i.e., a source of bias that would yield a spurious negative causal relationship between richness and productivity in our design). Said another way, this analysis answers the question, "How much correlation between the unobserved variable and the richness and productivity variables would be sufficient to change our conclusions?"

We applied a sensitivity test following the method introduced by Altonji et al.[72] and further developed by Oster[34]. More details on the method are in SI Section S7a and Table S7. We set $\pi = -0.10$ and Rmax = 1, which would mimic a powerful potential unobserved confounder in our design: a confounder that is so strongly correlated with productivity and richness that, were we able to observe it (along with the other variables in the equation), we could predict with near certainty which of two plots would have higher productivity and which would have higher richness. Estimating the effect of richness on productivity with those implausible parameter values yields an upper bound on the impact of this confounder on the estimated effect of richness on productivity in our *Main Design*.

The estimated upper bound is still negative: a 10% increase in richness implies a 2% decrease in productivity, on average. In other words, in the presence of an unobserved confounder that is negatively associated with richness and positively associated with productivity relationship (thus creating some spurious negative correlation between richness and productivity), we would still infer that there is a negative relationship between richness and productivity. To infer a positive relationship between the two variables would require an infeasible value for $\pi$: it requires $\pi > 1$, which implies the confounder would have to be more influential in explaining variation of productivity than the plot-level, time-invariant attributes and the site-level, time-varying attributes that are in Equation 2.

For completeness, we also considered an unobservable confounder that was positively associated with both richness and productivity and thus could be masking some of the negative effect of richness on productivity (i.e., positive selection bias). In other words, we also calculate a lower bound on the estimated effect by setting $\pi = 0.10$ (see SI Section S6b.ii).

**Testing hypotheses about moderators of the causal effect**

In the SI (Tables S4–S6), we present results from hypothesis tests about moderators of the plot-level richness effect on productivity. The potential moderators are: (1) the average level of productivity at a site (i.e., *does the effect of richness on productivity differ between high versus low productivity sites, as reported by*[38]*?*); and (2) the average level of richness at a site (i.e., does the effect of richness on productivity differ between high versus low richness sites, as hypothesized by[73]?).

To conduct these tests, we expanded Equation 2 by adding an interaction term between ln $Richness_{pst}$ and the moderator variables. We measured site-level productivity in four ways, which vary by the discreteness of the measure and by the way time is incorporated into the measure. In terms of discreteness of the measure, we measured

productivity both as a continuous variable and, using classifications from Wang et al.[38], as categorical variables for high, medium, and low productivity. Wang et al.[38] used cross-sectional analyses to study this moderator. Because we have longitudinal data, we can measure the continuous and categorical measures of site-level productivity in two ways: average productivity per site over the entire time series and site-level productivity per year. More details on the motivations for selecting these moderating variables for analysis can be found in the SI (Section S5).

### Exploring effect of species richness on biomass conditional on species type

We assign the labels "rare" and "non-rare" based on relative abundance at each site, and species' origin was origin was determined by the site coordinators in the Nutrient Network (SI: Section S8). We define relative abundance based on relative aboveground cover. We use relative cover as our metric for abundance because we believe it better captures the range of mechanisms through which rare species may decrease productivity, including taking space formerly occupied by more productive species. However, we test the sensitivity of our results to this decision by also using a relative frequency metric to define rarity (see SI Section S8c.ii). In Section S8c.iii, we also test the sensitivity of our conclusions to different cutoff values for assigning a species to the "rare" and "non-rare" categories. We modify and use the *Main Design* for each way of defining the four species groups. Finally, using Chi Squared tests, we tested the null hypothesis that the species richness of the groups had the same effects on live biomass.

### Reporting summary

Further information on research design is available in the Nature Portfolio Reporting Summary linked to this article.

## Data availability

The processed data used in this study have been deposited on Zenodo under https://doi.org/10.5281/zenodo.7675340. The raw data for unmanipulated plots that were not included in the analyses, because they did not meet the inclusion criteria, are available under restricted access for which permission can be obtained by contacting the Nutrient Network at https://nutnet.org.

## Code availability

All analysis code and output are available through our GitHub project site https://github.com/LauraDee/NutNetCausalinf and are released on Zenodo (https://doi.org/10.5281/zenodo.7675340). All code for data processing, including of the raw data, main analyses, and supplemental analyses is available.

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

## Acknowledgements

We thank E. Seabloom, A. Asmus, H. Correa, J. Firn, and F. Isbell for their input. The views herein are solely those of the authors and do not necessarily reflect those of the Federal Reserve Bank of Philadelphia or the Federal Reserve System. We thank UMN Supercomputer Institute for hosting data and IoE for hosting Network meetings. We acknowledge support from NASA BioSCape to L.E.D. and P.A. We acknowledge support from SF Long- 354 Term Ecological Research (LTER) Network Communications Office and US NSF 823 DEB- 355 1545288. This study used data from the Nutrient Network (http://www.nutnet.org), funded at the site-scale by individual researchers. Coordination and data management was supported by funding to E. Borer and E. Seabloom from NSF-DEB-1042132 and Institute on the Environment (DG-358 0001-13) and to E.B., and P.B.R from LTER (DEB-1234162, DEB-1831944). We also acknowledge support from DEB-

0620652, NSF grants DEB-1242531, DEB-1753859, and DBI-202189 to P.B.R.; TULIP Laboratory of Excellence (ANR-10-LABX-41) to M.L.; and CEREEP-Ecotron Ile De France (CNRS/ENS UMS 3194), Regional Council of Ile- de-France (DIM Program R2DS I-05-098/R), GoF/ANR's Investissements d'Avenir program (ANR-11-INBS-0001 AnaEE France; ANR-10-IDEX- 0001-02 PSL) to X.R. This is Kellogg Biological Station Contribution no. 2338. Soil analyses were supported, in part, by Oregon State University, University of Minnesota, and USDA-ARS grant 58-3098-7-007 to E.T.B.

## Author contributions

L.E.D. led the paper. L.E.D and C.N.S. initiated the study concept, developed the study questions, and conceived the outline of study. L.E.D., C.N.S., and P.J.F. designed the study and conducted the data analyses. L.E.D. and P.J.F. wrote the paper and Supplementary Information, performed the revisions, and designed and made the figures. L.E.D. and K.A.K. processed and cleaned the data with input from the Nutrient Network and code checking provided by the coauthors listed below. C.N.S., E.T.B., M.L., P.B.R., J.C., K.A.K., A.T.C., J.E.K.B., K.B., A.J.W., A.S.M, and A.H. contributed to the conception and framing of the paper, the interpretation, and writing. C.N.S., A.T.C., K.K., J.M, R.L.C., and S.A.P. contributed to the Supplementary Information. C.N.S. and A.T.C. contributed to revisions/response to reviewers. E.T.B., K.D., P.A., A.M., YH, J.M.K, T.O., M.D.S., L.S., C.S., K.K., J.E.K.B., A.H., X. R., S.A.P., J.W.M., C.A.A., and J.L.M. contributed data and edited and commented on the manuscript. E.B. planned and coordinated experimental network. A.T.C., K.A.K., C.A.A., T.O., X.R., C.N.S., and L.E.D. checked R and/or STATA code for reproducibility. L.E.D. and C.N.S. created the supplementary RMarkdown tutorial and GitHub repository.

## Competing interests

The authors declare no competing interests.

## Additional information

[1]Department of Ecology and Evolutionary Biology, University of Colorado, Boulder, CO, USA. [2]Department of Environmental Health and Engineering, Bloomberg School of Public Health & Whiting School of Engineering, Johns Hopkins University, Baltimore, MD, USA. [3]Carey Business School, Johns Hopkins University, Baltimore, MD, USA. [4]Research Department, Federal Reserve Bank of Philadelphia, Philadelphia, PA, USA. [5]Department of Ecology, Evolution, and Behavior, University of Minnesota, St. Paul, MN 55108, USA. [6]Department of Biology, University of Massachusetts Boston, 100 Morissey Blvd, Boston, MA 02125, USA. [7]Institute of Biology, University of Graz, Holteigasse 6, 8010 Graz, Austria. [8]Ecology and Biodiversity Group, Department of Biology, Utrecht University, Padualaan 8, 3584 CH Utrecht, The Netherlands. [9]Department of Plant Sciences, University of Oxford, Oxford OX1 3RB, UK. [10]Sorbonne Université, Université Paris Cité, UPEC, IRD, CNRS, INRA, Institute of Ecology and Environmental Sciences, iEES Paris, Paris, France. [11]Institute for Global Change Biology, and School for Environment and Sustainability, University of Michigan, Ann Arbor, MI, USA. [12]Department of Forest Resources, University of Minnesota, St. Paul, MN 55108, USA. [13]Hawkesbury Institute for the Environment, Western Sydney University, Penrith, NSW 2751, Australia. [14]Department of Biological Sciences, California State University Los Angeles, Los Angeles, CA, USA. [15]Department of Physical and Environmental Sciences, University of Toronto at Scarborough, Toronto1265 Military TrailON M1C 1A4, Canada. [16]Department of Integrative Biology, University of Guelph, Guelph, Ontario N1G 2W1, Canada. [17]Research Center for Advanced Science and Technology, The University of Tokyo, 4-6-1 Komaba, Meguro, Tokyo 153-8904, Japan. [18]Department of Biology, Colorado State University, Fort Collins, CO 80523, USA. [19]Graduate Degree Program in Ecology, Colorado State University, Fort Collins, CO 80523, USA. [20]Department of Wildland Resources and the Ecology Center, Utah State University, Logan, UT 84322, USA. [21]School of Environmental and Forest Sciences, University of Washington, Box 354115 Seattle, WA 98195-4115, USA. [22]Global Water Security Center, The University of Alabama, Box 870206 Tuscaloosa, AL 35487, US. [23]Smithsonian Environmental Research Center, Edgewater, MD 21037, USA. [24]Department of Health and Environmental Sciences, Xián Jiaotong-Liverpool University, Suzhou, China. [25]Department of Plant and Soil Sciences, University of Kentucky, Lexington, KY 40546-0312, USA. [26]School of Biological Sciences, Monash University, Clayton, VIC 3800, Australia. [27]Department of Ecology, Environment and Evolution, La Trobe University, Bundoora, VIC 3086, Australia. [28]Department of Plant Biology, Michigan State University, East Lansing, MI 48824, USA. [29]Kellogg Biological Station, Michigan State University, Hickory Corners, MI 49060, USA. [30]Lancaster Environment Centre, Lancaster University, Lancaster LA1 4YQ, UK. [31]Theoretical and Experimental Ecology Station, CNRS, 09200 Moulis, France. ✉e-mail: laura.dee@colorado.edu; pferrar5@jhu.edu

