## [Peer Review File · Nature Communications]

REVIEWER COMMENTS

Reviewer #1 (Remarks to the Author):

This work, borrowing the approach of causality analysis developed from econometrics, investigates biodiversity effects on productivity, using longitudinal observational data. I agree author' arguments in introduction: 1) experimental design for BEF research is unrealistic and cannot be generalized to natural system; thus, longitudinal (time series) data from natural systems are needed. I also agree 2) confounding variables create problem in studying Biodiversity <-> Ecosystem Functioning; however, I am not convinced that the linear decomposition approach used in this work has solved the issue. Further, I like to point out 3) another critical issue is that BEF interaction is embedded in complex feedback systems (Tilman et al 2014 Annu. Rev. Ecol. Evol. Syst.). For example, biomass->nutrient->diversity->biomass (a cyclic path in network terminology). Other more complex feedbacks exist in natural systems. This critical issue was not correctly resolved by their approach.

This study made little progress beyond Grace et al 2016 (Nature). In fact, Grace et al. found positive BEF effects based on a dataset to some degree overlapped with the data used in this study. To compare with Grace et al (2016), the authors made extensive paragraphs (in the SI) to say that the difference in finding is due to the difference in approach. Well, how to reconcile the opposite findings then? Grace et al. (2016) used a global-scale meta-analysis and applied SEM to rule out the confounding effects explicitly, whereas the present study accounts for the potential confounding factors using site and/or plot effects. From statistical point of view, I am not convinced that including site and/or plot as effect variables solves the issue of confounding variables and the feedback problems. Moreover, the framework of "linear decomposition" of productivity variation into diversity effects + the plot and site effects does not fit in the real-world situation; that is, BEF behaves as context-dependent complex feedback systems. I have several critical concerns on their approach.

1. I have mathematical concerns on their definition of "causal effect". The authors define: "the causal effect of R on P is the difference in productivity between two states of the world in which all other factors are held constant except R ". This statement is not mathematically correct. The key problem is that other factors are NOT held constant and cannot be assumed held constant. Rather, a better mathematical definition of context-dependent causal effect is partial derivative. That is, $\frac{\partial(p)}{\partial(R)}$ conditional on the instantaneous state of other confounding variables.

2. Real-world systems are nonlinear dynamical systems. Time-invariant attribute does NOT exist in nonlinear dynamical systems. The linear model employed in this study assumes that the biodiversity effects and other confounding variables (whether they are observable or not) have additive effects on biomass (function). Unfortunately, in real worlds, biodiversity and other variables are nonlinear interactive (context-dependent) and are NOT linear decomposable (those effects are NOT additive). Essentially, there is nothing called "invariant diversity effect" in natural systems. In reality, diversity effect always depends on other variables (e.g., T, nutrient, precipitation) in a nonlinear way (the dependency cannot be linearly partial out as executed in this study); that is, diversity effect is time-varying in natural systems. The effects from confounding variables (e.g., T, nutrient, precipitation) are also not invariant and cannot be linearly partial out. Assuming the confounding variables as instrumental variables hidden by sites or plots effects does not resolve the problem. In fact, BEF has been studied from nonlinear dynamical system viewpoint, e.g., Clark, et al. (2015) Ecology.

3. Natural systems (in particular the BEF issue) cannot be described as DAG. Obviously, biomass may influence nutrients (or other variables in soils) which then affect diversity (or species composition), which then affects biomass. A feedback cycle in the other direction (diversity->nutrient->biomass->diversity) also exists. In nature, BEF is a feedback control system and thus cannot be modelled as DAG.

4. The authors investigated the potential reverse causality. However, they did not investigate more complex feedback through environmental variables (such as nutrients). Also in reality, the effect of P on R occurs together with the effect of R on P and confounding factors on P. Furthermore, as admitted by the authors, the effect of P on R is not invariant (e.g., can be linear positive or negative or hump-shape, or no effect). Thus, their statistical models are unrealistic. Again, my fundamental concern is that the linear model(s) developed by the authors cannot solve the BEF causality problem in real-world observational data.

5. The dynamic panel design did not consider feedback (reverse or through the third variable such as nutrient). Mathematically speaking, the linear models employed by the authors do not possess the capacity to accommodate the time-varying dynamical feedbacks.

6. The findings presented in this study are not well organized, and explanations on their analysis are unclear throughout the main text. Figures are full of texts, but the ideas of their analysis remain poorly explained (Fig. 2, 3). In particular, the main differences in five analytical scenarios in Figure 3 are not clearly presented. It is difficult to tell how these scenarios are relevant to the 'Common Design' or 'Main design' in figure 2. Fig. 4 is just like making up by three separate ppt slides. It also includes too many statistical jargons from DAG and is unfriendly to the general audience. Therefore, it is difficult to follow the main text that is not self-sustained without checking SI in detail. Even worse, the SI itself is not well-organized. It is hard to clarify the main text and find the important supplementary materials, e.g., various confounding variables, assumptions, and analytical designs.

Minor comments:

P3, L43-44: Jochum et al. 2020 Nature Ecol. Evol. also explored the realistic sequence of biodiversity loss.

P4, L67: I don't think the methods applied in this study are the basis of any one of the Nobel Prize in 2021. Which one of the Nobel Prize the author meant?

P6, L89-90: What does 'external validity' mean?

P6, L96-97: Not clear about the importance of repeat observations on revealing various confounding variables. What are those confounding variables exactly?

P6, L98-100: What are those assumptions exactly? In addition, the citation for S2, S4, S7, S9; >20 SI pages) is not precise enough to guide the audience.

P6, L105-106: Again, the author mentioned 'repeat observations' but still lacks a clear explanation.

P7, L120-121: The quantification of 10% increase cannot be inferred from bivariate correlation analysis.

P8, L136: What is the set of confounding variables concerned in this study that makes BEF negative? Can they be pointed out specifically?

Figure 2: In panel B, what is the difference between (i) & (ii). They look confusing. Panel A allows time-varying confounder at site level, but I don't understand how this design is critical to their analysis. Overall, this conceptual diagram poorly explains the motivations for these designs, and I cannot find the main ideas in the main text.

P10, L159-160: I don't agree such an argument that the more controls we have the more unbiased results we will obtain. It is correct only if those controls can be correctly considered. However, I don't see robustness analysis to validate the way of including additional confounders in their statistical models.

P10, L165: There are five designs presented in Fig. 3 rather than four.

P11, L180: Lack of clear explanation about 'conditioning'. This is a statistical jargon.

Figure 3: This figure fails to convey a clear idea among different designs. The effect size panel with inconsistent names to the titles of other panels. Overall, there are too many words and statistical jargons without clear definitions in this figure.

P17, L256: I don't see this information from realistic sites presented in figure 4.

Reviewer #2 (Remarks to the Author):

In this study, Dee and co-authors investigate the relationship between natural gradients of plant diversity and biomass productivity, using novel methods (within the discipline of ecology) for causal inference. A main finding is that, contrary to expectations, a higher species richness led to declines in productivity, which was mostly caused by rare-non-native species typically not used in biodiversity experiments.

I found the study very well written. I appreciated both the ecological insights of the study, i.e. that when correcting for biases caused by observed and especially unobserved covariates, causal effects of richness on productivity appear to be negative instead of positive (mostly caused by rare / invasive species), as well as for using a new analytical approach, that has so far hardly been used in ecological studies. At the same time, I have to admit that I am no expert on the analytical approach that was used, and that I hope that another reviewer can assess the validity of its use for this paper better than I can.

That said, I would have appreciated some more details on the analytical approach, as well as on the empirical data, in the main manuscript. At present, the main manuscript does not have any methods section, and the methods are either (very) briefly described in other sections (introduction/results), or in the supplements. Given that many readers might not read the supplementary materials, it would be good if the authors give more details on e.g. the data (currently in S3) and the Main Design Estimator (currently in S4) in the main manuscript. Obviously, I would not expect full details here, but I definitely would appreciate a bit more information on the methods in the main manuscript.

Another point that I felt was missing, is that while one of the main points of this manuscript is that B-EF relationships are more negative when accounting for unobserved variables (something typically overlooked by other observational studies), there is hardly discussion on what these unobserved variables could be. The authors corrected for >50 environmental variables, which is (as they also emphasize themselves) much more than most other studies do, but despite this, apparently there are other, not explicitly considered factors, also important in altering correlations between species richness and biomass production. It would be very interesting to know what these unmeasured factors could have been. This would also lead to testable predictions of future studies.

I also have a number of more detailed comments, given below:

Line 65-67: I would mention the name of the approach here.

Line 67: "which were the basis for a 2021 Nobel Prize": I would remove this, as it reads like an (implicit) argument based on authority on why the method is suitable, which I don't think is the type of argument that is most valid.

Line 67: although I realize the analytical approach is explained in more detail in the methods section, it would be nice to see one or two sentences explaining the core of the approach. Perhaps this can be done by adding another panel in figure 1.

Line 74-75: it would be good to shortly mention which climatic regions/ecosystem types are represented by the data

Fig. 1A: This figure suggests that in biodiversity experiments, there is no relationship between the proportion of common/rare species and species richness. While this may be true when such experiments are established (in most cases, they initially have maximum evenness), over time, in most cases some species become more dominant, while others become rarer or even extinct. So I would argue that it is premature to make this assumption, especially since many biodiversity experiments have publicly accessible data, so it would be relatively straightforward to either support, or reject this assumption after some simple analyses.

Fig 1B: in the caption, the authors rightly point out the possibility that productivity can also affect richness ("However, data on Z and M can help address unobserved confounders and differentiate the effect of R on P from the effect of P on R."). It would make sense to also depict potential effects of P on R in the figure.

Line 134-135: it would be good to mention a few of those un-accounted variables.

Fig 2. I really like the schematic overview of the assumptions of both the main and common design.

Line 187-189: Figure 3 shows, however, that while the richness->productivity relationship is still negative, it is not significant anymore. I think this nuance should be added in the main text.

Figure 4B: For clarity, I suggest to replace "R" by "richness"

Fig 4B, right side: it is a mathematical inevitability that when richness increases, the average relative abundance of species decreases: this is as true for natural, as for experimental systems. Therefore, I am not entirely convinced whether this helps to explain differences with experimental systems. Of course, in natural systems, abundance distributions often tend to be log-normal (and hence rather un-

even), which is different from starting conditions in most BD experiments. After a few years of establishment, abundance distributions in experiments are much more un-even than under starting conditions, although still on average (but not always) less uneven than in natural situations (Jochum et al, NEE, 2020). So my feeling is that while natural systems differ indeed in relative abundance distributions from experimental systems, a bit more nuance is needed in this figure, as currently one might interpret unevenness as something unique for natural systems.

Fig. 5: it is remarkable that even for native, non-rare species (presumably those typical for BD experiments), the richness productivity is still not significantly positive (although not negative either), in contrast to relationships in BD experiments. In other words, when rare or invasive species are corrected for, the relationship switches from negative to neutral, but does not yet become significantly positive. It would be nice to add some discussion on this.

Reviewer #3 (Remarks to the Author):

Dee et al. present a new design for analyzing the biodiversity-productivity relationship, drawing on the econometrics literature and leveraging an impressive dataset. Using their new analysis method, they found that biodiversity has a weak negative correlation with productivity in observational data. However, this statement obscures more interesting patterns related to rare and non-native species.

This is a thorough, well-written, and impressive contribution. I especially enjoyed the quality of writing in the introduction, the section on rare and non-native species, and the depth of sensitivity analyses in the supplementary material. The biodiversity-productivity relationship has been one of the major questions in ecology for decades and there is no question that this contribution would be of broad interest.

My main concern is that I didn't get a clear sense of what the new analysis is actually doing. I don't doubt that the authors are using it correctly, but the manuscript (to me) spent too much time arguing for the novelty of this approach and/or comparing it to previous analyses, and not enough helping the reviewer understand how the analysis works. Causal inference will not be familiar to many ecologists. I have made many suggestions in the attached pdf files (mostly the main text, but to a lesser extent the SI). In particular think it would be very helpful to walk through ecological examples when explaining the new analysis.

To be clear, I think readers will buy the idea that previous observational studies missed some confounding variables. But I think to seal the deal you need to do more to explain to readers exactly how your new analysis improves on this issue.

I enjoyed the FAQ section in the SI (even though it's not traditional, as the authors point out). I think the FAQ section is especially useful because the idea of positive biodiversity-productivity relationships is so widespread that there may initially be pushback against the authors' results, and the FAQ pre-empts many of these critiques. (I understand that the current manuscript supports the findings of BEF if one focuses on native, non-rare species).

Reviewer #4 (Remarks to the Author):

Review of "Clarifying the effect of biodiversity on productivity in natural ecosystems with longitudinal data and new methods for causal inference"

Summary: This manuscript applies methods from economics to estimate the arguably causal relationship between productivity and species diversity. The authors introduce the methods used by economist, and the show graphically how these methods can lead to causal insights. Their empirical example is convincing and finds a result that differs from what other studies have shown. Overall, the paper makes major contributions both methodologically and empirically. My hope is these methods will be more widely used in ecology, and the SI provides a useful primer on their use.

Major comments

1. The manuscript is well written and organized, yet it faces a very difficult challenge of introducing novel (to this field) methods while also showing a new, and perhaps unexpected, empirical finding that itself is of note for ecology. This is difficult to do in 3000 words. I have some suggestions for re-organization, although I encourage the authors to take them as suggestions, not orders.

a. Lines 101 – 106 feel unnecessary – the main point, that there are repeated timesteps is repeated in the next paragraph.

b. Lines 111-114 are also unnecessary – they do not actually discuss the study design, rather they are a claim about reverse causality. This claim, and the mechanism to control for reverse causality is discussed elsewhere.

c. I suggest the authors explicitly introduce the econometric idea of a “Fixed effect” in the main text. The “main design” is instantly recognizable to anyone familiar with econometrics as a panel regression with plot level fixed effects, yet this is never stated in the main text (although it is described nicely in the SI). I suspect the text will be easier for the non-economist if fixed effects are explicitly defined in the text and then this term is used in the text.

d. Fixed effects are described as “controlling for a much broader set of potential confounders.” In other places they are described as “for a broader suite of confounding variables.” Personally, I find this an unintuitive way to describe fixed effects. It makes it sound as though you know all the extra variables that are controlled for in the fixed effects model, when in fact one does not know what these unobservables are. Likewise, the way this is written, I would think many readers would say “why not just add these confounders to the mixed effects regression.” It may not be clear that what is being controlled for is unobserved. And of course, when you run the fixed effects model you have far fewer independent variables in the model, since all the time invariant variables are absorbed by the fixed effect.

i. You might consider instead describing the fixed effect as absorbing variation in the dependent variable that is the result of time invariant variables – both observed and unobserved.

e. Personally, I found the equations in the SI very helpful for understanding the three models. I think most people who will be interested in the models might find these useful as well. A potential reorganization would be to replace the last two paragraphs of the study design section with these three models.

f. This will have the added benefit of cleaning up the results section. Currently the results are a mixed of methods “We next estimate the relationship using a “Common Design in Ecology” – a multivariate design that mirrors advanced statistical designs that aim to control for confounding variables by directly measuring and including them as covariates in regression models (a “conditioning on observables” analysis like7). Such designs require a strong assumption for interpreting the correlation between richness and productivity as causal36: any site or plot attributes not include in the regression are assumed to be uncorrelated with species richness 129 and therefore not a source of statistical bias” is really a methods paragraph, not a results paragraph. Moving this to the study design section, and adding an equation might be helpful.

Reviewer #5 (Remarks to the Author):

Summary

The study investigates the causal effects of biodiversity (richness) on ecosystem function (productivity) on longitudinal data from 43 grasslands in 11 countries using a sophisticated fixed-effects methodological study design. Contrary to prior observational and experimental work, the authors find that increases in plot-level richness causes productivity to decline rather than increase. The authors attribute the differences to observational studies to prior work not controlling for further confounding factors, and to experimental studies since these plant fewer rare and non-native species than exist in nature. While increases in native species increase productivity, increases in rare species decrease productivity, which explains the difference.

Overall recommendation

I am not deeply familiar with the application field here to assess novelty and significance in this respect. But the authors transparently lay out in much detail the differences and relations to prior work. Their novelty lies in the fixed-effects causal approach and I can comment on this methodological part. I very

much liked the transparent way the authors communicate the assumptions and how effects change for different sets of assumptions. Figures 2 and 3 (and more in the SI) are a beautiful example of this. These allow for a transparent and informed discussion of how other assumptions might change results. I have some questions on several potentially relevant details. If these are better analyzed and explained, I would recommend the paper for publication.

Methodological approach

The authors' goal is to quantify the causal effect of plant diversity richness R on biomass productivity P from purely observational data in the presence of hidden confounding. While previous studies tried to overcome confounding by conditioning on a large number of covariates, the authors here make use of panel time series ("t") data at different sites "s" and plots "p" and then use the fixed-effects causal effects design:

$$\ln(P_{pst}) = \beta \ln(R_{pst}) + \delta_{ps} + \mu_{st} + \epsilon_{pst}$$

where β is the quantity of interest. Here hidden confounding by time-invariant plot attributes can be de-confounded by de-meaning (or introducing plot-dummy variables δ_{ps}), and hidden confounding by site-specific time-varying variables is attributed for by μ_{st} .

As the authors nicely and transparently lay out, this model depends on a number of assumptions that are extensively discussed and further tested in the comprehensive and well-written supplement. In the following I list some questions that I still have:

Positive effect in prior studies

The authors explain this by "First, on average, many of the observed site and plot variables [...] are negatively associated with richness and positively associated with productivity [...] unobserved site and plot variables (U_s , U_p , U_{st} in Figure 1B) are, on average, positively associated with both richness and productivity. Failing to control for them in the Common Design creates statistical biases in the positive direction. The Main Design, with its greater set of controls, is thus less biased"

You further expand on this in the SI, but still: Why is it that, coincidentally (?), the *observed* covariates tend to have a negative effect and the *unobserved* covariates tend to have a positive effect? The larger set of unobserved covariates includes the observed ones. You observe these effect directions, but do you have a good explanation for this "covariate-selection" bias here?

Controlling for Reverse Causality by blocking a mediator

Here the potential causal feedback $R \rightarrow P \rightarrow M \rightarrow R$ with M =shading is investigated. You block the M here, but M is a "forbidden node" in causal effect estimation since it is a child of the effect and induces a bias in estimating $R \rightarrow P$: However, you find no change in section S6b.i. That either means shading has no or only a weak relation with P or is maybe a cause of P , contrary to your graph above. Or what could explain this?

Controlling for Reverse Causality by instrumental variables

Here, as always in IV-designs, the credibility depends on whether the assumptions of IV are fulfilled and as a non-domain expert I cannot assess this here, but the authors explain this quite comprehensively.

Dynamic panel designs

Here I see some potentially more problematic issues and would need some clarification since I cannot fully follow the analysis:

- The text in section S6c says that your approach cannot deal with both of the links Pt-1  Pt AND Pt-1  Rt, but Fig. S7 doesn't show the three remaining options and misses the one with Pt-1  Rt. Later the text clarifies that it is about not allowing a mediating path Pt-1  Rt  Pt

- Regarding Fig. S6: The confounders U and I are both assumed invariant (=constant) over time. Does this mean that they are plot-specific attributes? But the fixed-effects setting can deal with this by adding a dummy-plot variable to the regression (as in Eq. (3)), no? This dummy would in the graph block paths through U and I. Then the only confounding would occur through Pt-1, but that one can just be conditioned on. I don't see any confounder path in the graph then.

If, however, one *cannot* condition on U, which is a direct confounder of Rt and Pt, then the case is indeed non-identifiable. The confounding due to I can always be conditioned away by including Pt-1 in the regression, even if it was time-varying.

- The graphs in Fig. 7 would also present no problem using a plot-dummy variable. Here one would *not* need to condition on Pt-1. (However, conditioning on Pt-1 would reduce estimation variance.)

- The effect in the graphs in Fig. 8 both can be de-confounded by Pt-1.

- Eq (5): It think what I don't get is why you don't include a dummy-plot variable together with a dynamic graph setup here. I have actually implemented a toy model to test this and it works.

- Fig S4: Here, if we assume away the reverse causality of Pt on Rt, then conditioning on the time-dummy variable μ_{st} would block all non-causal paths through ϵ_{1t} . From the graph there is no need to condition on the many factors as in Eq. (6). Rather, these would just increase estimation variance since you remove variance from the cause.

It could be that I have just misunderstood or overlooked something here, but please clarify.

Nonlinearity

The authors state "[Log-transforms] permit the underlying relationship in levels (untransformed variables) to be non-linear, while still preserving the regression model's assumption of linearity"

A log-transform only makes a fully multiplicative equation linear, that is, if

$$P = R * \text{confounders} * \text{noise, then} \\ \log(P) = \log(R) + \log(\text{confounders}) + \log(\text{noise})$$

But if $P = c R + \text{confounders} + \text{noise}$, then the regression model would not be as desired, no? I am not sure your other tested regressions $\log(P) = \beta R$, $\log(P) = \beta R^2$ etc cover all the cases. In essence, I would like to see what the assumed nonlinearity is that your transformation then makes linear.

Heterogeneous Effects of Rare, Non-rare, and Non-native Species on Productivity

Here in Eq. (9) you estimate the individual *direct* effects of five different species categories on productivity adjusted for time-invariant plot-level and time-varying site-specific confounders. Now what about interactions among the different species? Suppose $R1 \text{ -(+)-> } R2 \text{ -(-)-> } P$ and $R1 \text{ -(+)-> } P$. Then you would measure a positive effect in the second part, while the *total* effect of R1 on P may actually be negative, depending on the strength of the indirect and direct path.

But the experimental setup, I would presume, analyzes the *total effect* (including potential mediating paths through other categories) and not the direct effect. Hence, you might not be comparing like with like here. Could this alter your conclusions about the positivity and negativity of effects?

Lagged effects

You only consider lag-0 (within the same year) effects. What about lagged effects $R_{t-1} \rightarrow P_t$? Here I mean direct effects, of course with auto-causation in R also an instantaneous effect would lead to an indirect effect $R_{t-1} \rightarrow R_t \rightarrow P_t$. Maybe this issue is partly addressed in the FAQ on short vs long-term effects, but please clarify.

Selection/missing sample bias

You mention missing samples, but what percentage is actually missing? And can there be selection bias or missing sample effects, i.e., the reason for missingness is related to richness or productivity? I guess not, but just a thought.

Typos

"how our conclusions change (see SI: section S6c)"
- this should be just "section S6"

"casual"
- replace by 'causal'.

Final remarks

I would like to very much thank the authors for the whole way this article was written, the transparency, the figures, and the large SI were really a joy to read and made it easily accessible. I would hope more papers do this!

As a side remark: I found it interesting that you use causal graphs throughout, but cite potential outcome literature instead of causal graphical models literature (eg Pearl only in SI). None of the references 12, 13 and 35 (that you often cite) contains a single causal graph.

Sincerely,
Jakob Runge

Below, **the reviewer comments are in bold** and our responses are in regular font.

Summary to All Reviewers

We appreciate the time and attention that the five reviewers invested in reviewing our manuscript and their mostly favorable responses. We are pleased that four reviewers highlighted that our study is well-written, thorough, rigorous, transparent, and provides ecological and analytical insights of broad interest. As noted by two of the reviewers, a big challenge for our manuscript is to distill, in a readable format, several decades of advances in empirical research that are not widely used in ecology, while also providing new and important ecological results. The reviewer comments helped us improve our manuscript and particularly helped us identify where we could revise for clarity in the manuscript and the Supplementary Information (SI). The editor recommended that we add a Methods section, separate from the SI, which we think will clarify methodological questions raised by the reviewers and help readers who are new to these concepts feel less overwhelmed by the new material.

Major revisions to description of methods in this resubmission

A reoccurring theme in the reviews was the need for additional detail on our methods in the main text for readers who are unfamiliar with the methods. As R4 points out, we face “a very difficult challenge of introducing novel (to this field and discipline) methods while also showing a new, and perhaps unexpected, empirical finding that itself is of note for ecology.”

To meet this challenge, we made two major revisions: 1) we revised the main text to include more intuition on the methods, and 2) we added a dedicated Methods section. With more information on the methods and the underlying intuition in the main text and Methods section, the reader does not need to read the SI to understand the main analysis (R1, R2, R3). The Methods section addresses the following points raised by the reviewers:

- (1) Include a more formal definition of the causal effect we are estimating and revise that definition to be more precise (R1).
- (2) Add more information on how our approach controls for unobserved and observed confounding variables (R1, R2, R3, R4).
- (3) Add additional details about each of the robustness checks – and about the sources of potential bias that each check addresses - so that a reader does not have to review the entire SI (R1, R2, R3, R5).
- (4) Add estimation equations for the Main Design and the robustness analyses, so that readers can easily see how the estimation works without having to review the entire SI or read the publicly available code or tutorial on GitHub (R1, R2, R3, R4).
- (5) Improve the exposition around the dynamic panel designs, which we did through new text in the Methods section and revised text in the SI (R5).

To highlight our extensive revisions to main text and the SI, we have also uploaded a marked-up document with tracked changes.

Response to Reviewer 1

Summary

We greatly appreciate the time that you dedicated to reading our manuscript and to providing detailed feedback. The feedback forced us to sharpen our language to clarify our aims and methods, which has improved the manuscript. Among the five reviewers, you were uniquely negative in your review. Although it can be challenging to understand from where a reviewer is coming based solely on the reviewer's comments, we believe that your review is approaching our research question from a different "school of thought" about causality in complex natural systems, which may lead to a misunderstanding between you and us about our target parameter (what we're trying to estimate) and the designs that we use to estimate that target parameter. Below, we briefly try to explain what we believe are the roots of these misunderstandings and how we have revised our writing to be more precise and, hopefully, to reduce misunderstandings.

We hope that through our responses and changes, we make a convincing case that our approach to the research question is valid, even though it may differ from how you would approach the question, and that our study advances ecological science and debate by bringing in new ideas and perspectives for attacking important causal questions in ecological systems.

Your comments imply a familiarity with a different perspective on making causal claims in ecology, which is exemplified by the convergent-cross mapping (CCM) approach, rather than the design-based approach we use here. Therefore, we try to first elaborate on the differences between our approach to causal inference and the approach to causal detection embodied in modeling approaches like CCM and referenced in your comments (note that Adam Clark and Jane Cowles in the "Clark et al" study in your comments are also co-authors on our manuscript and recognize how our approach differs from CCM-like approaches; note also that their 2015 paper did not apply CCM to study biodiversity effects). We hope that this elaboration will help clear up some of the misunderstandings that we believe exist between us. After this summary, we address each of your original comments in detail.

Our three key points are: (1) *Different Notions of Causality*: the notions of causality in our approach and in modeling approaches exemplified by methods like CCM are not the same; (2) *Different Objectives*: rather than seek to detect a causal relationship, which is the objective of modeling approaches exemplified by methods like CCM, we seek to quantify the sign and magnitude of the average causal effect of species richness on productivity; and (3) *Different Assumptions*: all empirical approaches to making causal claims rely on untestable assumptions and these assumptions may vary depending on the research goal and data available.

Causal Relationships in Ecology

Our approach to causal inference is a "design-based" approach, which looks to randomized experiments as the ideal benchmark for inferring causal relationships from data and for estimating the magnitude of causal effects (Rubin, 2007 *Statist. Med*). A design-based approach,

in the form of hundreds of randomized biodiversity experiments, has been the primary tool for causal inference about the effect of biodiversity on productivity (e.g., Naeem et al 1994 *Nature*, Tilman et al. 2001 *Science*, Hector et al. 1999 *Science*, Tilman et al. 2006 *Nature*, Reich et al. 2004 *PNAS*, Reich et al. 2012 *Science*; reviewed in Cardinale et al. 2006 *Nature*, Balvanera et al 2006 *Ecology Letters*, Hooper et al. 2005 *Ecol. Monographs*, Tilman et. al 2014 *Annual Review of Ecology, Evolution, and Systematics*).

The notion of causality in the design-based approach is often called an “intervention-based” notion of causality (Pearl, 2009, *Causality*; Holland, 1986, *JASA*). The idea is that, in principle, the causal variable can be manipulated and take on different values. That manipulation may happen via humans or via natural forces. In the design-based approach, the dominant way of thinking about a causal effect is the difference in some value, Y , under two states of the world (e.g., productivity in a plot when richness is two species and productivity when richness is four species). Only one of these states of the world will be observable at a specific point in time and space – the other “counterfactual” state must be estimated from data. Thus, we can summarize in another way what the design-based approach aims to do: to estimate counterfactual outcomes based on observable outcomes. That estimation is our aim here, which, thanks to the reviewer’s comments, we now clarify more explicitly (see added text defining our target causal effect in the new Methods section). As we explain below, this notion of causality differs from the notion of causality that is often used in time-series modeling approaches, like CCM.

The empirical focus of the design-based approach is often called “identification.” Roughly, identification is an attempt to specify plausible conditions (assumptions) under which one can infer that a correlation reflects (or “identifies”) a causal relationship and that the estimated magnitude of this correlation reflects the true magnitude of the causal relationship. In a randomized experiment, for example, identification requires untestable assumptions that are often called “excludability”, “interference” and “consistency” assumptions (see Kimmel et al., 2021, *TREE*). The design-based approach to causal inference has been developed through decades of advances, tracing its roots back to the experimental research of A. Fisher and J. Neyman, which was then extended into the non-experimental domain by statisticians D. Rubin, J. Robins, and many others.

The design-based approach is essentially a two-step approach: a statistical estimation step and a causal interpretation step. For the estimation step, much of the research using design-based approaches, in both experimental and non-experimental studies, is focused on estimating an average causal effect for some population or sub-population – in other words, not just detecting whether a causal relationship exists or not, but also estimating its sign and average magnitude. More precisely, researchers using this approach often aim to estimate a conditional expectation function (CEF). The CEF can summarize the predictive power of changes in richness on productivity (whether these changes are causal or not). The CEF for plot-level productivity, p_i , given a level of richness, r_i (and perhaps other covariates), is the expectation, or population average, of p_i with r_i held fixed: $E[p_i|r_i]$, a function of r_i . If we only had two richness values

(1=high, 0=low), the CEF would take on two values: $E[p_i|r_i = 1]$ and $E[p_i|r_i = 0]$. The average causal effect of moving from low to high richness would be a difference in the CEF at each value: $E[p_i|r_i = 1] - E[p_i|r_i = 0]$. With more richness values, one would have more potential causal effects (more contrasts of values of r_i). The CEF is a common target for causal inference because it has many well understood properties and has a close connection to saturated regression models (Angrist and Pischke, 2009, *Mostly Harmless Econometrics*). To be used for causal inference, these regression models require researchers to do two things: (1) characterize how inference from an observed sample to the unobserved population will take place (statistical inference); and (2) characterize the conditions under which one can interpret the output from the CEF as reflecting a causal relationship (causal inference). We do both, but it is this second step that is the focus of our study.

The difference between (a) an estimate of the causal effect delivered by a statistical estimator and (b) a true causal effect that arises from a change in a variable in the system is (c) the estimation error. This error can be described as having three parts (i.e., a function of three elements):

$$\text{Estimation error} = (\text{Estimate from estimator} - \text{True causal effect}) = f(\text{Design bias, Misspecification bias, Statistical noise})$$

Design biases come from unobserved confounders and reverse causal relationships in the data. In experimental studies, researchers try to eliminate these biases via randomization of the treatment. In observational studies, researchers try to eliminate these biases through other means (e.g., statistical control of observable confounders). Modeling biases come from misspecification of statistical models that are used to draw inferences about the conditional expectation function. When seeking to quantify a causal effect, researchers cannot avoid specifying a model. Researchers try to eliminate misspecification biases through careful selection of flexible parametric or non-parametric estimators. Statistical noise comes from sampling error. Researchers try to eliminate noise via control over the sampling procedures and environment.

A researcher's aim is to reduce design bias, misspecification bias, and statistical noise to zero. Of course, these sources of error can never be reduced to zero, and thus researchers need to bound their uncertainty about the degree to which one source of estimation error could change their conclusions; for example, confidence intervals to quantify uncertainty about statistical noise, and sensitivity bounds to quantify uncertainty about design bias. To judge the quality of the study, a reader will assess the way in which the researchers reduce or quantify estimation error. In our study context, a reader would assess whether our "identification assumptions" are plausible and whether there exist plausible alternative (rival) explanations for the inferences that we consistently draw across various sets of assumptions and sets of methods for quantifying uncertainty about the sources of estimation error.

When we read your comments, we infer that your concerns are not related to the way in which we characterize uncertainty related to statistical noise, but rather how we address design bias. Your comments don't include the phrase "design bias," but they raise concerns about how we address potential confounders and reverse causality. Moreover, your comments imply that we have introduced misspecification bias by relying on linear models to estimate the CEF. In our detailed responses to your comments below, we hope to convince you that (1) we have done far more than any other observational ecological study to address sources of design bias; and (2) misspecification bias is unlikely to be an important source of estimation error in our study design.

Your comments, including your encouragement that we read Clark et al., seem to suggest a different approach for attacking causal questions in ecological systems: a prediction-oriented, modeling-based approach where a one specifies a dynamic manifold or other 'structural model.' The CCM approach is an exemplar of this approach. In CCM, scholars detect causation from data by matching subsets of consistent dynamics between two timeseries – that is, identifying time periods in timeseries A that share similar dynamical histories, and testing whether these correspond to time periods in timeseries B that also have similar dynamical histories, using nonparametric methods that make few assumptions about underlying functional forms and processes. Such approaches differ from ours in two important ways.

First, causality in these approaches is assessed based on correspondence (and convergence) between model predictions and observed data. This predictive-based notion of causality, however, is different from the intervention-based notion in our design-based approach. To differentiate this notion of causality from intervention-based notions, users of the modeling-based approaches often use phrases like "X CCM-causes Y."

Second, CCM methods largely focus on "detecting" and "discovering" *whether* a causal link among variables exists, rather than attempting to estimate the *strength and direction* of the underlying relationship. Although a large and growing number of methods have been developed to extract effect size proxies from CCM-like analyses (e.g., Empirical Dynamic Modeling (EDM) tools), they have not yet fully addressed the potential sources of bias from a wide range of confounding variables, which are the focus of our analysis. In the broad class of predictive modeling-based designs for timeseries data, of which CCM is one example, confounding factors must be addressed either by observing them all or by making additional assumptions. In these designs, the ability to observe all confounders (all "direct common drivers") is often called the "Causal sufficiency" assumption, and it is combined with Causal Markov Condition assumption to make causal claims from the data (the latter assumption states that, in a graphical model, an outcome *Y* is independent of every variable not affected by *Y* conditional on *Y*'s direct causes; see Runge et al., 2019, *Nat Comm.*). Most CCM-related approaches that aim to estimate effect sizes, such as those from Deyle et al. (2015, *Proc Royal Society B*), rely on locally weighted ordinary least squares (OLS) estimates of effect sizes. Thus, although EDM approaches do an excellent job at addressing effects of complex, highly nonlinear dynamics, without additional

corrections they are still vulnerable to all of the classic problems that arise when attempting to draw causal inference from regressions. That said, a wide range of EDM-like extensions are being developed that use more sophisticated techniques to estimate interaction coefficients, such as Censi et al. (2019, *Methods in Ecol & Evol*) for noisy data or Chang et al. (2021, *Ecology Letters*) for very high dimensional data. We are therefore optimistic that, in the future, many of the causal inference approaches that we introduce to ecologists in our study will make their way into EDM and CCM-like techniques, which will greatly expand the opportunities for causal analyses in complex dynamical systems.

Beyond the differences between the two approaches in terms of their notions of causality and their ability to quantify, in an unbiased way, the causal effect of one variable on another, approaches like CCM would be challenging in our context given the structure of our data. The broad distribution of sites in the Nutrient Network is valuable from the perspective of generalizability, but it also poses a problem for CCM due to highly heterogeneous dynamics among sites and regions and relatively short time series. In their 2015 paper, co-authors Clark, Cowles, et al. did indeed show that multiple short timeseries could, in some cases, be stitched together to form synthetic long timeseries for CCM analyses (“multi-spatial CCM”). However, to be stitched together like this, the timeseries must be dynamically similar enough that observations from one site can be extrapolated to make predictions across many other sites, but also dynamically different enough that they cover distinct regions of the dynamical manifold. These requirements, as well as suggestions for diagnostic tests, are discussed in detail in Hsieh et al. (2008, *American Naturalist*). In a test analysis of the Nutrient Network data, we find that classic simplex embedding produces cross-site predictions with a Pearson correlation coefficient of less than 0.2 between predictions and observations for both richness and productivity, indicating much too little correspondence in between-site dynamics to apply multi-spatial CCM. We speculate that, with much time and effort, carefully chosen multivariate embeddings (or, more likely, multi-view embeddings, *sensu* Ye and Sugihara (2016, *Science*)) could be found that could produce relatively accurate forecasts for a significant subset of the biodiversity and richness dynamics observed in Nutrient Network. However, finding these embeddings would be a research project in its own right, and to our knowledge, the methods needed to apply CCM to multivariate or multi-view embeddings have not yet been developed.

The CCM approach, and its subsequent extensions, were developed as a response to very simplistic use of linear statistical models in which the correlations estimated from those models were interpreted, after some statistical test, as reflecting causal relationships and the absence of statistically relevant correlations was interpreted as reflecting the absence of a causal relationship (i.e., so-called “Granger causality” models, in which the notion of causality is also different from ours, with users often writing “ X Granger-causes Y ”). That simplistic linear approach is extremely problematic when applied to time series data in non-linear systems, as pointed out by Sugihara and collaborators (*Science*, 2012).

Our design is a response to the same concerns about the misuse of simple statistical models that can mimic or mask true causal relationships. But our design addresses the concerns by leveraging different data (panel data), applying different assumptions, and targeting a different causal parameter (an average effect of changing from one value of richness to another value, which we do indeed allow to be nonlinear in our analyses). Those who developed the approaches we use (see references in main text) made it clear that the use of linear statistical models per se is not the root of the problems identified by Sugihara and collaborators but rather the simplistic ways in which those models were implemented in prior studies. We avoid such simplistic implementations.

In sum, we want to emphasize that we do not see our approach as superior to the approaches that your comments seem to encourage us to adopt, but rather we see it as complementary in getting at the very challenging question of causality in complex ecological systems. All approaches to causal inference rely on untestable assumptions, and different approaches often answer different questions (i.e., estimate different parameters). We drew our causal inferences under much weaker assumptions than prior studies have used, and we drew similar inferences across a wide range of plausible causal assumptions. In other words, we believe we have made the best attempt to date to address estimation errors in the biodiversity-productivity literature and have done so with an approach that can be used to answer myriad other causal questions in ecology. As we have tried to emphasize in our manuscript, we cannot find any evidence for a positive average effect of richness on productivity except when we use empirical designs that match what ecologists typically use, which we have explained are biased towards finding a positive effect. We understand that estimating, without bias, an average effect of changing species richness across multiple richness levels, multiple sites, and multiple time periods is not the end goal for ecologists, but we believe that this estimate is an extremely important first step that has not yet been accomplished in ecology. Future studies can further explore the role of moderators and mediators in these systems.

Reviewer 1 Comments

This work, borrowing the approach of causality analysis developed from econometrics, investigates biodiversity effects on productivity, using longitudinal observational data. I agree author' arguments in introduction: 1) experimental design for BEF research is unrealistic and cannot be generalized to natural system; thus, longitudinal (time series) data from natural systems are needed. I also agree 2) confounding variables create problem in studying Biodiveristy <-> Ecosystem Functioning; however, I am not convinced that the linear decomposition approach used in this work has solved the issue. Further, I like to point out 3) another critical issue is that BEF interaction is embedded in complex feedback systems (Tilman et al 2014 Annu. Rev. Ecol. Evol. Syst.). For example, biomass->nutrient->diversity->biomass (a cyclic path in network terminology). Other more complex feedbacks exist in natural systems. This critical issue was not correctly resolved by their approach.

We hope that our summary text above helps address some of these comments. Here, we first clarify how we interpret your comments. Then we explain how we have revised text in the manuscript to clarify our points and reduce the likelihood that another reader will have the same reaction.

To address the first comment about linear decomposition, we believe calling our approach a “linear decomposition” is not an accurate characterization of what we do. As noted above, we aim to estimate a CEF and then conduct statistical and causal inference about the values of that CEF. Non-linear relationships between the confounding variables and productivity or richness, as well as linear and non-linear interactions among those variables, are addressed in our Main Design through the plot and site-by-year “econometric fixed effects” in our estimation model. The fixed effects are powerful in their flexibility (i.e., we do not need to make any strong functional form assumptions for inference). They can control for any time-invariant (or slow-changing) factors over the study period, including interactions of these factors at the plot and site level, and they control for any time-varying factors, or interactions of these factors, at the site level. In other words, given our target estimand for our Main Design (average difference in CEF values), the linearity of our model is a strength rather than a weakness. Moreover, the instrumental variable (IV) design, when its key identifying assumptions are valid, also eliminates the confounding effects from non-linear relationships between the confounding variables and productivity or richness, as well as linear and non-linear interactions among those variables.

To make these points clearer in the manuscript, we have added a Methods section (lines 327-644) where we define our target estimand (lines 327-349), elaborate the specific model specifications used in the analyses (see Equations 1- 6) and the intuition for how they address confounding variables, and explain how our approach can capture non-linear functions of confounding variables and their interactions (lines 445-447).

In response to the second comment about lack of attention to feedbacks, we are not clear about why the issue seems “unresolved” in our analysis. We would like to emphasize that, as we wrote in the original manuscript (previous manuscript lines 189-190), we are not saying that there are no feedbacks (i.e., no cyclic paths). We are instead saying that our analyses suggest that these other feedbacks do not invalidate or greatly confound our estimate of the relationship between richness and productivity. Your comments correctly note that ecological feedbacks exist, but they do not describe how our three analyses for assessing the potential bias from effects that run from productivity to richness are flawed (i.e., the Instrumental Variables design, dynamic panel designs, and mechanism blocking design).

To make these points clearer in the manuscript, we have revised the main text to explain and elaborate on the dynamic panel modeling approaches, which were previously only described in the SI. In response to your comments and comments by R5, we have also rewritten the SI section on the dynamic panel designs (Section S6bi) to further clarify what feedbacks they capture and to report on dynamic models that are not reported in Fig. 3 but whose results are similar to those reported in Fig. 3.

This study made little progress beyond Grace et al 2016 (Nature). In fact, Grace et al. found positive BEF effects based on a dataset to some degree overlapped with the data used in this study. To compare with Grace et al (2016), the authors made extensive paragraphs (in the SI) to say that the difference in finding is due to the difference in approach. Well, how to reconcile the opposite findings then? Grace et al. (2016) used a global-scale meta-analysis and applied SEM to rule out the confounding effects explicitly, whereas the present study accounts for the potential confounding factors using site and/or plot effects.

In response to the internal review process with our coauthors that include many coauthors from the Grace et al. paper (2016, *Nature*), we went to great lengths in the manuscript and the SI to clarify how our analysis builds on and advances that paper. To help readers to understand both how the studies differ and why those differences yield different results, we included ample detail in our main text and our SI, which we summarize below. To avoid making the entire focus of the main text in our manuscript “How is Dee et al. different from Grace et al.?” we refrain from placing more emphasis on elaborating the differences between our study and Grace et al.

So that you do not need to re-read the manuscript, we first reiterate four of the advances that are present in our study and missing in Grace et al. and other observational BEF studies. Then we revisit how design differences between our study and Grace et al. lead to differences in results.

First, our design controls for a much broader range of confounders than the Grace et al. design. Our analysis uses both *spatial* variation across sites and plots and *temporal* variation across sites and plots. Leveraging temporal variation across sites and plots is a key innovation of our study. In Grace et al., the authors use *only spatial* variation across sites and plots – said another way: Grace et al. do not use temporal variation at all, and only include one year of data per plot (note: Grace et al. do not use a “global-scale meta-analysis”). Our design eliminates the spatial variation that comes from the “between-plots” comparisons because we believe those comparisons will yield biased inferences about the relationship between richness and productivity – i.e., hidden bias that comes from unobserved confounding variables (see our accompanying Rmarkdown tutorial for elaboration and a visual).

Second, our design exerts better control over potential bias from the effect of productivity on richness. We address the threat from reverse causality using three different designs: an instrumental variable design, a mechanism-blocking design, and a dynamic panel design. Each design lays out plausible, but different, assumptions for assessing the potential threat to inferences from reverse causality (complex feedbacks from productivity to richness). To assume no bias from reverse causality in the Grace et al. design, one must assume that their soil suitability index measure, comprised of three soil attributes, affects productivity but not richness, after controlling for the small set of covariates in their structural equation model. The assumption that soil attributes affect productivity but not richness is, in our view, a very strong and implausible assumption. Third, we explore how our inferences change with plausible changes in

causal assumptions, a hallmark of frontier studies in causal inference but a feature that is lacking in ecological studies. Fourth, as noted by other reviewers, our study is much more transparent about its causal assumptions and its target estimand (the causal effect we aim to estimate). All of these features make our study an important advance over Grace et al. and all of the other prior observational studies in the BEF literature.

Some of our coauthors read an early version of this manuscript and asked the same question posed in your comments: how it is possible that Dee et al. estimate a negative average effect of plot-level richness on productivity while Grace et al. estimate a positive effect, even though both studies use unmanipulated plots from the Nutrient Network? To answer their question and convince them of the veracity of our analysis, we created a "Frequently Asked Questions" (FAQ) document. We included those FAQs in the Supplemental Information (SI) file (Section S9) because we anticipated that readers of the study would also have the same question. FAQs #5-#7 relate to how and why the analysis design in our study is an advance that builds on Grace et al, and why the results differ due to important design differences. In our original manuscript, we had summarized the key ideas from these FAQs in the main text (lines 145-160).

In our revised manuscript, we have changed the main text that summarizes these key ideas by adding more sentences and working to make the original sentences clearer in the Results section (lines 175-188). Furthermore, we have retained the FAQ in the SI, where readers can go for extensive details on exactly how the two studies differ in design and why they generate different answers to what appears to be the same question. In FAQ #5, we make it clear that the differences between the structural equation modeling in Grace et al. and the single regression models of Dee et al. do not account for the differences we observe in the results. In FAQ #6, we make it clear that both studies aim to estimate the same causal parameter (same estimand).

FAQ #7 speaks to the crux of your comment. For your convenience, we reproduce the first paragraph of the FAQ here:

“Why is the estimated average effect in Dee et al. negative while in the Grace et al. study it is positive? Is it because each study makes different assumptions about what is driving changes in richness and productivity?” In short: Yes, we believe this reason is the main reason for the different results. The key insight from Dee et al. is that imposing different causal assumptions (summarized in Figure 2) leads to different causal models, and different models can yield different conclusions. If the underlying assumptions were false, the estimated correlation between richness and productivity may not reflect a causal relationship. The key distinction between Grace et al. and Dee et al. is in our analysis study *designs* – and the ability of our design to rule out more confounding variables than included in Grace et al.’s design.”

On pages 64-74 of the SI, we contrast, in detail, the assumptions being made in the two study designs – i.e., our Main Design (Figure 2) and the Grace et al. design. We show precisely how the assumptions in the two designs differ and thus can lead to different conclusions. We describe

how Grace et al.'s assumptions are much more restrictive (less plausible) than our assumptions. Importantly, and unlike Grace et al., we also use multiple designs that each require different assumptions for drawing casual inferences from the data (Figure 3). Through those designs, we probe the robustness of our results to violations in the assumptions in the Main Design, which is not done in Grace et al. or in most ecological analyses. The alternative assumptions in our complementary analyses are described in detail in the SI (section S6).

After describing in detail how causal inferences in the two studies rely on very different sets of assumptions, we explain how these differences could explain the opposite signs of the estimated effects in the two studies in the main text (lines 166-188). The **FAQs #6 and #7** in the SI elaborates the intuition through an extensive, step-by-step explanation and three supplementary analyses that support the explanation (Section S9, pp. 65-72). Here, for your convenience, we reproduce the final two paragraphs of this FAQ:

“What’s happening? The Nutrient Network sites are likely to experience site-specific “shocks” that vary each year (e.g., weather shocks, like a particularly dry April, or herbivory shocks, like higher herbivore pressure than the prior year). We don’t know what exactly these shocks are, but because we observe the same sites over many years, we can control for them. In our data, we observe that these shocks affect productivity and richness in the same direction, on average. We also observe that many of the observable variables collected by Nutrient Network researchers affect productivity and richness in opposite directions, on average. So, as noted by Grace et al., when one looks at a single year of data, the bivariate correlation will not show anything of note. If one does not have a valid IV, then when one controls for observable confounders (like soil attributes), one will see a positive correlation between richness and productivity. However, when one controls for a much larger range of sources of positive and negative bias, as done in Dee et al., you see an overall negative average effect. *[In the main text, we elaborated on this point about site-specific, unobserved shocks (see line 175)]*

So, ‘Yes,’ we believe that some or all the difference in results between Dee et al. and Grace et al. is attributable to the studies making different assumptions. Which, if any, of these sets of assumptions better approximates the truth is something a reader must decide. Future research could focus on assessing the ecological conditions under which each set of assumptions may be plausible.”

From statistical point of view, I am not convinced that including site and/or plot as effect variables solves the issue of confounding variables and the feedback problems. Moreover, the framework of “linear decomposition” of productivity variation into diversity effects + the plot and site effects does not fit in the real-world situation; that is, BEF behaves as context-dependent complex feedback systems. I have several critical concerns on their approach.

Below, we address each of the numbered comments in your review. Here, we address two potential misunderstandings. First, we hope our discussion above in response to your concerns about “linear decomposition” have clarified the average causal effect that we are aiming to estimate, how our designs make plausible assumptions for interpreting our estimates in a causal way, and how our model specifications do indeed address, in a flexible way, the potentially complex interactions and feedbacks in these systems. Second, in our original manuscript, we did *not* write that the site-by-year and plot effects solve the feedback issue. We used other approaches to address potential sources of bias from feedbacks (i.e., the IV design, the mechanism-blocking design, and dynamic panel designs). In response to questions by R5, we have also expanded and clarified our exposition about the dynamic panel designs in the SI (Section S6bi – starting at pp. S-31).

6. I have mathematical concerns on their definition of “causal effect”. The authors define: “the causal effect of R on P is the difference in productivity between two states of the world in which all other factors are held constant except R “. This statement is not mathematically correct. The key problem is that other factors are NOT held constant and cannot be assumed held constant. Rather, a better mathematical definition of context-dependent causal effect is partial derivative. That is, $\partial P / \partial R$ conditional on the instantaneous state of other confounding variables.

As we noted above, we believe that we could have been clearer in describing our target causal effect. We believe that our mathematical notation was clear, but the English around it, in retrospect, was not clear or precise enough. We should *not* have written, “all other factors are held constant except R” because that phrase would imply a change in richness does not set in motion a change in other factors. In response, in the new Methods section, we have added a subsection called “Target Causal Effect” (starting at line 333), in which we provide a precise definition of what we seek to estimate in our study. We reproduce it here:

“To formalize the causal relationship we seek to estimate, we use the potential outcomes framework⁵⁵⁻⁵⁷. The causal effect of a change in richness from R^I to R^{II} on productivity P in plot I is defined as $[P_i(R^{II}) - P_i(R^I)]$, where $P_i(R^{II})$ is the potential productivity outcome when $R = R^{II}$ and $P_i(R^I)$ is the potential productivity outcome when $R = R^I$ ($R^I \neq R^{II}$). The difference in these two potential productivity outcomes (i.e., productivity under two potential richness conditions) is the causal effect of a change in richness in a plot. For a specific location and time, only one of these potential outcomes will be directly observable; the counterfactual values for the other potential outcomes must be estimated from data. The average causal effect of a change in biodiversity from R^I to R^{II} across all plots is $E[P_i(R^{II}) - P_i(R^I)]$, where $E[\cdot]$ is the expectation operator. We seek to estimate the average causal effect of a one-unit change in R across all plots (i.e., the average effect across all possible one-unit changes). When used for causal inference, non-experimental studies aim to replicate, conceptually, the idealized experimental design in which the factor or factors that affect variation in R only affect P via their effects on R . In other

words, to permit credible causal inferences, a non-experimental design seeks, via design and statistical methods, to eliminate the confounding effects of U_p , U_s , U_{st} , and U_{pt} in Figure 1, as well as the effects of reverse causality in Figure 3.”

Note that an average causal effect is defined by the units in a population and their attributes and the point in time at which the causal variable changes. Calculus, by itself, cannot communicate about causality (Pearl, 2009, *Causality: Models, reasoning, and inference*). The partial derivative dP/dR is the rate at which P changes when R changes, net of the effects of other variables in the model that influence P . In fact, what we are estimating is the derivative: dP/dR – the rate of which P changes when R changes. More accurately, we are estimating $dP/d(do-R)$, as noted by Pearl (2009, *Causality*) who developed the do-calculus to allow researchers to express causal concepts using calculus.

With respect to the context-dependence of the causal effect, we thought we had acknowledged and stated this point explicitly in the text (lines 303-310) and implicitly by acknowledging potential site and plot-specific moderators (lines 163-165). For example, lines 303-310 say, “Our results also imply that any estimated effect of changes in species richness on productivity may not generalize to different spatial or temporal scales, or to other ecosystems, forms of biodiversity, or ecosystem functions. Multiple ecological mechanisms underlie a relationship between richness and productivity (e.g., below-ground competition, niche complementarity), and their strength may vary across places and time depending on which types of species are changing and how (Figure 4). When the operative mechanisms depend on the version of richness that is changing, ecologists face what has been called “treatment-variation relevance”⁵³ or “consequential variation of the treatment”⁵⁴. In such cases, interpreting and generalizing causal effects is challenging, whether ecologists use experimental or observational designs.”

Moreover, the context-dependent nature of our target causal effect is clear from the conceptual framework that we use for describing this effect: the Rubin-Neyman potential outcome framework, which has been shown to be equivalent to the Pearl do-calculus framework (Morgan and Winship, 2015, *Counterfactuals and Causal Inference*). Causal effects are always context-dependent, defined by the units in the population and their attributes and the point in time at which the causal variable changes. In our study, we are measuring an average effect across sites and years, but we recognize and state that the average effect in our study population during our study period may not generalize to another place or time (lines 303-308). We do not assume “transportability” of our results – in fact, our conclusion is precisely that heterogeneous causal effects are what drives the difference between our results and prior experimental research results.

2. Real-world systems are nonlinear dynamical systems. Time-invariant attribute does NOT exist in nonlinear dynamical systems. The linear model employed in this study assumes that the biodiversity effects and other confounding variables (whether they are

observable or not) have additive effects on biomass (function). Unfortunately, in real worlds, biodiversity and other variables are nonlinear interactive (context-dependent) and are NOT linear decomposable (those effects are NOT additive). Essentially, there is nothing called “invariant diversity effect” in natural systems. In reality, diversity effect always depends on other variables (e.g., T, nutrient, precipitation) in a nonlinear way (the dependency cannot be linearly partial out as executed in this study); that is, diversity effect is time-varying in natural systems. The effects from confounding variables (e.g., T, nutrient, precipitation) are also not invariant and cannot be linearly partial out. Assuming the confounding variables as instrumental variables hidden by sits or plots effects does not resolve the problem. In fact, BEF has been studied from nonlinear dynamical system viewpoint, e.g., Clark, et al. (2015) Ecology.

The comment has several points, to which we respond and clarify here and in the text.

- **“Time-invariant attribute does NOT exist in nonlinear dynamical systems.”** We disagree. By time-invariant attributes of the site or plot, we mean attributes that do not meaningfully change in value over the study period and thus are controlled for by studying within-plot variation in richness. Examples include truly time-invariant attributes, like plot history, as well as slow-changing attributes, like topography and soil structure. The smaller the panel is (the fewer the number of years), the more plausible is the idea that such attributes are common confounders. Depending on the plot, the panels run from 5 to 11 years, with a mean and median of 8 years. In a statistical sense, we are talking about systematic, persistent effects on productivity that come from the plot or site. There is nothing about the definition of a nonlinear, dynamical system that rules out time-invariant attributes of the system. To make sure readers understand what we mean by this phrase, we updated the text (lines 119-123): “Confounding variables that affect both richness and productivity could arise from conditions at the plot or the site. The values of these variables may be essentially invariant during the study period (e.g., soil texture, topography, land-use history) or they may vary through time (e.g., surrounding land-use change, drought conditions that differ by both site and year).”
- **“in real worlds, biodiversity and other variables are nonlinear interactive (context-dependent) and are NOT linear decomposable (those effects are NOT additive).”** We hope that our extensive discussion in the previous pages has cleared up this misunderstanding.
- **“Essentially, there is nothing called “invariant diversity effect” in natural systems.”** We are not certain to what exactly this comment is referring here. However, we hope that our text in the previous page has made it clear that we do not believe in such invariance (i.e., that diversity itself has an invariant or constant effect on productivity) and never made such a claim in our study.
- **“In reality, diversity effect always depends on other variables (e.g., T, nutrient, precipitation) in a nonlinear way (the dependency cannot be linearly partial out as**

executed in this study.” We hope that our extensive discussion about our target causal effect (i.e., an average effect represented in terms of CEFs) has cleared up this misunderstanding. We absolutely agree that the effect of diversity on productivity depends on other variables (i.e., there are system attributes that moderate the causal effect across space and time). We are not in any way “linearly partialing out this dependency.”

- **“Assuming the confounding variables as instrumental variables hidden by sits or plots effects does not resolve the problem.”** We do not assume that “confounding variables are instrumental variables.” We do not understand what this comment means. Instrumental variables are a solution to mitigating bias from unobservable confounders. They are not confounding variables and we do not assume they are. We refer the reviewer to our description of instrumental variables in the Methods (lines 491-538; equations 4 & 5) and the SI (section S6a.ii).
- **In fact, BEF has been studied from nonlinear dynamical system viewpoint, e.g., Clark, et al. (2015) Ecology.** As we explained on pp. 2-6 of this response document, the Clark et al. approach is different and not, in our view, superior. In particular, it is not appropriate for quantifying the causal effect we seek to quantify. Moreover, because of data limitations, Clark et al. (2015) do *not* use the CCM approach to study the BEF relationship in their study. They use CCM to answer a different ecological question (recall that two of the authors of the Clark et al. 2015 study, including the first author, are co-authors on our study and agree with our assessment of the contrast between our study and the Clark et al. study).

3. Natural systems (in particular the BEF issue) cannot be described as DAG. Obviously, biomass may influence nutrients (or other variables in soils) which then affect diversity (or species composition), which then affects biomass. A feedback cycle in the other direction (diversity->nutrient->biomass->diversity) also exists. In nature, BEF is a feedback control system and thus cannot be modelled as DAG.

Any temporal sequence of events can be represented by a DAG. The effect of changes in richness on productivity and the effect of changes in productivity on richness do not occur at exactly the same time. A species appears or disappears, and then subsequently any effect of that change in richness on productivity will occur. As effects always run from past conditions to current conditions, the path from a perturbation of richness to a change in productivity can be represented by a DAG. For simplicity in our presentation in the main text figures, we do not include time notation – a temporal sequence is assumed. However, we do include time notation in the SI for the sections in which we use the dynamic panel designs (SI S6b.i pp. 31-35, and Figures S6-S8).

If we had continuous monitoring of the system, we could look back in time and identify when changes in *R* and changes in *P* took place. However, as we describe in the main text (lines 493-495) and in the SI (p. S-24), the field measures of richness and productivity are taken at the same

time in our data. Thus, it is indeed possible that the DAGs we present are incorrect abstractions of the relevant causal pathways that are addressed in our Main Design. In particular, we acknowledge in several places (lines 71-72; 130-134; and especially 195-221; Figure 1 caption) the possibility for a directed edge from P back to R (i.e., a cyclic rather than acyclic graph). If the influence of that edge were to persist after we condition out the effect of time-invariant plot and site attributes and time-varying site attributes, we could have bias in our design. To address the threat from this bias, we use three different designs in our empirical analysis: the IV design, dynamic panel designs, and mechanism blocking design.

4. The authors investigated the potential reverse causality. However, they did not investigate more complex feedback through environmental variables (such as nutrients). Also in reality, the effect of P on R occurs together with the effect of R on P and confounding factors on P. Furthermore, as admitted by the authors, the effect of P on R is not invariant (e.g., can be linear positive or negative or hump-shape, or no effect). Thus, their statistical models are unrealistic. Again, my fundamental concern is that the linear model(s) developed by the authors cannot solve the BEF causality problem in real-world observational data.

We hope that our text in the preceding responses has addressed the issues raised here, which were raised in your earlier comments. You wrote that you have a “fundamental concern that the linear model developed by the authors cannot solve the BEF causality problem in real-world observational data.” We hope we have allayed your concern. If not, it would help us to see exactly – i.e., mathematically – what is the “BEF causality problem” and how exactly our design fails to “solve” it. We want to reiterate that our goal is not to model the grassland ecosystems in our data set. Instead, we aim to estimate an average causal effect over space and time, as randomized experiments seek to do. If we were aiming to accurately model the system, functional form assumptions would be very important and would have to be different from the ones we use for estimating the conditional expectation function (CEF) that we seek to estimate. For us, confounding variables are a major concern for estimating our target causal effect – the same issue that motivates the use of randomized treatments to estimate average treatment effects. In the newly added Methods section of the main text, we clarify our target causal effect (see p.12 in this response document) and the assumptions under which each design in our analysis can estimate this effect without bias from observable data.

5. The dynamic panel design did not consider feedback (reverse or through the third variable such as nutrient). Mathematically speaking, the linear models employed by the authors do not possess the capacity to accommodate the time-varying dynamical feedbacks.

We hope that our text in the preceding responses has addressed the issue raised here with regard to dynamic feedbacks, which were raised in your earlier comments. Our Main Design addresses the potential for hidden biases from dynamic feedbacks among confounders. Moreover, the IV design and the mechanism-blocking design in the main text address the threat

of bias from within-year (cotemporaneous) feedbacks between richness and productivity, and the dynamic panel designs in the main text and the SI consider the threat of bias from intra-year feedbacks (including, for example, via changes in nutrients). In particular, we want to emphasize that the IV design, when the assumptions are met, eliminates bias from feedbacks from productivity in current and prior years and from other variables, like nutrients, in current and prior years. Given how well developed mathematically the IV design is (in terms of how it works and what causal effect it can identify), and that it was originally developed decades ago specifically to address reciprocal feedbacks in complex systems, we are surprised that the reviewer asserts that it “does not possess the capacity to accommodate time-varying dynamical feedbacks.” The linearity of the estimation model in the IV design in no way prevents us from estimating our target average causal effect. To read more about these methods, please see the references 11, 73-76, 77, and 80, lines 491-538 in *Methods*, and section S6a.ii in the SI.

6. The findings presented in this study are not well organized, and explanations on their analysis are unclear throughout the main text. Figures are full of texts, but the ideas of their analysis remain poorly explained (Fig. 2, 3). In particular, the main differences in five analytical scenarios in Figure 3 are not clearly presented. It is difficult to tell how these scenarios are relevant to the ‘Common Design’ or ‘Main design’ in figure 2. Fig. 4 is just like making up by three separate ppt slides. It also includes too many statistical jargons from DAG and is unfriendly to the general audience. Therefore, it is difficult to follow the main text that is not self-sustained without checking SI in detail. Even worse, the SI itself is not well-organized. It is hard to clarify the main text and find the important supplementary materials, e.g., various confounding variables, assumptions, and analytical designs.

In response to this comment, we have made major revisions to reorganize the text – most notably revising the Study Design section and adding a thorough Methods section so that a reader does not need to refer to the SI to understand the analyses. In response to your request, the Methods section now provides an organized explanation of our analyses, with particular attention to (1) “the main differences in five analytical scenarios in Figure 3” and (2) how the different analyses relate to our Main Design. With the exception of some changes to the formatting of Figure 4 (requested by R2), we opted to keep our figures unchanged based on feedback from the other reviewers. Other reviewers, including two without prior exposure to the empirical methods, reported that the figures are not only helpful in understanding the designs (R2, R4) and concepts, but also in ensuring that the required assumptions for drawing causal inferences from each design are transparent (R5).

Minor comments:

P3, L43-44: Jochum et al. 2020 Nature Ecol. Evol. Also explored the realistic sequence of biodiversity loss.

In our revised manuscript, we include Jochum et al. Specifically, we added text in the Discussion to briefly make a connection between that study and our study (lines 293-296, and *pasted below in this response*). The Jochum et al. study focuses on using two sites to compare *some* compositional features of natural communities to post-treatment, compositional features of experimental communities. There are key differences between their aims and our aims. Unlike our study, Jochum et al. *do not test* if richness in the natural communities has different effects on productivity than richness in the experimental communities has. Also, unlike our study, they *do not estimate* effects for the observed, natural communities. From conversations with Dr. Malte Jochum, we have learned that one reason they did not estimate these effects is precisely because of the challenges of empirically controlling for the influence of confounding variables. Unlike our study, they did not have a large panel data set across many sites, which one could leverage, as we did, to control for confounding variables (they have two sites, whereas we have 43 sites).

We believe that their study is relevant to ours because the implication of their study is that the ecological attributes of experimental and non-experimental communities are similar, which contrasts with our conclusion. However, Jochum et al. only compare experimental and natural communities for 12 attributes, which they combined into a single index via principal components analysis (the attributes were SLA, seed mass, leaf P, leaf N, height, and several community composition metrics comprised of evenness, functional diversity of those traits mentioned). Importantly, the authors do *not* compare the communities in terms of their compositions of rare, non-rare, native, or non-native species groups, which we believe is driving the difference between our results and the results of many (but not all) experiments. In support of our conjecture about what drives the difference, we have also added a new citation for an experiment in which researchers experimentally removed non-dominant species from randomly assembled communities and reported an increase in biomass one year after the removal (Schmid et al. 2022, *Grassland Research*). That result is consistent with our results.

Furthermore, Jochum et al. compare a *static snapshot* of experimental and natural communities, but do not contrast temporal variation in richness in the communities (i.e., how numbers and types of species are changing from year to year), which is the variation that our Main Design uses. Thus, it is unclear whether the Jochum et al. comparison implies that natural and experimental communities are similar across all important dimensions that would affect the causal relationship between diversity and productivity.

In sum, we connected the Jochum et al study to a broader discussion in our manuscript about assessing the external validity, or the extent to which inferences can be generalized, of BEF experiments in light of the many dimensions over which natural and experimental communities can differ. This added text says at lines 292-296 says “Our results also highlight the challenge of determining the representativeness of experimental systems. For example, a recent comparison of natural and experimental systems identified many similarities in attributes but did not assess whether the patterns of rare or non-native species in experimental systems match the patterns in natural systems⁵².”

P4, L67: I don't think the methods applied in this study are the basis of any one of the Nobel Prize in 2021. Which one of the Nobel Prize the author meant?

In response to a comment from R2, we cut this statement from the paper. The 2021 Nobel Prize in Economics was awarded to two scholars who have made advances in developing and applying these design-based approaches to causal inference to economic research questions (see the links below for details).

- <https://www.nobelprize.org/prizes/economic-sciences/2021/angrist/lecture/>
- <https://www.nobelprize.org/prizes/economic-sciences/2021/imbens/lecture/>

P6, L89-90: What does 'external validity' mean?

We replaced the term 'external validity' with 'generalizability' in the text. External validity is the extent to which inferences can be generalized (e.g., across sites, time periods, contexts, or scales); see the Glossary in SI Section S1. External validity contrasts with "internal validity", which, as described in our glossary, is the extent to which a study design allows one to infer a causal relationship from a correlation by ruling out rival explanations.

P6, L96-97: Not clear about the importance of repeat observations on revealing various confounding variables. What are those confounding variables exactly?

In response to this comment, we have rewritten the Study Design section to try to better explain why repeated observations of each plot help us eliminate the confounding variation in our design, and why that is important (lines 104-129). As noted earlier, we also add a Methods section to the main text to explain in more detail how the estimation works (see, for example, equations 2 and 3).

With regard to the question "what are those confounding variables exactly?", we listed examples in the manuscript in Figure 1B (e.g., topography, weather, micro-climate, measurement error, land-use history, herbivory, disturbance). This comment seems to be asking us to provide an exact list of the unobserved confounding variables. The advantage of our design, in comparison to prior designs in Ecology, is precisely that we do *not* have to identify or observe every possible confounder. As we note in the text (lines 107-109), ecological systems are so complex that to identify and measure them all would be a daunting challenge. Instead, like a randomized experiment, we seek to mitigate their influence through a combination of design and untestable assumptions, which we make transparent in the text. As noted in the text, we "account for the ecological complexity of grasslands without making strong assumptions about our ability to measure all confounding variables" (lines 69-71) and thus "[w]e improve upon prior observational studies by controlling for a broader suite of confounding variables without needing to directly measure them as covariates" (lines 115-116).

Our key messages are that (i) accurately measuring the myriad confounding at the site and plot levels is a serious challenge and (ii) our design, or more precisely our designs, does not

require one to inventory and accurately measure all confounders but instead leverages panel data to reduce bias from these confounding factors without observing them directly.

P6, L98-100: What are those assumptions exactly? In addition, the citation for S2, S4, S7, S9; >20 SI pages) is not precise enough to guide the audience.

At this line in the revised manuscript (line 101), we have added text to say where to locate details about these assumptions more precisely “(these assumptions are described in more detail in Figures 2 and 3, Results, and Methods).” Further, these assumptions are now also explained in detail in the new Methods section of the main text.

P6, L105-106: Again, the author mentioned ‘repeat observations’ but still lacks a clear explanation.

Our new Methods section makes the longitudinal data structure clearer: an observation comes from a plot p located within a site s in a year t (Equation 1, 2 and 3). The notation in Equations 2 and 3, and the text around them, also emphasizes the nested, longitudinal nature of the data. Moreover, Table S1 displays the number of plots (i.e., number of observations per year) included per site and year in the dataset. This data structure is referred to as “panel data” in the statistical literature.

In the Study Design section, we describe the data structure in several locations (e.g. “repeated observations on the same plots and sites to control for confounding variables” on lines 97-98 and “our multi-site data is longitudinal (‘panel data’) and thus includes variation in species richness in both time and space” on lines 118-119).

P7, L120-121: The quantification of 10% increase cannot be inferred from bivariate correlation analysis.

This comment refers to the simple bivariate correlation. We did not claim in the original manuscript that this correlation represents a causal relationship. The bivariate correlation analysis serves only to compare our inferences to the inferences that were made using prior designs with this data (i.e., Adler et al. 2011 *Science* using bivariate correlations as a strawman). To make our point clearer, we revised this paragraph in two ways: (1) we changed the verb used from “increases” to “is associated with”; and (2) we added a sentence to emphasize the implausible assumptions that would be required to imbue this correlation with a causal interpretation (line 146-151):

“We first report the bivariate correlation between plot richness and productivity. Consistent with prior studies³³, we find a statistically weak, positive relationship between richness and productivity when we do not control for any confounding variables: a 10% increase in richness is associated with a 1.4% increase in productivity, 95% CI [-0.6, 3.4]. To give that estimated correlation a causal interpretation requires an implausible assumption that there are no confounding variables in the system (or that they perfectly cancel each other out).”

P8, L136: What is the set of confounding variables concerned in this study that makes BEF negative? Can they be pointed out specifically?

We hope that our response above with regard to your similar comment about text on p.6, lines 96-97, also answers this question; i.e., the motivation for our design is precisely that pointing out all of the potential confounders is challenging and the advantage of our design is that one does not have to point them out. We make clear in the manuscript that time-varying, site-level confounding variables are a big source of the bias that is missed in traditional ecological designs. We show that not controlling for them leads to a spurious, positive estimate (Figure 2 – Common Design). These confounding variables are likely to include attributes of weather, pollination, and herbivory that are not measured in the annual surveys at the sites. But we can only speculate on what exactly are the missing confounders and we do not see what purpose such speculation would serve in the manuscript.

Figure 2: In panel B, what is the difference between (i) & (ii). They look confusing. Panel A allows time-varying confounder at site level, but I don't understand how this design is critical to their analysis. Overall, this conceptual diagram poorly explains the motivations for these designs, and I cannot find the main ideas in the main text.

The figure contrasts not only the different estimates between the Common Design and our Main Design but also the difference in their underlying assumptions for assigning a causal interpretation to the estimated correlations (the estimated effects). As we note in our manuscript (lines 115-135 and lines 403-473), our Main Design controls for a much broader set of potential confounders and comes to the opposite conclusion about the relationship between richness and productivity. Figure 2 highlights how much stronger (and thus less credible) the assumptions are in the Common Design. The difference between assumptions (i) and (ii) is in whether the confounding attributes change over time (e.g., weather, herbivory) or do not change over time (e.g., topography, historical land use).

We hope that the newly added Methods section, which includes the estimating equation and a full discussion of what factors each element in the equation controls, will make these concepts clearer to the reader. For example, the time-varying site attributes (assumption (ii) in panel B) are modeled in a fully flexible way that allows a year-specific effect for each site (in the estimation, an indicator for each year is interacted with an indicator for each site). In other words, it controls, in a very flexible way, for factors that affect richness and productivity at the site level and change in value each year.

P10, L159-160: I don't agree such an argument that the more controls we have the more unbiased results we will obtain. It is correct only if those controls can be correctly considered. However, I don't see robustness analysis to validate the way of including additional confounders in their statistical models.

The sentence in the text to which this comment refers to is: “The *Main Design*, with its greater set of controls [for confounders], is thus less biased [than the Common Design].” The added text in brackets follows from the prior sentences, which focus on illustrating why the estimated effect goes from positive in the Common Design to negative in our Main Design. In this paragraph, we are making the point that the sign switch arises primarily because the Common Design does not control for time-varying site-level attributes. In the quoted text, we are not making a general claim that bias always goes down when one adds an additional control variable. In fact, our analysis shows precisely the opposite: when one moves from no controls (bivariate correlation) to many controls for observable attributes (Common Design), the bias gets worse (a phenomenon that is sometimes called “point-by-point bias” in the statistical literature).

P10, L165: There are five designs presented in Fig. 3 rather than four.

We edited this text to clarify that we are presenting four additional approaches, i.e., in addition to the Main Design already presented in Figure 2. We edited the sentence accordingly (line 192): “Using four additional approaches, we use different assumptions than made in our Main Design (Figure 2A) and assess how our conclusions change (see SI: Section S6).”

P11, L180: Lack of clear explanation about ‘conditioning’. This is a statistical jargon.

We revised this sentence to remove the term ‘conditioning.’ Throughout the Study Design section, we have revised the text to reduce jargon and, instead, add intuition (see lines 104-129).

Figure 3: This figure fails to convey a clear idea among different designs. The effect size panel with inconsistent names to the titles of other panels. Overall, there are too many words and statistical jargons without clear definitions in this figure.

We opted to keep this figure because other reviewers (R2, R5) point out that the figures are helpful for readers to understand the designs (R2) and critically important for allowing us to be transparent about our assumptions (R5) for causal interpretations of the estimates.

P17, L256: I don’t see this information from realistic sites presented in figure 4.

We do include information from panel C for our sites, specifically plotting the rank abundance curves for all sites in the data in Figure S10, which we paste here for convenience:

Figure S10. Most species are rare in these grassland ecosystems. Rank abundance curves (RAC) for each Nutrient Network site in our analysis shown in Table S1. These RACs are for the pre-treatment year, which we use to define species as rare or non-rare. Here, species abundance is based on relative live cover at the site-level.

The purpose of the other panels (A & B) in Figure 4 are conceptual. They accompany the discussion and provide an illustration of the ideas. Specifically,

- Panel A presents a visual (conceptual) definition of a heterogeneous treatment in the context of research on biodiversity-ecosystem functioning.
- Panel B presents a visualization of our hypotheses for the potential mechanisms that could underlie a negative average treatment effect, which we discuss as avenues for future experimental work on lines 274-277; 297-302; and 315-317.
- We do not have the data to test these hypotheses, but we propose them as fruitful explorations for future experiments and data collection. Reviewer 3 highlights these hypotheses as beneficial and exciting. We view panel B as an illustration of our ideas from the discussion and promising ideas to test in future research because the focus in most of the literature on this topic has been on mechanisms (e.g., complementarity) that could lead to a positive effect of positive on functioning, rather than a negative or neutral effect that we find here. Also see negative effects in the experimental work of Schmid et al (2022, *Grassland Research*), consistent with our results and hypotheses herein.

Response to Reviewer 2

Thank you for your positive feedback highlighting the quality of the writing, the ecological insights, and the novelty of our study. We also appreciate your detailed comments, which helped

us improve the clarity of our exposition and ensure that our study will influence a broad audience of ecologists. In response to your comments, we made substantial revisions, which we hope improved the clarity of the exposition in the main text and supplemental materials.

In this study, Dee and co-authors investigate the relationship between natural gradients of plant diversity and biomass productivity, using novel methods (within the discipline of ecology) for causal inference. A main finding is that, contrary to expectations, a higher species richness led to declines in productivity, which was mostly caused by rare-non-native species typically not used in biodiversity experiments.

I found the study very well written. I appreciated both the ecological insights of the study, i.e. that when correcting for biases caused by observed and especially unobserved covariates, causal effects of richness on productivity appear to be negative instead of positive (mostly caused by rare / invasive species), as well as for using a new analytical approach, that has so far hardly been used in ecological studies.

Thank you for this positive feedback on the writing, ecological insights, and novelty of the approach in ecology.

At the same time, I have to admit that I am no expert on the analytical approach that was used, and that I hope that another reviewer can assess the validity of its use for this paper better than I can. That said, I would have appreciated some more details on the analytical approach, as well as on the empirical data, in the main manuscript. At present, the main manuscript does not have any methods section, and the methods are either (very) briefly described in other sections (introduction/results), or in the supplements. Given that many readers might not read the supplementary materials, it would be good if the authors give more details on e.g. the data (currently in S3) and the Main Design Estimator (currently in S4) in the main manuscript. Obviously, I would not expect full details here, but I definitely would appreciate a bit more information on the methods in the main manuscript.

We agree that more information would be useful for readers. In response, we reorganized, revised, and elaborated to provide more intuition about how the methods work in the main text where we introduce the Study Design (at lines 104-129). In response to this comment, we have also added a Methods section in the main manuscript (see final section starting at line 327), which includes information on the Main Design (previously in SI Section S4), along with additional text providing high-level details needed to understand the core ideas and assumptions of all the other analyses we performed and the rationale for why we did them. We hope this additional section clarifies the key aspects of our analyses without requiring a reader to review the detailed supplemental materials.

Another point that I felt was missing, is that while one of the main points of this manuscript is that B-EF relationships are more negative when accounting for unobserved variables (something typically overlooked by other observational studies), there is hardly discussion on what these unobserved variables could be. The authors corrected for >50

environmental variables, which is (as they also emphasize themselves) much more than most other studies do, but despite this, apparently there are other, not explicitly considered factors, also important in altering correlations between species richness and biomass production. It would be very interesting to know what these unmeasured factors could have been. This would also lead to testable predictions of future studies.

Our key messages are that (i) accurately measuring the myriad confounding at the site and plot levels is a serious challenge, and (ii) our design, or more precisely our designs, does not require one to inventory and accurately measure all confounders but instead leverages panel data to reduce bias from these confounding factors without observing them directly. In the main text, we listed examples of potential confounders in Figure 1B (e.g., topography, weather, micro-climate, measurement error, land-use history, herbivory, disturbance). This comment seems to be asking us to provide an exact list of the unobserved confounding variables. The advantage of our design, in comparison to prior designs in Ecology, is precisely that we do *not* have to identify or observe every possible confounder. As we note in the text (lines 54-56), ecological systems are so complex that to identify and measure them all would be a daunting challenge. Instead, like a randomized experiment, we seek to mitigate their influence through a combination of design and untestable assumptions, which we make transparent in the text. As noted in the text, we “account for the ecological complexity of grasslands without making strong assumptions about our ability to measure all confounding variables” (lines 70-71) and thus “[w]e improve upon prior observational studies by controlling for a broader suite of confounding variables without needing to directly measure them as covariates” (lines 115-129). The motivation for our design is precisely that identifying them all out is challenging and the advantage of our design is that one does not have to identify them. We make clear in the manuscript that time-varying, site-level confounding variables are a big source of the bias that is missed in traditional ecological designs. We show that not controlling for them leads to a spurious, positive estimate (Figure 2 – Common Design). These confounding variables are likely to include attributes of weather, pollination, and herbivory that are not measured in the annual surveys at the sites. But we can only speculate on what exactly are the missing confounders and we do not see what purpose such speculation would serve in the manuscript.

I also have a number of more detailed comments, given below:

Line 65-67: I would mention the name of the approach here.

We use a suite of designs that make different assumptions, so there is no single name for the approach to casual inference in our study. To make clearer that we use a suite of approaches, we edited this sentence at line 67.

Line 67: “which were the basis for a 2021 Nobel Prize”: I would remove this, as it reads like an (implicit) argument based on authority on why the method is suitable, which I don’t think is the type of argument that is most valid.

We agree. We removed this statement.

Line 67: although I realize the analytical approach is explained in more detail in the methods section, it would be nice to see one or two sentences explaining the core of the approach. Perhaps this can be done by adding another panel in figure 1.

We added text to provide more intuition about how our study design works in the Study Design section at lines 104-129 and 175-182. We highlight the main changes at lines 115-144 below for your convenience. Further, we added a Methods section with additional detail on the designs and approaches, but less detail than is presented in the comprehensive SI. Specifically see lines 391-473. The Methods section also provides the core equations used (equations 2 and 3 for the main design).

“We improve upon prior observational studies by controlling for a broader suite of confounding variables without needing to directly measure them as covariates (see Methods and ^{37,41}). To understand the intuition for how this control is possible, recall that, in contrast to most prior observational studies on this topic (e.g., ^{5-7,33,37}), our multi-site data is longitudinal (‘panel data’) and thus includes variation in species richness in both time and space. Confounding variables that affect both richness and productivity could arise from conditions at the plot or the site. The values of these variables may be essentially invariant during the study period (e.g., soil texture, topography, land-use history) or they may vary through time (e.g., surrounding land-use change, drought conditions that differ by both site and year). With our multi-site panel data, we can directly control for time-varying, site-level conditions, whether they are observable or not, via a regression estimator that includes a simple interaction of binary variables for each site and year (see Methods, Equation 2). Further, we can eliminate the confounding effects of time-invariant plot and site conditions by taking deviations from mean conditions, after which variables that do not change over time no longer have any explanatory role and thus are eliminated as a source of bias (Methods). Using alternative designs, we can also quantify the potential threat of additional sources of bias from unobserved, time-varying plot-level confounders and from reverse causality (by bias from reverse causality, we mean bias that could arise when a causal effect also runs from productivity to richness; see Methods). In contrast to our approach, virtually all observational analyses reviewed in ⁸ omit important confounding variables (e.g., from human activities and land management) and ignore the potential for reverse causality (reviewed in ⁴⁶).

To demonstrate how our study design builds on and advances prior research, we first apply two study designs that have been used in prior studies and then contrast them to our design. Specifically, we estimate a simple bivariate correlation of richness and productivity (like ³³) and then we estimate the relationship between richness and productivity using a multivariate design

that mirrors advanced statistical designs that aim to control for confounding variables by directly measuring and including them as covariates in regression models (a “conditioning on observables” analysis, like ⁷). The multivariate design, which we label “Common Design in Ecology,” controls for over 60 variables (far more than prior studies), including attributes of the soil, habitat, historical management, and weather (Table S10). More details are provided in Methods section.”

Line 74-75: it would be good to shortly mention which climatic regions/ecosystem types are represented by the data

We agree. In the newly added Methods section, we have added additional detail about the ecosystem types covered in the data, which are: mesic grasslands and prairies, savanna, desert grasslands, montane meadows, old fields, and alpine tundra. We also added to the caption of Table S1 the fact that the dataset includes sites from 11 countries and 5 continents (North America, Australia, Europe, South America, and Africa).

Fig. 1A: This figure suggests that in biodiversity experiments, there is no relationship between the proportion of common/rare species and species richness. While this may be true when such experiments are established (in most cases, they initially have maximum evenness), over time, in most cases some species become more dominant, while others become rarer or even extinct. So I would argue that it is premature to make this assumption, especially since many biodiversity experiments have publicly accessible data, so it would be relatively straightforward to either support, or reject this assumption after some simple analyses.

In response to this comment, we revised the horizontal axis of Figure 1A for experimental communities to be “Planted Species Richness,” which is a better way to communicate the intent of the values in our figure. We agree that, in experiments, the post-treatment abundances and numbers of species often differ from what was planted. However, researchers using experimental data analyze the *randomized treatment of planted richness*, not the post-treatment compositions, which is why we focus on the treatment of planted richness in Figure 1’s panel (a) for experiments. Said another way, researchers analyze the randomized treatment of *planted richness*, not the *realized richness* that occurs after a few years of establishment when proportions of species are increasingly uneven. Thus, we believe the relevant comparison is between the randomized treatment of planted richness in experiments and the observed richness in natural communities. Note that analyzing the effects of the “realized richness” in experiments would be subject to the same challenges of identifying causal effects in observational data; e.g., the realized richness and its effect on productivity are potentially influenced by many confounding variables.

To further clarify our intent, we also elaborated the description of this panel in the figure caption:

“A. Experimental designs permit credible causal inferences with few modeling assumptions. Yet experiments often manipulate richness in random permutations, plant limited sets of species, and weed out colonizers. Such designs can yield ecological processes that differ from processes in natural systems. In panel A, when common species are more likely to be planted in experiments than rare species, the proportion of common species is higher than the proportion of rare species regardless of the planted richness level. In contrast, in natural communities, higher richness is associated with more rare species than common species.”

We believe that, given our intent in this figure, additional analyses of experimental data would not provide any further insights.

Fig 1B: in the caption, the authors rightly point out the possibility that productivity can also affect richness (“However, data on Z and M can help address unobserved confounders and differentiate the effect of R on P from the effect of P on R.”). It would make sense to also depict potential effects of P on R in the figure.

We note the potential effects of *P* on *R* in the figure caption, and we include a figure with feedback from *P* to *R* in Figure S4. Fig 1B is already very complex. Cognizant that we are introducing a lot of new ideas for many ecologists, we opted to not make the figure more complex by presenting it as a cyclic (recursive) causal graph (rather than a directed acyclic causal graph, or DAG). Our main emphasis in this figure, and the corresponding text in the introduction, is on confounding variables in observational studies rather than reverse causality, which we address later in the manuscript.

Line 134-135: it would be good to mention a few of those un-accounted variables.

As we noted above, the advantage of our design, in comparison to prior designs in Ecology, is precisely that we do not have to identify or observe every possible confounder. The reviewer seems to be asking us to list the unobserved confounders that have a negative selection effect. We make clear in the manuscript that time-varying, site-level confounders are a big source of the bias that is missed in traditional ecological designs. These confounders are likely to include attributes of weather, pollination, and herbivory that are not measured in the annual surveys at the sites. But we can only speculate on what exactly are the missing confounders and we do not see what purpose such speculation would serve in the manuscript. Our key messages are that (i) accurately measuring the myriad confounders at the site and plot levels is a serious challenge, and (ii) our design, or more precisely our designs, does not require one to inventory and accurately measure all confounders but instead leverages panel data to reduce bias from these confounding factors without observing them directly.

Fig 2. I really like the schematic overview of the assumptions of both the main and common design.

Thank you.

Line 187-189: Figure 3 shows, however, that while the richness->productivity relationship is still negative, it is not significant anymore. I think this nuance should be added in the main text.

In the caption for Fig. 3, we write, “No evidence for a positive effect.” In all four approaches, the estimated effect is negative. We are not doing null hypothesis statistical testing with a binary cut-off to declare an estimate “significant” or not. For the instrumental variables (IV) design, and to a lesser extent the lagged dependent variable design, the confidence intervals overlap zero, but they overlap zero for predictable reasons based on theory and knowledge about these estimators, which we explain in the Methods section and below. In our revision, and as described below, we have added nuance and interpretation for why we think these designs generate estimates with 95% confidence intervals that overlap zero.

First, for the IV design, we have revised the text to clarify why the confidence intervals are larger than those in the Main Design. Using the IV design, we give up precision in an attempt to remove bias. The Methods section makes this point clearer for readers who are unfamiliar with IV designs (see lines 491-538): “When this assumption is valid, the design addresses both reverse causality and all forms of confounding in Figure 1B, at the cost of drawing inferences from only a subset of the data, which can dramatically decrease the precision of the estimate.” We would expect the IV’s confidence intervals to be larger because it’s well established that the IV estimator is less efficient, which makes the confidence intervals larger. This disadvantage of the IV design is laid out well in (Morgan & Winship 2015, *Counterfactuals and causal inference*) page 303 in their book: “By using only a portion of the covariation in the causal variable and the outcome variable, IV estimators use only a portion of the information in the data. This represents a direct loss in statistical power, and a result IV estimators tend to exhibit substantially more expected sampling variance than other estimators.”

Second, econometric theory shows that estimates from a lagged-dependent variable (LDV) design (Ashenfelter 1978 *Rev. Econ. Stat.*) and our Main Design (using two way fixed effects estimator) can bracket the true causal effect (Angrist & Pischke 2009 *Mostly Harmless Econometrics*; Ding & Li 2019 *Polit. Anal.*). The Main Design and a LDV design rely on different assumptions for identifying a causal effect (Angrist & Pischke 2009, *Mostly Harmless Econometrics*) and therefore give upper and lower bounds to the true causal effect. Specifically, the lagged-dependent variable (LDV) design (see SI section S6bi. Dynamic panel designs) assumes that the observed and unobserved confounding variables are dynamic (i.e., from prior years). The Main Design assumes the confounding variables are site and plot specific but cotemporaneous (within year). By making different assumptions, these two approaches “bracket” the true coefficient estimate when one or both sets of assumptions are valid. We expand upon this bracketing approach in the main text (at lines 557-584 and below for this response).

As observed in Fig. 3B, this bracketing exercise in our paper implies the true effect is negative, albeit potentially smaller in absolute value than the estimate in Fig. 3A. As expected

based on theory around bracketing (Angrist & Pischke 2009, *Mostly Harmless Econometrics*; Ding & Li 2019 *Polit. Anal.*), the estimated effect from the LDV model is less negative (smaller in absolute magnitude) than the Main Design. Here the LDV provides an upper bound on the effect of richness on biomass. Still, this estimate implies a negative effect, with the upper 95% confidence interval overlapping zero, and the lower 95% confidence interval overlapping the estimate from the Main Design.

We have added this additional explanation about bracketing effects to the text at lines 557-584 in the newly added Methods section:

“In this analysis, we consider the possibility that prior productivity affects both current richness and productivity. We re-estimate the effect of richness on productivity using a lagged-dependent variable (LDV) design⁷⁹, which relies on different causal assumptions for identifying a causal effect. The Main Design assumes that the relevant confounders are time-invariant over the study period, or they vary over time at the site level rather than the plot level (e.g., site and plot-level differences in evolutionary history, age in community assembly, grazing intensity at a site, and history of disturbances and recovery stage in each plot). Instead, the LDV design considers: what if current species richness and productivity were determined by last year’s productivity, in addition to, or instead of, site-level conditions varying through time (e.g., precipitation)? The LDV design, in contrast to the main design, assumes that the relevant confounders vary over time at the site level and, at the plot level, their static and dynamic effects can be controlled by controlling for past productivity (in other words, the effects of confounders are mediated directly and indirectly through prior productivity at the plot level). To achieve this control, we estimate an equation of the following form:

$$\ln \text{LiveMass}_{i,t} = \beta \ln \text{Richness}_{i,t} + \theta \ln \text{LiveMass}_{i,t-1} + \mu_{i,t} + \varepsilon_{i,t} . \quad (6)$$

Under certain conditions^{11,80}, the estimated effects of richness in our Main Design and in the LDV design “bracket” the true causal effect. If the assumptions of the Main Design are valid, but the LDV design are invalid, the estimate from the LDV design provides an upper bound estimate. If the assumptions of the LDV are valid, but the Main Design are invalid, the estimate from the Main Design provides a lower bound estimate. As observed in Fig. 3B, this bracketing exercise implies the true effect is negative.”

Figure 4B: For clarity, I suggest to replace “R” by “richness”

We agree. We updated this figure to simplify the text for clarity and moved much of the text to the caption, as well as spelling out *R* as “richness.”

Fig 4B, right side: it is a mathematical inevitability that when richness increases, the average relative abundance of species decreases: this is as true for natural, as for experimental systems. Therefore, I am not entirely convinced whether this helps to explain differences with experimental systems. Of course, in natural systems, abundance distributions often tend to be log-normal (and hence rather un-even), which is different

from starting conditions in most BD experiments. After a few years of establishment, abundance distributions in experiments are much more un-even than under starting conditions, although still on average (but not always) less uneven than in natural situations (Jochum et al, NEE, 2020). So my feeling is that while natural systems differ indeed in relative abundance distributions from experimental systems, a bit more nuance is needed in this figure, as currently one might interpret unevenness as something unique for natural systems.

We think that the reviewer may have misread the axes of Fig. 4B – *right side*. Apologies if we are incorrect in this interpretation. But, to clarify, we are not plotting the average relative abundance of species of in a community against the number of species in that community. We are plotting a rank abundance curve, which shows the relative abundance of a *given species*, ordered from most to least abundant (from left to right). Each point is meant to represent a unique species (not a community with different average relative abundance plotted against richness). Here, the horizontal axis is the rank of individual species (not overall diversity). We do not write anywhere that unevenness is something unique for natural systems, and we do not see how one might draw that conclusion from Figure 4B. Thus, we did not change the figure. If we missed something in your comment and you can clarify, we would be happy to consider a revision.

In Figure 4B, the more important take away about the difference between natural and experimental communities is that most biodiversity-ecosystem functioning experiments do not plant a representative proportion of rare species, compared to the proportion of rare species that are found in many natural communities. To support this conjecture, we examined the “BigBio” and “BioCon” experiments in Cedar Creek, Minnesota, USA (included in Jochum et al.). We compared species composition in those two experiments to species compositions in long-term surveys of natural grasslands around the state of Minnesota, from Ratcliffe et al. 2022, *Ecological Applications*. The proportion of species that are rare in the experiments is much smaller than the proportion found in the natural communities. Few species that are rare in natural communities were planted in either experiment, and in cases when rare species are planted, they are less likely than non-rare species to establish (see Box 2 in FAQ #3 on page S-61).

We do, however, agree that citing Jochum et al. is a good idea. In our revised manuscript, we added text in the Discussion to briefly make a connection between that study and our study (lines 299-302). We believe that their study is relevant to ours because the implication of their study is that the ecological attributes of experimental and non-experimental communities are similar, which contrasts with our conclusion. However, there are four important distinctions between our study (and what we are trying to show in Figure 4) and Jochum et al, *Nature Ecology & Evolution* (2020), which we describe below. First, and most importantly, Jochum et al. (2020) do not study the same species features that we focus on in our Discussion. Jochum et al. compare some post-treatment compositional features of one experimental site and contrast them to features of one natural site. For the two sites, Jochum et al. used Principal Components Analysis

to create a single value for each site based on 12 plant-community properties: SLA, seed mass, leaf P, leaf N, height, and several community composition metrics (evenness, functional diversity of those traits mentioned). They then compare the PSA values. Yet, there are many dimensions that they do not include in their study, including the dimensions on which we focus: rare vs non-rare or native vs non-native species groups. Second, Jochum et al. compare *static snapshots* of the experimental and natural communities. They do not have any temporal variation – i.e., how numbers and types of species are changing from year to year – which is the variation that our Main Design uses. Third, in contrast to Jochum et al, who use only 2 sites, we use 43 sites across many countries, which enhances the generalizability of our results. Finally, Jochum et al do not estimate the difference in the effects of natural communities and experimental communities on productivity (i.e., test if their effects are the same). Through conversations with Dr. Malte Jochum, we discovered that one reason they did not estimate these effects is due to the empirical challenges of confounding variables that we highlight and address in our study.

So, to be more explicit about the distinctions between our studies, and the challenge of comparing experimental and natural communities, we added the following discussion at lines 299-302: “Our results also highlight the challenge of determining the representativeness of experimental systems. For example, a recent comparison of natural and experimental systems identified many similarities in attributes but did not assess whether the patterns of rare or non-native species in experimental systems match the patterns in natural systems.⁵²”

Fig. 5: it is remarkable that even for native, non-rare species (presumably those typical for BD experiments), the richness productivity is still not significantly positive (although not negative either), in contrast to relationships in BD experiments. In other words, when rare or invasive species are corrected for, the relationship switches from negative to neutral, but does not yet become significantly positive. It would be nice to add some discussion on this.

In the analysis for Figure 5, we separate the species into subgroups. The smaller sample sizes of each subgroup will reduce the precision with which we can estimate any single subgroup causal effect. Moreover, our Main Design uses the within-plot, temporal variation in richness to estimate (“identify”) the effect of richness on productivity (see newly added text in Methods section). The temporal variation in the numbers of native, non-rare species is not very large, as one can see in the figure below. That limited variation also reduces the precision of our estimate, i.e., it explains why there are large error bars around this estimate. The estimated effect, however, is our best estimate of the true effect. As noted above, we are not engaged in simple binary null hypothesis testing (i.e., declaring an effect to exist if $p < 0.05$ for the test of the null of zero average effect, and declaring it not to exist if $p > 0.05$). The estimated effect is positive, which is consistent with ecological experiments that plant native, non-rare species. We do not believe that one should attribute biological meaning to the size and location of the error bars. We present those bars so that readers can evaluate for themselves how informative the estimates in the figure are.

The figure above shows plot-level changes in non-native, non-rare species from year to year.

Response to Reviewer 3

Thank you for your positive feedback highlighting the quality of the writing, and the potential contributions from, and broad interest in, our study. We also appreciate your detailed comments, which helped us improve the clarity of our exposition. In response to your comments, we made substantial revisions, which we hope improved the clarity of the exposition in the main text and supplemental materials.

Dee et al. present a new design for analyzing the biodiversity-productivity relationship, drawing on the econometrics literature and leveraging an impressive dataset. Using their new analysis method, they found that biodiversity has a weak negative correlation with productivity in observational data. However, this statement obscures more interesting patterns related to rare and non-native species.

This is a thorough, well-written, and impressive contribution. I especially enjoyed the quality of writing in the introduction, the section on rare and non-native species, and the depth of sensitivity analyses in the supplementary material. The biodiversity-productivity relationship has been one of the major questions in ecology for decades and there is no question that this contribution would be of broad interest.

Thank you for this positive feedback.

My main concern is that I didn't get a clear sense of what the new analysis is actually doing. I don't doubt that the authors are using it correctly, but the manuscript (to me) spent too much time arguing for the novelty of this approach and/or comparing it to

previous analyses, and not enough helping the reviewer understand how the analysis works. Causal inference will not be familiar to many ecologists. I have made many suggestions in the attached pdf files (mostly the main text, but to a lesser extent the SI). In particular think it would be very helpful to walk through ecological examples when explaining the new analysis. To be clear, I think readers will buy the idea that previous observational studies missed some confounding variables. But I think to seal the deal you need to do more to explain to readers exactly how your new analysis improves on this issue.

Thank you for this feedback. It helps us to improve the clarity of our paper and its potential impact in terms of helping increase the adoption of these methods in ecology. In response, we have added substantially more text in the main manuscript to walk the reader through the analyses more thoroughly and clearly. Specifically, we completely revised the Study Design section and added a new Methods section. These revisions and additions include more intuition about the approaches and how they work, and more details on the equations. To view these substantial revisions, please see the attached file with track changes, particularly lines 115-135, and 175-182, and comparing the added Methods section on the Common Design versus Main Design.

Recall also that we also provided an Rmarkdown html tutorial (with all R files and the core data included), which carefully walks the reader through the ecological example from our study context to shed light on the rationale and interpretation of our study designs. We created this supplemental Rmarkdown file to be used as a tutorial for people who wish to learn these methods, as well as for classroom use. We believe additional examples from other ecological systems are out of the scope of this manuscript.

I enjoyed the FAQ section in the SI (even though it's not traditional, as the authors point out). I think the FAQ section is especially useful because the idea of positive biodiversity-productivity relationships is so widespread that there may initially be pushback against the authors' results, and the FAQ pre-empts many of these critiques. (I understand that the current manuscript supports the findings of BEF if one focuses on native, non-rare species).

Thank you for pointing out the utility of the FAQ section!

To address your additional comments from the marked-up PDF, we have cut and pasted the larger comments here:

(1) I believe that this clause is true and i'm excited to see the results, but...

(2) I think the writing is too emphasized on reminding/arguing that your approach is better than previous approaches. I'm repeating myself, I know, but I would like to see more about how the method works and less time trying to place this paper at the forefront of its subfield. If the method is as good as advertised the paper will naturally assume an influential place. (No need to respond to this, but just for context, I am very interested in quantitative methods, was already skeptical of the inference that could be drawn from

experiments, and am well convinced of the difficulty of observational studies. A different reader might prefer a different version of the paper.)

We included and retained the comparison between our approach and more common approaches because we received a lot of questions from our coauthors about why and how our design and results differed from prior analyses, namely Grace et al. 2006 *Nature*, which also used the Nutrient Network control plot data. Our coauthors asked questions like “How can we get different results using the same data?” and “Which result should we believe?” (for the full range of questions, see the FAQs #5-7 in the SI). Reviewer 1 also raised similar questions, and we anticipate that other readers of our study will also have them. For this reason, we have retained the comparison to common approaches to highlight that the key innovation of our paper is the study design, which weakens the assumptions required for causal inference compared to common designs used in ecology.

I don't feel like reading this paper has prepared me to use these analyses. If this is the "most important" thing, I think you need to do more to explain how others could use your method. I recognize that by digging through the SI and the code, one could probably figure out how to use the analysis. But my comment is that, if this is the most important contribution of the MS, it should be easier.

As noted above, we hope that the revisions to the main text, the newly added Methods section, the extensive supplementary materials, and the online tutorial and R Markdown will provide a solid foundation for readers who wish to use these methods. Note that we believe that our manuscript makes two important contributions: (1) it answers a central ecological question in the scientific literature using methods that provide new insights into the question; and (2) it introduces ecologists to these new methods and how they can advance ecological research.

Figure 3 -- I understand the goal here, and I liked the similar Figure 2, but I think this is too wordy in the figure itself. Is there a way to simplify the text or move some to the legend?

The credibility of our designs, as with all causal inference designs, depends on whether the assumptions are valid. Thus, we believe that the transparency of our approach (and the display of the assumptions in Figure 3) is its strength, as noted by Reviewer 5. We explained these assumptions as intuitively and comprehensively as we could in the main text and the SI. We believe Figure 3 is an important summary for readers who do not want to dig into the Methods or SI. From our perspective, one novelty and significance of our study is how we rigorously assess the robustness of our results by implementing designs that make different assumptions for drawing casual inferences from observational data.

We considered your suggestion of moving some of the text about assumptions to the legend, but we were worried it would be hard to follow if a reader needed to switch back and forth from

looking at the causal graphs (DAGs) and then the caption/legend. To simplify the figure, we could get rid of the DAGs, but then we would lose the thread from Figure 2, which you seemed to like (and we also like for showing how each design differs).

Response to Reviewer 4

Thank you for your positive feedback highlighting the quality of our study and its potential contributions. We also appreciate your suggestions on how we could improve the exposition. In response to your comments, we made substantial revisions, which we hope improved the clarity of the exposition in the main text and supplemental materials.

Review of “Clarifying the effect of biodiversity on productivity in natural ecosystems with longitudinal data and new methods for causal inference”

Summary: This manuscript applies methods from economics to estimate the arguably causal relationship between productivity and species diversity. The authors introduce the methods used by economist, and the show graphically how these methods can lead to causal insights. Their empirical example is convincing and finds a result that differs from what other studies have shown. Overall, the paper makes major contributions both methodologically and empirically. My hope is these methods will be more widely used in ecology, and the SI provides a useful primer on their use.

Thank you for this positive feedback on the contributions of our manuscript.

Major comments

1. The manuscript is well written and organized, yet it faces a very difficult challenge of introducing novel (to this field) methods while also showing a new, and perhaps unexpected, empirical finding that itself is of note for ecology. This is difficult to do in 3000 words. I have some suggestions for re-organization, although I encourage the authors to take them as suggestions, not orders.

We agree with your description of the challenge we face. Thank you for your helpful suggestions and encouragement.

a. Lines 101 – 106 feel unnecessary – the main point, that there are repeated timesteps is repeated in the next paragraph.

Per your other suggestions on adding more intuition about the design (and how the econometric fixed effects work), we did a major revision of this section of the manuscript by rewriting and reorganizing it. In doing so, we reorganized the Study Design section to better differentiate it from the Results section. As a result, the text flagged by your comment has changed, which we believe resolves this comment.

b. Lines 111-114 are also unnecessary – they do not actually discuss the study design, rather they are a claim about reverse causality. This claim, and the mechanism to control for reverse causality is discussed elsewhere.

We agree. We have removed this sentence.

c. I suggest the authors explicitly introduce the econometric idea of a “Fixed effect” in the main text. The “main design” is instantly recognizable to anyone familiar with econometrics as a panel regression with plot level fixed effects, yet this is never stated in the main text (although it is described nicely in the SI). I suspect the text will be easier for the non-economist if fixed effects are explicitly defined in the text and then this term is used in the text.

We revised the manuscript to add a multi-page Methods section at the end of the manuscript, which introduces and defines the econometric idea of a “fixed effect” in the main text as suggested by your comment (lines 391-471). We also heavily revised to the Study Design section to provide more intuition about how the approach works, why we use it, and how it compares to a more traditional conditioning on observables strategy. However, to avoid confusion, we do not explicitly use the term “fixed effect” outside of the Methods section and the SI. We avoid using that jargon because most ecologists have a different definition in mind when they use, or read, the term “fixed effect”; specifically, the notion of “fixed effects” in the context of mixed effects models, which are used more prevalently in ecology (see SI Section 1: Glossary, where we contrast the two uses of the term on page S 4-5).

d. Fixed effects are described as “controlling for a much broader set of potential confounders.” In other places they are described as “for a broader suite of confounding variables.” Personally, I find this an unintuitive way to describe fixed effects. It makes it sounds as though you know all the extra variables that are controlled for in the fixed effects model, when in fact one does not know what these unobservables are. Likewise, the way this is written, I would think many readers would say “why not just add these confounders to the mixed effects regression.” It may not be clear that what is being controlled for is unobserved. And of course, when you run the fixed effects model you have far fewer independent variables in the model, since all the time invariant variables are absorbed by the fixed effect.

Your comment raises a good point. Indeed, several of the other reviewers read the paper and thought we knew – or could know – the identities of all of the variables controlled for in the fixed effect models, which of course is not the case! To remedy this confusion, we did a major revision on the text describing what the fixed-effects estimator does, why we use it, and how it improves upon common approaches used in ecology. Here, we highlight the areas of major revisions:

- We clarified the assumption made by a *Common Design* in Ecology, which requires measuring and controlling for confounding variables directly in the regression (lines 110-

114): “In other words, prior studies require a strong assumption for interpreting the correlation between richness and productivity as causal: any site or plot attributes not included in the statistical estimation model are assumed to be uncorrelated with species richness and therefore not a source of statistical bias. Our design relaxes this strong assumption.”

- We clarified how the econometric fixed effects work, with more intuition in the Study Design section (lines 123-129): “With our multi-site panel data, we can directly control for time-varying, site-level conditions, whether they are observable or not, via a regression estimator that includes a simple interaction of binary variables for each site and year (see Methods, Equation 2). Further, we can eliminate the confounding effects of time-invariant plot and site conditions by taking deviations from mean conditions, after which variables that do not change over time no longer have any explanatory role and thus are eliminated as a source of bias (Methods).”
- We added text and equations to show how the error term in the Common Design (Equation 1) could contain several confounding variables that are removed by our Main Design in Equation 2 (and explained how they are removed, e.g. Equation 3). See Methods section lines 368-447.
- In the Methods section, we now refer to the SI section where we provide a brief comparison to mixed effects models and why we believe our approach is preferable to a mixed effect modeling approach for achieving our goals (see lines 451-452, SI Section S7).

i. You might consider instead describing the fixed effect as absorbing variation in the dependent variable that is the result of time invariant variables – both observed and unobserved.

Although we do not want to introduce more economics jargon like “absorbing” into our manuscript, we do agree, as noted above, that we needed to improve the explanation of the methods. As described above, we have added text to better describe what the fixed-effects estimator does, why we use it, and how it improves upon common approaches used in ecology.

e. Personally, I found the equations in the SI very helpful for understanding the three models. I think most people who will be interested in the models might find these useful as well. A potential reorganization would be to replace the last two paragraphs of the study design section with these three models.

We agree and thank you for this useful suggestion. We now include equations in the new Methods section.

f. This will have the added benefit of cleaning up the results section. Currently the results are a mixed of methods “We next estimate the relationship using a “Common Design in

Ecology” – a multivariate design that mirrors advanced statistical designs that aim to control for confounding variables by directly measuring and including them as covariates in regression models (a “conditioning on observables” analysis like7). Such designs require a strong assumption for interpreting the correlation between richness and productivity as causal36: any site or plot attributes not include in the regression are assumed to be uncorrelated with species richness 129 and therefore not a source of statistical bias” is really a methods paragraph, not a results paragraph. Moving this to the study design section, and adding an equation might be helpful.

We agree. Following your suggestion, we reorganized the Study Design and Results sections, including moving to the Study Design section the text that your comment flagged. In adding a Methods section to the main text of the paper, we also added the equations for the Common Design and for the Main Design (see Equations 1 and 2).

Response to Reviewer 5

Thank you for your positive feedback highlighting the novelty of our study design and the transparent analysis and discussion in the manuscript. We also appreciate your detailed and careful look at our manuscript and supplemental materials! In response to your comments, we made substantial revisions, which we hope improved the clarity of the exposition in the main text and supplemental materials and strengthened the reliability of our conclusions.

Summary

The study investigates the causal effects of biodiversity (richness) on ecosystem function(productivity) on longitudinal data from 43 grasslands in 11 countries using a sophisticated fixed-effects methodological study design. Contrary to prior observational and experimental work, the authors find that increases in plot-level richness causes productivity to decline rather than increase. The authors attribute the differences to observational studies to prior work not controlling for further confounding factors, and to experimental studies since these plant fewer rare and non-native species than exist in nature. While increases in native species increase productivity, increases in rare species decrease productivity, which explains the difference.

Overall recommendation

I am not deeply familiar with the application field here to assess novelty and significance in this respect. But the authors transparently lay out in much detail the differences and relations to prior work. Their novelty lies in the fixed-effects causal approach and I can comment on this methodological part. I very much liked the transparent way the authors communicate the assumptions and how effects change for different sets of assumptions.

Figures 2 and 3 (and more in the SI) are a beautiful example of this. These allow for a transparent and informed discussion of how other assumptions might change results. I have some questions on several potentially relevant details. If these are better analyzed and explained, I would recommend the paper for publication.

We are glad that you appreciate the transparency of our text, both in terms of how we communicate the differences between our study and prior studies and how we communicate our causal assumptions. Below, we respond to each of your comments and questions.

Methodological approach

The authors' goal is to quantify the causal effect of plant diversity richness R on biomass productivity P from purely observational data in the presence of hidden confounding. While previous studies tried to overcome confounding by conditioning on a large number of covariates, the authors here make use of panel time series ("t") data at different sites "s" and plots "p" and then use the fixed-effects causal effects design:

$$\ln(P_{pst}) = \beta \ln(R_{pst}) + \delta_{ps} + \mu_{st} + \epsilon_{pst}$$

where β is the quantity of interest. Here hidden confounding by time-invariant plot attributes can be de-confounded by de-meaning (or introducing plot-dummy variables δ_{ps}), and hidden confounding by site-specific time-varying variables is attributed for by μ_{st} .

As the authors nicely and transparently lay out, this model depends on a number of assumptions that are extensively discussed and further tested in the comprehensive and well-written supplement.

Your comment is correct in its characterization of the assumptions used in our Main Design. It is also correct in highlighting that we do not simply assert that these causal assumptions are valid but instead clarify the degree to which our conclusions change when our assumptions change.

In the following I list some questions that I still have:

Positive effect in prior studies

The authors explain this by "First, on average, many of the observed site and plot variables [...] are negatively associated with richness and positively associated with productivity [...] unobserved site and plot variables (U_s , U_p , U_{st} in Figure 1B) are, on average, positively associated with both richness and productivity. Failing to control for them in the Common Design creates statistical biases in the positive direction. The Main Design, with its greater set of controls, is thus less biased"

You further expand on this in the SI, but still: Why is it that, coincidentally (?), the *observed* covariates tend to have a negative effect and the *unobserved* covariates tend to have a positive effect? The larger set of unobserved covariates includes the observed ones. You observe these effect directions, but do you have a good explanation for this "covariate-selection" bias here?

The advantage of our design, in comparison to prior designs in Ecology, is precisely that we do not have to identify or observe every possible confounder in these complex ecological systems. The Nutrient Network sites are likely to experience site-specific “shocks” that vary each year (e.g., weather shocks, like a particularly dry April, or herbivory shocks, like higher herbivore pressure than the prior year). In our data, we observe that these shocks affect productivity and richness in the same direction, on average. We don’t know what exactly these shocks are, but because we observe the same sites over many years, we can control for them. We now revise the manuscript to clarify this point at lines 176-177.

Your comment seems to be asking us to list the unobserved confounders behind these shocks. We make clear in the manuscript that time-varying, site-level confounders are a big source of the bias that is missed in traditional ecological designs. These confounders are likely to include attributes of weather, pollination, and herbivory that are not measured in the annual surveys at the sites. But we can only speculate on what exactly are the missing confounders and we do not see what purpose such speculation would serve in the manuscript. Our key messages are that (i) accurately measuring the myriad confounders at the site and plot levels is a serious challenge, and (ii) our design, or more precisely our designs, does not require one to inventory and accurately measure all confounders but instead leverages panel data to reduce bias from these confounding factors without observing them directly.

Controlling for Reverse Causality by blocking a mediator

Here the potential causal feedback $R \rightarrow P \rightarrow M \rightarrow R$ with M =shading is investigated. You block the M here, but M is a "forbidden node" in causal effect estimation since it is a child of the effect and induces a bias in estimating $R \rightarrow P$: However, you find no change in section S6b.i. That either means shading has no or only a weak relation with P or is maybe a cause of P , contrary to your graph above. Or what could explain this?

In this part of the analysis, our concern is the potential threat to inference from within-year causal effects of P on R (i.e., within-year reverse causality). In our Main Design, we assert that, after conditioning on plot-level fixed effects and site-by-year effects, any correlation between R and P reflects the effect of R on P . But if the correlation instead were to reflect, in whole or in part, the effect of P on R , then conditioning on a mechanism that mediates the effect of P on R will induce bias, as noted by the reviewer. In other words, if our estimator is picking up the effect of P on R , then M (shading) is indeed a “forbidden node.” It lies on the causal path that is

being picked up by our estimator (i.e., the reverse direction from P to R). Thus, controlling for shading in our regression would induce bias that would be visible via a change in the estimated coefficient. However, if after conditioning on plot-level fixed effects and site-by-year effects, our estimator only identifies the effect of R on P , we would observe little change in the estimated coefficient when we add shading to the model – i.e., in that case, shading is not a forbidden node because it's not on the causal path identified by the estimator.

The reviewer asks why we don't see a change in the estimated coefficient. There are three main possibilities: (1) the effect of P on R is small in our sample of plots; (2) the effect of P on R is not small, but any mediating effect of P on R that arises from a change in shading is weak after conditioning on plot-level fixed effects and site-by-year effects (recall that, if there are other important mediators of P on R , our test has nothing to say about them); and (3) shading does not mediate the effect of P on R . Reason (1) and (2) support our conclusion that the Main Design is detecting the effect of R on P . Reason (3) implies our test is uninformative. Reason (3) is untestable, and we make that point clear in our main text by saying “If shading were not an important mechanism through which productivity would affect richness in our sample, or if our measure of shading is a poor measure of the shading mechanism, our mechanism-blocking design would fail to quantify the potential threat of reverse causality.” Our “mechanism blocking design” is often called a “test of known effect” in the causal inference literature because it relies on “knowing” that shading is an important mediator of P on R . We cannot test this assumption. If it's incorrect, then our “mechanism blocking design” reveals nothing about the threat of within-year reverse causality. The uncertainty over the veracity of this assumption motivates our instrumental variable design, which relies on a different, but plausible, set of assumptions to rule out the role of reverse causality in our negative estimated effect.

As noted in your comments, all causal inferences rely on untestable assumptions. We believe that the strength of our design arises from making these assumptions transparent and varying them in substantive ways to evaluate whether hidden biases could be masking a true positive effect of richness on productivity. We do not assert that any one of our designs is sufficient evidence to support a claim of a negative average effect of richness on productivity. Instead, we assert that the totality of evidence across all of the designs is sufficiently persuasive to warrant publication of our interesting results. In our study, the primary potential concern is that the true causal effect of R on P is positive, but our study design masks this relationship and erroneously points to a negative effect. We believe that the threat of such hidden bias is substantially mitigated by the totality of the evidence we bring forth in the manuscript.

Controlling for Reverse Causality by instrumental variables

Here, as always in IV-designs, the credibility depends on whether the assumptions of IV are fulfilled and as a non-domain expert I cannot assess this here, but the authors explain this quite comprehensively.

Correct. The credibility of this design, as with all causal inference designs, depends on whether the assumptions are valid. As your comment noted, we explained these assumptions as intuitively and comprehensively as we could in the main text and the SI. With our new Methods section, we hope the information will be even more accessible to readers. We drew on the field expertise of our two dozen plus co-authors to develop the most credible instrumental variable we could find for our context. We also note that none of the peer reviewers with topical expertise questioned our assumption that variation in out-of-sample, neighboring plot richness can affect variation in own-plot richness but not own-plot productivity. A lack of protest from reviewers, of course, doesn't validate the assumption.

As we noted above, the strength of our study arises from making our causal assumptions transparent and varying them in substantive ways to evaluate whether hidden biases could be masking a true positive effect of richness on productivity.

Dynamic panel designs

Here I see some potentially more problematic issues and would need some clarification since I cannot fully follow the analysis:

In reading your comments and re-reading the relevant sections in our SI, we realized that the SI could benefit from some re-organization and revisions to make the ideas clearer. Thank you for helping us clarify this section.

One error we made was to try to simplify the visuals for our target novice readers but forgetting to explain this simplification process in the SI section on dynamic panel designs (S6bi, starting on page S-31). In the previous version of the manuscript, we tried to direct the reader's attention to the new details in this section – namely the lagged temporal subscripts on R and P – and so we suppressed the spatial and temporal subscripts on the confounding variables. We compactly visualized all of them as time and space-invariant variables U and I . But, without any further explanation, that decision had the unfortunate effect of making it appear that only time-invariant confounders were an issue in the estimation that we do in these dynamic panel designs. In other words, we see now that the previous section on dynamic panel designs seemed disconnected from the Main Design and its concerns about both time-invariant and time-varying confounders. Furthermore, we believe we also may have caused some confusion by including lagged factors in Figure S4 in the prior Section 6b but not discussing them (instead saying we would discuss them in the prior Section 6c). Lastly, we believe we also did not clearly elaborate the thread that connects the analyses in the prior section S6b (on within-year reverse causality) to the analyses in the prior section S6c.

We have therefore thoroughly revised and reorganized the text and figures in SI Sections S6. We hope the section is now more logically organized and more comprehensive. We now start with our analyses to explore potential violations in our assumption that the effect we are estimating goes from richness to productivity, and not the other way around. Having demonstrated that we cannot detect any threat to our inferences from reverse causality, we then

explore potential violations in our assumption that there are no time-varying plot attributes that are systematically correlated with richness and productivity. In response to your comments below, we also added some supplemental analyses to this section. In none of the analyses do we find any evidence suggesting that hidden biases in our analyses are masking a positive effect of richness on productivity.

- The text in section S6c says that your approach cannot deal with both of the links Pt-1  Pt AND Pt-1  Rt, but Fig. S7 doesn't show the three remaining options and misses the one with Pt-1  Rt. Later the text clarifies that it is about not allowing a mediating path Pt-1  Rt  Pt

As noted above, in our effort to try to limit the length of our 80+ page SI, we left some paths out of the figure visualizations. In our revision of this section, we visually portray all of the possibilities and describe them in the text. We hope that not only is the text more complete, but it also makes clearer the connections between the dynamic designs and our Main Design. We now have (i) a revised Fig S6, which has both time-invariant plot/site confounders and time-varying site confounders (we eliminate the variable *I* that comprised factors that affected only productivity because that variable was not required to make our points in this section); (ii) a revised, multi-panel Fig S7, which reflects relationships that our Main Design can handle, and (iii) a revised, multi-panel Fig S8, which reflects relationships that our Lagged Dependent Variable design can handle.

Below, we show the visuals for each of the paths identified in your comments.

1. $P_{t-1} \rightarrow R_t, R_t \rightarrow P_t, \text{ and } P_{t-1} \rightarrow P_t$

In the revised manuscript, this figure is Fig S6, which eliminates the variable *I* and shows the two types of confounders that are the focus of the Main Design: time-invariant, plot-level confounders (U_p) and time-varying, site-level confounders (U_{st}).

2. $P_{t-1} \rightarrow R_t, \text{ but no others.}$

This figure is Fig S7, upper left panel (panel A), in the revised manuscript.

3. $R_t \rightarrow P_t$, but no other. This path was in the left panel of the original Figure S7.

This figure is Fig S7, upper right panel (panel B), in the revised manuscript.

4. $P_{t-1} \rightarrow P_t$, but no other.

This figure is Fig S7, lower left panel (Panel C), in the revised manuscript.

5. $R_t \rightarrow P_t$ and $P_{t-1} \rightarrow P_t$. These paths were in the right panel of the original Figure S7.

This figure is Fig S7, lower middle panel (panel D), in the revised manuscript.

6. $P_{t-1} \rightarrow R_t$ and $P_{t-1} \rightarrow P_t$.

This figure is Fig S7, lower right panel (panel E), in the revised manuscript.

7. $P_{t-1} \rightarrow R_t$, $R_t \rightarrow P_t$, and $P_{t-1} \rightarrow P_t$ [where time-invariant, plot level attributes only affect productivity]

This figure is Fig S8, left panel (panel A), in the revised manuscript.

8. $P_{t-1} \rightarrow R_t, R_t \rightarrow P_t$, and $P_{t-1} \rightarrow P_t$ [i.e., time-invariant, plot level attributes do not affect R or P or, implausibly, only affect R_t and P_t but not P_{t-1}]

This figure is Fig S8, right panel (panel B), in the revised manuscript.

Note: We exclude the permutation from your comment where $P_{t-1} \rightarrow R_t$ and $R_t \rightarrow P_t$ and no other direct edges exist. We exclude it because this pattern is not ecologically possible, i.e., a case where prior productivity had an effect on current productivity and richness would be the sole mediator or, equivalently, prior productivity has no effect on current productivity except through its effect on current richness.

Recall that our Main Design handles all scenarios in Fig S7, and our Lagged Dependent Variable design handles the scenarios in Fig S8.

For the data generating process illustrated by revised Figure S6 (cut and pasted again below for your convenience), we now present two designs.

In the first design, which was in our original SI, we draw on the approach of Beck and Katz (2011) and assume that the persistent causes of productivity across years (the part of U_p that goes to P_{t-1} and P_t) comprises autoregressive disturbances of order 1, AR(1), and thus the effect of R on P can be estimated using an autoregressive distributed lag equation of order 2 in

autoregression and order 1 in distributed lags (Beck and Katz, 2011, *Annu. Rev. Polit. Sci.*). In the second design, recommended in your comments below, we combine the Main Design and Lagged Dependent Variable design and we re-estimate the effect of richness on productivity, noting the potential bias in the combined estimator.

In both designs, we obtain a negative estimated effect with a similar magnitude and precision to the estimate from the Main Design (SI pp. 35-36). In the first design, we infer that a 10% increase in richness leads to a 2.4% decrease in productivity [95% CI: -4.4, -0.4]. In the second design, we infer that a 10% increase in richness, on average, decreases productivity by 2.8% [95% CI: -4.6, -1.1].

- Regarding Fig. S6: The confounders U and I are both assumed invariant (=constant) over time. Does this mean that they are plot-specific attributes? But the fixed-effects setting can deal with this by adding a dummy-plot variable to the regression (as in Eq. (3)), no? This dummy would in the graph block paths through U and I . Then the only confounding would occur through P_{t-1} , but that one can just be conditioned on. I don't see any confounder path in the graph then. If, however, one *cannot* condition on U , which is a direct confounder of R_t and P_t , then the case is indeed non-identifiable. The confounding due to I can always be conditioned away by including P_{t-1} in the regression, even if it was time-varying.

As noted in our introductory response on the dynamic panel designs, the decision to present U and I as time-invariant was an expositional choice. We now see that the choice caused confusion and reduced clarity about what we did, so we revised this section and the figure. As requested in your comment, we have added to the SI a model that has both plot-level fixed effects (dummy-plot variables) and a lagged productivity variable (P_{t-1}). Note, however, that we do not know of a way, without more assumptions, to estimate a consistent model (in the statistical sense) that has both plot-level fixed effects and a lagged productivity variable. We explain the reasoning in more detail below, but this estimation challenge is why we just estimated the model with P_{t-1} (Lagged Dependent Variable design) and explained how the combination of the fixed-effects estimator (Main Design) and the Lagged Dependent Variable estimator bound the treatment effect, under certain assumptions. In re-reading our SI text, we recognized that we could have been clearer about this bracketing result (i.e., the results in Table S9 bracket the true causal effect when either the causal assumptions of the Main Design or the Lagged Dependent Variable design are true). We now write, in the Methods section the following text at lines 557-584:

“In this analysis, we consider the possibility that prior productivity affects both current richness and productivity. We re-estimate the effect of richness on productivity using a lagged-dependent variable (LDV) design (Ashenfelter 1978), which relies on different causal assumptions for identifying a causal effect. The Main Design assumes that the relevant confounders are time-invariant over the study period, or they vary over time at the site level rather than the plot level (e.g., site and plot-level differences in evolutionary history, age in

community assembly, grazing intensity at a site, and history of disturbances and recovery stage in each plot). Instead, the LDV design considers: what if current species richness and productivity were determined by last year's productivity, in addition to, or instead of, site-level conditions varying through time (e.g., precipitation)? The LDV design, in contrast to the main design, assumes that the relevant confounders vary over time at the site level and, at the plot level, their static and dynamic effects can be controlled by controlling for past productivity (in other words, the effects of confounders are mediated directly and indirectly through prior productivity at the plot level). The LDV design, which also includes the prior year's live mass, estimates:

$$\ln \text{LiveMass}_{\#t} = \beta \ln \text{Richness}_{\#t} + \theta \ln \text{LiveMass}_{\#(t-1)} + \mu_{\#t} + \varepsilon_{\#t} .$$

Under certain conditions (Angrist & Pischke 2009; Ding & Li 2019), the estimated effects of richness in our Main Design and in the LDV design “bracket” the true causal effect. If the assumptions of the Main Design are valid, but the LDV design are invalid, the estimate from the LDV design provides an upper bound estimate. If the assumptions of the LDV are valid, but the Main Design are invalid, the estimate from the Main Design provides a lower bound estimate. As observed in Fig. 3B, this bracketing exercise implies the true effect is negative, albeit potentially smaller in absolute value than the estimate in Fig. 3A (consistent with the theory on bracketing, see (Angrist & Pischke 2009; Ding & Li 2019)).”

To narrow the brackets or generate a point estimate, one would need to make different assumptions. In our analysis in the SI, we explore one of set of plausible assumptions: we assume that the persistent causes of productivity across years (I) comprise autoregressive disturbances of order 1, AR(1). With that assumption, the effect of R on P can be estimated using an autoregressive distributed lag equation of order 2 in autoregression and order 1 in distributed lags (Beck and Katz, 2011). This approach is represented by Equation 6 in our SI. The estimated effect is similar to our main estimate in Fig. 2: a 10% increase in richness leads to a 2.1% decrease in productivity, 95% CI [-4.0, -1.8]. Another approach would be to find an instrumental variable for P_{t-1} . In some non-ecology studies (Arrellano and Bond, 1991, *Rev of Econ Studies*), scholars use the dependent variable lagged two periods: P_{t-2} in our case. But the assumption that P_{t-2} is uncorrelated with $(E_t - E_{t-1})$, where E_t is the error term, seems unlikely to be reasonable in our context. Thus, we do not implement this estimator (the length of our panel may also be too short for this estimator to be consistent).

With regard to the challenge of creating a consistent estimating model by combining the Main Design and the Lagged Dependent Variable design in one estimator (i.e., combining plot-level fixed effects and lagged productivity), the details can be found in the original article from Nickell (1981, *Econometrica*) or the textbook we cite in our manuscript (Angrist & Pischke 2009, *Mostly Harmless Econometrics, Sections 5.3-5.4*). The idea is that combining the two features can create a correlation between regressors and the error term in the model, which makes the estimator inconsistent (i.e., the estimator may not converge in probability to the true

value of our target parameter even as the number of plots in our panel goes to infinity). This potential problem is not caused by an autocorrelated error process. The problem can arise even if the error process is *i.i.d.* If the error process is autocorrelated, the potential problem is even more severe.

Nevertheless, for completeness, we now present the results of exactly this combined estimator and explain in the SI why it may be biased (pp. S-35-36). The estimated effect of richness on productivity is larger in absolute value than the estimated effect in our Main Design: a 10% increase in richness, on average, decreases productivity by 2.8% [95% CI: -4.6, -1.1].

- The graphs in Fig. 7 would also present no problem using a plot-dummy variable. Here one would *not* need to condition on Pt-1. (However, conditioning on Pt-1 would reduce estimation variance.)

We believe your comment refers to Figure S7, and yes, we agree that our estimator in the Main Design addresses the cases presented in Figure S7. The text explains that “all panels in Figure S7 are allowed [in our Main Design]” (p. S-33). As we noted above, adding lagged productivity to the Main Design is not straightforward, but we did so and found that the estimated effect is larger in absolute value (more negative).

- The effect in the graphs in Fig. 8 both can be de-confounded by Pt-1.

Yes, that insight is our motivation for presenting the results in Table S8. As we note in the text, “The causal processes implied by the causal graphs in Figure S7 and S8 are observationally indistinguishable.” Thus, we supplement our Main Design with an estimator that conditions on P_{t-1} .

- Eq (5): It think what I don't get is why you don't include a dummy-plot variable together with a dynamic graph setup here. I have actually implemented a toy model to test this and it works.

As we noted above, including a plot fixed effect (“dummy-plot variable”) together with a lagged dependent variable in dynamic panel model (i.e., with lagged, plot-level productivity) creates a potentially inconsistent and biased estimator. As explained above, for completeness, we now present the results of exactly this combined estimator and explain in the SI why it may be biased (p. S-35-36) (see also Nickell 1981, *Econometrica*). The estimated effect of richness on productivity is larger in absolute value than the estimated effect in our Main Design: a 10% increase in richness, on average, decreases productivity by 2.9% [95% CI: -4.6, -1.1].

- Fig S4: Here, if we assume away the reverse causality of Pt on Rt, then conditioning on the time-dummy variable μ_{st} would block all non-causal paths through ϵ_{st} . From the graph there is no need to condition on the many factors as in Eq. (6). Rather, these would

just increase estimation variance since you remove variance from the cause. It could be that I have just misunderstood or overlooked something here, but please clarify.

As noted above, we believe we created some confusion by introducing lagged variables in Figure S4 when that section is focused on within-year reverse causality, not biases that arise from lagged variables (the focus of the next section in the SI). We have removed those variables from the figure. Having removed those variables, we believe there is nothing further to address with regard to your comment.

Nonlinearity

The authors state "[Log-transforms] permit the underlying relationship in levels (untransformed variables) to be non-linear, while still preserving the regression model's assumption of linearity"

A log-transform only makes a fully multiplicative equation linear, that is, if

$P = R * \text{confounders} * \text{noise}$, then

$\log(P) = \log(R) + \log(\text{confounders}) + \log(\text{noise})$

But if $P = c R + \text{confounders} + \text{noise}$, then the regression model would not be as desired, no? I am not sure your other tested regressions $\log(P) = \beta R$, $\log(P) = \beta R^2$ etc cover all the cases. In essence, I would like to see what the assumed nonlinearity is that your transformation then makes linear.

We thank the referee for this comment and acknowledge that the text from the SI that the reviewer quoted was unclear, and we have revised it and include it at the end of this response, for your convenience. Below, we reiterate the motivation for the natural logarithm transformation using the revised text, which we added to the SI Section 3b (pp. 11). For further clarification, we also address the role that linearity (or its absence) plays in our estimation.

Our main point is that given our target parameter is an average causal response (a conditional expectation function; see comments to R1 on pp. 3-4), our estimation models that use fixed-effects are not making strong assumptions about the functional form of the relationship between richness and productivity or among the confounders, richness, and productivity. Using our estimation models, we are averaging across deviations of richness and productivity from their means in the sample and thus can accommodate a wide range of non-linearities. So, our model is quite a flexible one for estimating our target causal effect (average change in productivity given a small change in richness). As evidence for these statements, this attribute of our design is one reason why the different specifications in Table S3 all yield a negative estimated relationship between richness and productivity.

To elaborate, consider the estimation model in Equation 2 of our Methods section:

$$\ln LiveMass_{pst} = \beta \ln Richness_{pst} + \delta_p + \mu_{st} + \epsilon_{pst} \quad (2)$$

We eliminate the effect of δ_p by taking deviations from plot-level means. Thus, we can also describe our estimation strategy as estimating a correlation between deviations of productivity around its mean and the corresponding deviations in richness around its mean -- regardless of the relationship between those means. That implies that one could interpret our model as assuming separability between (a) deviations in richness from its plot and site-year means and (b) those means and confounders. In this case, transforming R and P by taking the natural logarithm of their values facilitates OLS estimation and has three desirable statistical properties. First, both productivity and richness are strictly positive variables that exhibit right-skewed distributions (see Figure S1). Transforming by the natural logarithm reduces the skew of these variables, improving statistical efficiency (i.e., improves the precision of our estimates). Second, in an ecological sense, it is reasonable to assume that going from 2 to 4 species will on average have a bigger effect on productivity than going from 18 to 20 species but may have a similar proportional effect on average as a change from 10 to 20 species would have. In other words, the natural logarithm transformation makes sense in situations when it is better to compare relative changes rather than absolute changes. In other words, instead of assuming that P increases as a constant function of R , we assume that P increases as a relative function to the current level of P as a function of R . Another way to say the same thing is that in a graph with richness on the horizontal axis and productivity on vertical axis, a straight line will not be the best description of the relationship. Third, the coefficient on richness in our ln-ln specification has a well-defined interpretation, which is a valuable trait; for most readers, a single coefficient is more accessible and easier to evaluate than a non-linear surface.

To emphasize that we are not making any strong assumptions about the functional form of the relationship between richness and productivity and that, in fact, our model captures non-linearities between richness and productivity, we elaborate our estimation model here.

Consider the following reinterpretation of our estimation framework:

$$\ln NP_{pst}P = k \ln NR_{pst}PP + \delta_p + \mu_{st} + \varepsilon_{pst}$$

where $k(\cdot)$ is a continuously differentiable function; we assume this so that we can define $k'(x) = dk(x)/dx$. The non-linearities in the relationship between P and R are captured by $k(\cdot)$. Our target parameter (target causal response) is not necessarily the best-fit relationship between $\ln NP_{pst}P$ and $\ln NR_{pst}P$, but rather is interpreted as the average $k'(\cdot)$ across the levels of R observed in our data, conditional on plot and site-year fixed effects. That is, our effect is:

$$\beta = E_{\ln(R)} \{ dE \ln(P) / d \ln(R), \delta_p, \mu_{st} \}$$

where the outer expectation is over that values of R in the data. How does this target parameter relate to $k(\cdot)$? Well, $k(\cdot)$ is just the inner derivative:

$$k(\ln(R)) = dE \ln(P) / d \ln(R), \delta_p, \mu_{st}$$

(where here the expectation is implicitly across ε_{pst}). Thus $k(\cdot)$ represents the change in average productivity for a small change in richness. This value could—and likely does—vary by the level

of R and by the values of δ_p and μ_{st} (which, recall, are the plot and site-year averages). However, the outer expectation in our target parameter averages *over the observed levels of R in our dataset*. So, in our estimation, we recover is the average value of k in our sample, i.e., the average change in average productivity given a change in richness. Our regression estimator is well equipped to deliver this quantity under a set of reasonable assumptions. In the manuscript and the SI, we argue that this is a quantity that should be of interest to scientists precisely because it reflects the effects of changes richness in naturally occurring (non-experimental) ecosystems.

Revised SI text on this topic (pp. SI-10-11):

“Prior to estimating the effect of diversity on productivity, we transform our productivity variable (live biomass) and our diversity variables (richness, evenness, Simpson’s index) by taking the natural logarithm of each plot-level measure. This transformation has several advantages, which are all related in a statistical sense. First, both productivity and richness are strictly positive variables that exhibit right-skewed distributions (see Figure S1). Transforming by the natural logarithm reduces the skew of these variables, improving statistical efficiency (i.e., improves the precision of our estimates). Second, in an ecological sense, it is reasonable to assume that going from 2 to 4 species will on average have a bigger effect on productivity than going from 18 to 20 species but may have a similar proportional effect on average as a change from 10 to 20 species would have. In other words, the natural logarithm transformation makes sense in situations when it is better to compare relative changes rather than absolute changes. In other words, instead of assuming that P increases as a constant function of R , we assume that P increases as a relative function to the current level of P as a function of R . Another way to say the same thing is that in a graph with richness on the horizontal axis and productivity on vertical axis, a straight line will not be the best description of the relationship. Third, the coefficient on richness in our log-log specification has a well-define interpretation, which is a valuable trait; for most readers, a single coefficient is more accessible and easier to evaluate than a non-linear surface. In this SI (section S6), we also present the estimated effects of richness on productivity in levels (i.e., no transformation), including quadratic and cubic specifications that permit the estimated relationship to be non-linear.”

Heterogeneous Effects of Rare, Non-rare, and Non-native Species on Productivity
Here in Eq. (9) you estimate the individual **direct** effects of five different species categories on productivity adjusted for time-invariant plot-level and time-varying site-specific confounders. Now what about interactions among the different species? Suppose $R1 \text{ -(+)-> } R2 \text{ -(-)-> } P$ and $R1 \text{ -(+)-> } P$. Then you would measure a positive effect in the second part, while the **total** effect of $R1$ on P may actually be negative, depending on the strength of the indirect and direct path.

But the experimental setup, I would presume, analyzes the **total effect (including potential mediating paths through other categories) and not the direct effect. Hence, you**

might not be comparing like with like here. Could this alter your conclusions about the positivity and negativity of effects?

In our analysis in this section of the paper, we are pointing to possibilities that merit future research, including experiments (at lines 284-287; 301-306; 319-321). Thus, rather than view our analysis as confirmation that species traits matter, we view it instead as evidence that points to a hypothesis for further testing in experimental and non-experimental designs.

Your comment does raise an interesting thought experiment. As we note in the main text, experiments have too few rare or non-native species to estimate effects by species category as we do in our analysis. Thus, we don't know what would be estimated in these experiments if they did the type of analyses we do. However, we do add a citation to a recent experiment in which researchers experimentally removed non-dominant species from randomly assembled communities (Schmid et al. 2022, *Grassland Research*) and reported an increase in biomass one year after the removal; a result consistent with our results (lines 303-306). We add "For example, in a recent study²⁶, researchers experimentally removed non-dominant species from randomly assembled communities and reported an increase in biomass one year after the removal; a result consistent with our results. Extensions of their study can help elucidate the traits of these species that drive their effects on productivity."

Nevertheless, we agree with your comment that interactions among different categories (types) of species may complicate estimation and comparisons between experimental and non-experimental systems. We unfortunately do not have sufficient statistical power in our design to try to explore potential interactions across species. Yet, we do want to acknowledge this important issue. Thus, in our revision, we write (lines 279-281), "We acknowledge that there could also be positive and negative interactions across these species types, but we do not have sufficient statistical power to explore these potential interactions."

Lagged effects

You only consider lag-0 (within the same year) effects. What about lagged effects $R_{t-1} \rightarrow Pt$? Here I mean direct effects, of course with auto-causation in R also an instantaneous effect would lead to an indirect effect $R_{t-1} \rightarrow R_t \rightarrow Pt$. Maybe this issue is partly addressed in the FAQ on short vs long-term effects, but please clarify.

We did consider the effect of lagged richness $R_{t-1} \rightarrow Pt$. Please see Table S2 (columns 5 and 6). We realize that discovering all of our analyses can be challenging in a very long SI, so we are pasting the results here for ease of finding this analysis.

Table S2. The negative effect of ln of species richness on ln of productivity holds when clustering standard errors at the site level (column 2), when controlling for species evenness (column 3), and when using other measures of biodiversity (Simpson's Diversity – column 4) as

well as the lagged effect of species richness in the prior year ($\ln \text{SpeciesRichness}_{t-1}$) (columns 5 & 6). The estimated effect in column (1) is plotted in Figure 2.

Model with $\ln(\text{live biomass})$ as outcome:							
	(1)	(2)	(3)	(4)	(5)	(6)	(6)
$\ln(\text{SR})$	-0.2418 *** (0.08) [-0.40903; -0.0744]	-0.2418 ** (0.0892) [-0.4166; -0.0671]	-0.2237 *** (0.0851) [-0.3906; -0.0569]		-0.2187 ** (0.0939) [-0.4027; -0.0347]		-0.2059 ** (0.0948) [-0.3917; -0.021]
$\text{IHS}(\text{Evenness})$			-0.1866 (0.2122) [-0.6024; 0.2292]				-0.1453 (0.2387) [-0.6131; 0.3225]
$\ln(\text{Simpson})$				-0.1693 ** (0.0678) [-0.3023; -0.0364]			
$\ln(\text{lagged SR}_{t-1})$					-0.0143 (0.0904) [-0.1915; 0.1629]		-0.0093 (0.0903) [-0.1863; 0.1676]
Num. obs.	1231	1231	1231	1231	1093	1093	
R^2 (full model)	0.87	0.87		0.87	0.87	0.87	0.87

Signif. Codes: ***: 0.01, **: 0.05, *: 0.1. 95% CI are shown in bracket.

Robust Standard errors in parentheses, clustered at plot level in column 1 and clustered at site level in column 2.

Selection/missing sample bias

You mention missing samples, but what percentage is actually missing? And can there be selection bias or missing sample effects, i.e., the reason for missingness is related to richness or productivity? I guess not, but just a thought.

We are unsure about what your comment means by “missing samples.” We believe your comment is referring to the note in Table S7 in the SI, which mentions that some plot-year observations have missing richness values. Only two values (2 out of 1291) were missing and thus we are not concerned about potential bias from a relationship between missingness, richness, and productivity. We now make the number of missing values clearer in the text for the table. It’s also possible that your comment is referring to text in the SI about a site that had to be dropped for the analysis in Figure 5. We had addressed this issue in the SI (pp S-46 in revised SI): “Note that because we classify the species in the groups based on pre-treatment year data, the site “saline.us” is excluded from this analysis because the site does not have pre-treatment data. Dropping the 24 observations from the site “saline.us” does not change our main estimates in Figure 2 and 3.” In summary, we don’t think there is a threat of bias from missingness.

Typos

"how our conclusions change (see SI: section S6c)"

- this should be just "section S6)"

Fixed.

"casual"

- replace by 'causal'.

Fixed.

Final remarks

I would like to very much thank the authors for the whole way this article was written, the transparency, the figures, and the large SI were really a joy to read and made it easily accessible. I would hope more papers do this!

Thank you so much! We really appreciate this positive feedback.

As a side remark: I found it interesting that you use causal graphs throughout, but cite potential outcome literature instead of causal graphical models literature (eg Pearl only in SI). None of the references 12, 13 and 35 (that you often cite) contains a single causal graph.

In the manuscript, we reference Morgan and Winship in the main text, who use both DAGs and the potential outcome framework. We believe it's a nice Big Picture introduction to the concepts.

Morgan, S.L. & Winship, C. (2015). *Counterfactuals and causal inference*. Cambridge University Press.

Sincerely,

Jakob Runge

REVIEWERS' COMMENTS

Reviewer #2 (Remarks to the Author):

This is the second manuscript version of this study that I review. When I reviewed an earlier version of this manuscript, I was overall positive in my assessment. I had listed some concerns, and I am glad to see that the authors have addressed these overall very adequately. I only have one small concern left. Previously, I mentioned the almost absence of the methods in the main manuscript file. I was glad to see that the authors have addressed this. One remaining concern is that the data themselves (what kind of sites/ecosystem? How many sites? What is the time span?) is not mentioned yet in the main manuscript file outside the methods section. It would be good to (even if very briefly) mention the type of data in the introduction section.

Reviewer #3 (Remarks to the Author):

I was positive about this manuscript when it was first submitted, and like this version even more. I appreciate that the authors carefully considered my main comments, especially the addition of a dedicated Methods section and the fact that readers no longer need to read the Supplementary Information to understand the details of causal analysis.

I appreciate the authors consideration of shortening the Figure 3 legend. I see that there was conflicting advice among the reviewers, and I am fine with their decision to leave the legend as is.

Although I made little reference to this in my first review, I did want to offer my perspective on the Grace et al. (2016) issue. I think the authors have been very clear on how their work builds on Grace et al. (2016) and appreciated the detail they provided with this second revision.

Reviewer #4 (Remarks to the Author):

We thank the authors for their excellent work. I do not have any suggestions for further changes.

Reviewer #5 (Remarks to the Author):

Review NCOMMS-22-09180 revision

I thank the authors for their thorough revision.

My assessment of the comments of Reviewer #1 and the authors' response is as follows:

I agree with the authors that there are two different frameworks colliding here, the "intervention-based" notion of causality and the predictive notion followed by the convergent-cross mapping (CCM) approach and the EDM framework. The goal of CCM is mainly about what can be called *causal discovery*, i.e., to *detect* causation, rather than assume it exists and quantify it, as is the goal of this article. Importantly, both frameworks indeed have their distinct sets of assumptions and current capacities of dealing with various challenges. In the following, I will discuss these a little bit.

The predictive framework of CCM crucially assumes the existence of an (ideally low-dimensional) attractor that is then implicitly reconstructed using delay embedding with the free hyper-

parameters being the embedding delay and dimension. In my opinion the theory of this framework on dealing with unobserved confounding of the various forms treated in this article is much less developed. In the original CCM article it is simply assumed that, by successfully reconstructing the attractor, the whole state-space, including the influence of observed or unobserved confounders, is reconstructed (Taken's theorem). But if such a low-dimensional (and ideally not very noisy) attractor is **not** reconstructable, then the CCM framework performs poorly as demonstrated in many empirical studies. Krakovská et al (2018), Cobey et al (2016), and Runge (2018) all found an issue with inflated false positives with CCM. But in the present study, the focus is not on causal discovery.

Further, the theory of CCM is also not yet developed enough to deal with multiple heterogeneous datasets, as the authors note and also as far as I know. In summary, in my point of view, the article presents a thorough application of the intervention-based causal inference framework which merits publication and this task is, at present, not feasible to be addressed within the CCM framework.

Further, Reviewer #1 raises some points on advancements wrt to Grace (2016), confounding and feedbacks, and nonlinearity, as well as on the clarity of the article.

Grace (2016): In my opinion, the explanations address the distinction enough.

Confounding and feedbacks: Also here I think the authors have done a good job to explain that these two challenges can be addressed in this framework. Of, course, one can still go even further, but this can be the topic of subsequent research.

Nonlinearity: Reviewer #1's point that the underlying system is nonlinear implies that the authors have to address the issue of potential misspecification of the linear (log) model. The authors have, in the revision, spent much time on justifying that the model works also in a partially nonlinear setting and also clarified some confusion about which part of the model can be nonlinear.

Finally, I would think that the clarity has now been further improved, but still, this articles is challenging to read since it applies a deep and sophisticated framework, and a reader would have to learn more about it to fully appreciate its advantages.

The dispute regarding whether causal questions about nonlinear dynamical systems can be tackled in the causal inference framework is a very interesting one and I hope for many more workshops and joint papers from scientists of these two communities.

Regarding the responses to my raised points:

Wrt "Positive effect in prior studies", I understand that the idea of your design precisely is to **not** consider all possible confounders and my question was not about listing them. I am happy with the current sentences on the fact that the unobserved covariates tend to have a positive effect, rather than, for example, half of them having a negative and half having a positive effect leading to no change overall. Maybe you could mention that this fact is worth noting and can be investigated in further research?

I am also happy with the further explanations on my suggestion to include both dummy and lagged variables as a further test. I did not know that this design may be biased, although, from reading Nickell, that only applies to the setting with too few samples in the time dimension as compared to the plot dimension.

Regarding the log-transform and nonlinearity: If I try to summarize your point, then the log-transform is statistically beneficial since the marginal distributions of the data are skewed and it is

ecologically sensible. Further, your model assumes "separability between (a) deviations in richness from its plot and site-year means and (b) those means and confounders." and I leave it to domain-experts to assess whether that is a reasonable choice. This point is also further addressed in the responses to Reviewer #1.

Thank you also for further explaining the other aspects I raised and acknowledging the issue of species interactions, no further comments.

Hence, I would deem the article publishable now.

Jakob Runge

References

- * Krakovská, Anna, et al. "Comparison of six methods for the detection of causality in a bivariate time series." *Physical Review E* 97.4 (2018): 042207.
- * Cobey S, Baskerville EB (2016) Limits to Causal Inference with State-Space Reconstruction for Infectious Disease. *PLoS ONE* 11(12): e0169050. <https://doi.org/10.1371/journal.pone.0169050>
- * Runge, J. Causal network reconstruction from time series: From theoretical assumptions to practical estimation *Chaos An Interdiscip. J. Nonlinear Sci.*, 2018, 28, 075310

Response to Reviewers NCOMMS-22-09180-Revision

Below, **the reviewer comments are in bold** and our responses are in regular font.

Summary of Reviews

We appreciate the time and attention that the reviewers invested in re-reviewing our manuscript and their favorable responses.

- Reviewers 2, 3, 4 and 5 provided comments on the revised manuscript.
- *Only the comments of Reviewer 2 and Reviewer 5 require a response.* Reviewer 2 requested more details about the data, which was easy to address. To address it, we have added more detail to the manuscript on lines 79-85, including a reference to the Nutrient Network website. Reviewer 5 requested that we add a sentence that encourages future research on the specific components of the unobservable, but controllable, site-level shocks in our study, which can help researchers who do not have rich, longitudinal data like ours.
- **For completeness, we include the comments of the other reviewers in our response document.** In their comments, Reviewers 3 and 4 simply expressed satisfaction with our revised manuscript. Recall also that, in addition to expressing satisfaction with our revised manuscript, Reviewer 5 expressed satisfaction with our responses to Reviewer 1.

Response to Reviewer 2

This is the second manuscript version of this study that I review. When I reviewed an earlier version of this manuscript, I was overall positive in my assessment. I had listed some concerns, and I am glad to see that the authors have addressed these overall very adequately.

I only have one small concern left. Previously, I mentioned the almost absence of the methods in the main manuscript file. I was glad to see that the authors have addressed this. One remaining concern is that the data themselves (what kind of sites/ecosystem? How many sites? What is the time span?) is not mentioned yet in the main manuscript file outside the methods section. It would be good to (even if very briefly) mention the type of data in the introduction section.

In the section entitled *Study Context* (lines 73-76; lines 88-93), we did indeed describe the data, including the number of sites, the time period, and the type of ecosystems we are studying.

“We use repeated observations between 2007-2017 from 151 unmanipulated plots in 43 grassland sites in 11 countries¹⁰ (Table S1 in Supplementary Information (SI); SI Section 3). We define “productivity” as aboveground live biomass per year per 1m² (following^{6,7,33}). Each 1m² plot has between 1 and 37 species in a year, with an average of 11.3 (SD = 5.7) and median of

10. We use plots with five or more years of data, in contrast to most observational studies of biodiversity effects on productivity, which use a single year^{6,7,33,35,39}.”

In this paragraph, we agree we could be more specific about what kinds of grasslands we study, as we are in the *Methods*. Thus, in our revised manuscript, we move a relevant sentence from the *Methods* section to the second sentence of this paragraph:

“We use repeated observations between 2007-2017 from 151 unmanipulated plots in 43 grassland sites in 11 countries³⁴ from the Nutrient Network (<https://nutnet.org>), including mesic grasslands and prairies, savanna, desert grasslands, montane meadows, old fields, and alpine tundra (Table S1 in Supplementary Information (SI); SI Section 3). We define “productivity” as aboveground live biomass per year per 1m² (following^{3,10,31}). Each 1m² plot has between 1 and 37 species in a year, with an average of 11.3 (SD = 5.7) and median of 10. We use plots with five or more years of data, in contrast to most observational studies of biodiversity effects on productivity, which use a single year^{6,7,33,35,39}.”

Response to Reviewer 3

I was positive about this manuscript when it was first submitted, and like this version even more. I appreciate that the authors carefully considered my main comments, especially the addition of a dedicated Methods section and the fact that readers no longer need to read the Supplementary Information to understand the details of causal analysis. I appreciate the authors consideration of shortening the Figure 3 legend. I see that there was conflicting advice among the reviewers, and I am fine with their decision to leave the legend as is. Although I made little reference to this in my first review, I did want to offer my perspective on the Grace et al. (2016) issue. I think the authors have been very clear on how their work builds on Grace et al. (2016) and appreciated the detail they provided with this second revision.

We are pleased that we have addressed your concerns.

Response to Reviewer 4

We thank the authors for their excellent work. I do not have any suggestions for further changes.

We are pleased that we have addressed your concerns.

I thank the authors for their thorough revision. My assessment of the comments of Reviewer #1 and the authors' response is as follows:

I agree with the authors that there are two different frameworks colliding here, the “intervention-based” notion of causality and the predictive notion followed by the convergent-cross mapping (CCM) approach and the EDM framework. The goal of CCM is mainly about what can be called *causal discovery*, i.e., to *detect* causation, rather than assume it exists and quantify it, as is the goal of this article. Importantly, both frameworks indeed have their distinct sets of assumptions and current capacities of dealing with various challenges. In the following, I will discuss these a little bit.

The predictive framework of CCM crucially assumes the existence of an (ideally low-dimensional) attractor that is then implicitly reconstructed using delay embedding with the free hyper-parameters being the embedding delay and dimension. In my opinion the theory of this framework on dealing with unobserved confounding of the various forms treated in this article is much less developed. In the original CCM article it is simply assumed that, by successfully reconstructing the attractor, the whole state-space, including the influence of observed or unobserved confounders, is reconstructed (Taken's theorem). But if such a low-dimensional (and ideally not very noisy) attractor is *not* reconstructable, then the CCM framework performs poorly as demonstrated in many empirical studies. Krakovská et al (2018), Cobey et al (2016), and Runge (2018) all found an issue with inflated false positives with CCM. But in the present study, the focus is not on causal discovery. Further, the theory of CCM is also not yet developed enough to deal with multiple heterogeneous datasets, as the authors note and also as far as I know.

In summary, in my point of view, the article presents a thorough application of the intervention-based causal inference framework which merits publication and this task is, at present, not feasible to be addressed within the CCM framework.

We are pleased that the reviewer agrees with how we framed the contrast of the two causal frameworks and how we apply our framework, and that our manuscript merits publication.

Further, Reviewer #1 raises some points on advancements wrt to Grace (2016), confounding and feedbacks, and nonlinearity, as well as on the clarity of the article.

Grace (2016): In my opinion, the explanations address the distinction enough.

Confounding and feedbacks: Also here I think the authors have done a good job to explain that these two challenges can be addressed in this framework. Of, course, one can still go even further, but this can be the topic of subsequent research.

Nonlinearity: Reviewer #1's point that the underlying system is nonlinear implies that the authors have to address the issue of potential misspecification of the linear (log) model. The authors have, in the revision, spent much time on justifying that the model works also

in a partially nonlinear setting and also clarified some confusion about which part of the model can be nonlinear.

We are pleased that the reviewer agrees with our responses to Reviewer 1.

Finally, I would think that the clarity has now been further improved, but still, this article is challenging to read since it applies a deep and sophisticated framework, and a reader would have to learn more about it to fully appreciate its advantages. The dispute regarding whether causal questions about nonlinear dynamical systems can be tackled in the causal inference framework is a very interesting one and I hope for many more workshops and joint papers from scientists of these two communities.

We agree that readers will have to learn more about our framework to fully appreciate its advantages and we agree that our study can serve as input into important discussions and debates about how to tackle causal questions in nonlinear, dynamical systems.

Regarding the responses to my raised points:

Wrt "Positive effect in prior studies", I understand that the idea of your design precisely is to **not consider all possible confounders and my question was not about listing them. I am happy with the current sentences on the fact that the unobserved covariates tend to have a positive effect, rather than, for example, half of them having a negative and half having a positive effect leading to no change overall. Maybe you could mention that this fact is worth noting and can be investigated in further research?**

We are pleased that the reviewer is happy with our revision. With regard to mentioning that more research should be assigned to investigating why the observable covariates are associated with bias in one direction and the unobservables are associated with bias in the other direction, we added a sentence in the main text where we write that this topic could be investigated in future research (last line in paragraph below, lines 178-180 in the manuscript).

“Future research could elucidate what shocks are most relevant, thereby providing a way for researchers without longitudinal data to potentially control for the confounding effects of these shocks.”

I am also happy with the further explanations on my suggestion to include both dummy and lagged variables as a further test. I did not know that this design may be biased, although, from reading Nickell, that only applies to the setting with too few samples in the time dimension as compared to the plot dimension.

Yes, it applies to contexts like ours, where we have more plots than we have years per plot.

Regarding the log-transform and nonlinearity: If I try to summarize your point, then the log-transform is statistically beneficial since the marginal distributions of the data are skewed and it is ecologically sensible. Further, your model assumes "separability between (a) deviations in richness from its plot and site-year means and (b) those means and confounders." and I leave it to domain-experts to assess whether that is a reasonable choice. This point is also further addressed in the responses to Reviewer #1.

Thank you also for further explaining the other aspects I raised and acknowledging the issue of species interactions, no further comments.

Hence, I would deem the article publishable now.

We are pleased that you are happy with our revision and deem the article publishable. We want to thank you for the careful review and for adjudicating with Reviewer 1.